# Offline Learning for Combinatorial Multi-armed Bandits

**Xutong Liu** [1]  **Xiangxiang Dai** [2]  **Jinhang Zuo** [3]  **Siwei Wang** [4]  **Carlee Joe-Wong** [1]  **John C.S. Lui** [2]  **Wei Chen** [4]

## Abstract

The combinatorial multi-armed bandit (CMAB) is a fundamental sequential decision-making framework, extensively studied over the past decade. However, existing work primarily focuses on the online setting, overlooking the substantial costs of online interactions and the readily available offline datasets. To overcome these limitations, we introduce Off-CMAB, the first offline learning framework for CMAB. Central to our framework is the combinatorial lower confidence bound (CLCB) algorithm, which combines pessimistic reward estimations with combinatorial solvers. To characterize the quality of offline datasets, we propose two novel data coverage conditions and prove that, under these conditions, CLCB achieves a near-optimal suboptimality gap, matching the theoretical lower bound up to a logarithmic factor. We validate Off-CMAB through practical applications, including learning to rank, large language model (LLM) caching, and social influence maximization, showing its ability to handle nonlinear reward functions, general feedback models, and out-of-distribution action samples that exclude optimal or even feasible actions. Extensive experiments on synthetic and real-world datasets for these applications further highlight the superior performance of CLCB.

## 1. Introduction

Combinatorial multi-armed bandit (CMAB) is a fundamental sequential decision-making framework, designed to tackle challenges in combinatorial action spaces. Over the past decade, CMAB has been extensively studied (Cesa-Bianchi & Lugosi, 2012; Bubeck et al., 2012; Audibert et al., 2014; Neu, 2015; Gai et al., 2012; Kveton et al., 2015c; Combes et al., 2015; Chen et al., 2016; Wang & Chen, 2017; Merlis & Mannor, 2019; Saha & Gopalan, 2019; Zimmert et al., 2019; Liu et al., 2024b; Qin et al., 2014; Liu et al., 2023a; Choi et al., 2024; Hwang et al., 2023), driving advancements in real-world applications like recommendation systems (Kveton et al., 2015a; Li et al., 2016; Lattimore et al., 2018; Agrawal et al., 2019), healthcare (Lin & Bouneffouf, 2022; Verma et al., 2023; Bouneffouf et al., 2020), and cyber-physical systems (György et al., 2007; Kveton et al., 2015b; Li et al., 2019; Liu et al., 2023b).

Most success stories of CMAB have emerged within the realm of online CMAB[1], which relies on active data collection through online exploration. While effective in certain scenarios, this framework faces two major limitations. On one hand, online exploration becomes impractical when it incurs prohibitive costs or raises ethical and safety concerns. On the other hand, they neglect offline datasets that are often readily available at little or no cost.

For instance, in healthcare systems (Liu et al., 2020), recommending optimal combinations of medical treatments—such as drugs, surgical procedures, and radiation therapy—requires extreme caution. Experimenting directly on patients is ethically and practically infeasible. Instead, leveraging pre-collected datasets of prior treatments can help to make informed decisions while ensuring patient safety. Similar happens for recommendation systems (Chen et al., 2023) and autonomous driving (Kiran et al., 2020), offline datasets such as user click histories and human driving logs are ubiquitous. Leveraging these offline datasets can guide learning agents to identify optimal policies while avoiding the significant costs associated with online exploration—such as degrading user experience or risking car accidents. For more examples, see Appendix B for details.

To address the limitations of online CMAB, we propose the first offline learning framework for CMAB (Off-CMAB), where we leverage a pre-collected dataset consisting of $n$ samples of combinatorial actions and their corresponding feedback data. Our framework handles rewards that are

---

[1]ECE Department, Carnegie Mellon University, Pittsburgh PA, United States [2]CSE Department, Chinese University of Hong Kong, Hong Kong SAR, China [3]CS Department, City University of Hong Kong, Hong Kong SAR, China [4]Microsoft Research, Beijing, China. Correspondence to: Xiangxiang Dai <xxdai23@cse.cuhk.edu.hk>, Jinhang Zuo <jinhang.zuo@cityu.edu.hk>, Wei Chen <weic@microsoft.com>.

*Proceedings of the 42nd International Conference on Machine Learning*, Vancouver, Canada. PMLR 267, 2025. Copyright 2025 by the author(s).

---

[1]See Appendix A for a comprehensive discussion on the related works.

nonlinear functions of the chosen super arms and considers probabilistic feedback models that generalize the standard semi-bandit feedback model (Gai et al., 2010; Chen et al., 2013; Kveton et al., 2015c), supporting a wide range of applications such as learning to rank (Liu et al., 2009), large language model (LLM) caching (Zhu et al., 2023), and influence maximization (Kempe et al., 2003a). The objective is to identify a combinatorial action that minimizes the *suboptimal gap*, defined as the reward difference between the optimal action and the identified action.

The key challenge of Off-CMAB lies in the absence of access to an online environment, which inherently limits the number of data samples available for each action. Furthermore, this problem becomes even more challenging with the combinatorially large action space, which complicates the search for optimal solutions, and the potential presence of out-of-distribution (OOD) samples, where the dataset may exclude optimal or even feasible actions. To tackle these challenges, this work makes progress in answering the following two open questions:

*(1) Can we design a sample-efficient algorithm for Off-CMAB when the action space is combinatorially large? (2) How much data is necessary to find a near-optimal action, given varying levels of dataset quality?*

We answer these questions from the following perspectives:

**Algorithm Design:** To address the first question, we propose a novel combinatorial lower confidence bound (CLCB) algorithm that addresses the uncertainty inherent in passively collected datasets by leveraging the pessimism principle. At the base arm level, CLCB constructs high-probability lower confidence bounds (LCBs), penalizing arms with insufficient observations. At the combinatorial action level, CLCB utilizes an approximate combinatorial solver to handle nonlinear reward functions, effectively translating basearm pessimism to action-level pessimism. This design prevents the selection of actions with high-fluctuation base arms, ensuring robust decision-making.

**Theoretical Analysis:** For the second question, we introduce two novel data coverage conditions: (1) the infinitynorm and (2) 1-norm triggering probability modulated (TPM) data coverage conditions, which characterize the dataset quality. These conditions quantify the amount of data required to accurately estimate each action by decomposing the data needs of each base arm and reweighting them based on their importance. Under these conditions, we prove that CLCB achieves a near-optimal suboptimality gap upper bound of $\tilde{O}(K^*\sqrt{C_\infty^*/n})$, where $K^*$ is the size of the optimal action, $C_\infty^*$ is the data coverage coefficient, and $n$ is the number of samples in the offline dataset. This result matches the lower bound $\Omega(K^*\sqrt{C_\infty^*/n})$ derived in this work up to a logarithmic factor. Our analysis care-

fully addresses key challenges, including handling nonlinear reward functions, confining uncertainties to base arms relevant to the optimal action, and accounting for arm triggering probabilities, enabling CLCB to achieve state-of-the-art performance with tighter bounds and relaxed assumptions for the real-world applications as discussed below.

**Practical Applications:** We show the practicality of Off-CMAB by fitting real-world problems into our framework and applying CLCB to solve them, including (1) learning to rank, (2) LLM caching, and (3) social influence maximization (IM). For the LLM cache problem, beyond directly fitting it into our framework, we improve existing results by addressing full-feedback arms, extending our approach to the online LLM setting with similar improvements. For social IM, our framework handles nuanced node-level feedback by constructing base-arm LCBs via intermediate UCB/LCBs, with additional refinements using variance-adaptive confidence intervals for improved performance.

**Empirical Validation:** Finally, extensive experiments on both synthetic and real-world datasets for learning to rank and LLM caching validate the superior performance of CLCB compared to baseline algorithms.

## 2. Problem Setting

In this section, we introduce our model for combinatorial multi-armed bandits with probabilistically triggering arms (CMAB-T) and the offline learning problem for CMAB-T.

### 2.1. Combinatorial Multi-armed Bandits with Probabilistically Triggered Arms

The original combinatorial multi-armed bandits problem with probabilistically triggered arms (CMAB-T) is an online learning game between a learner and the environment in $n$ rounds. We can specify a CMAB-T problem by a tuple $\mathcal{I} := ([m], \mathbb{D}, \mathcal{S}, \mathbb{D}_{\text{trig}}, R)$, where $[m]$ are base arms, $\mathcal{S}$ is the set of feasible combinatorial actions, $\mathbb{D}$ is the set of feasible distributions for the base arm outcomes, $\mathbb{D}_{\text{trig}}$ is the probabilistic triggering function, and $R$ is the reward function. The details of each component are described below:

**Base arms.** The environment has a set of $[m] = \{1, 2, ..., m\}$ base arms. Before the game starts, the environment chooses an *unknown* distribution $\mathbb{D}_{\text{arm}} \in \mathbb{D}$ over the bounded support $[0, 1]^m$. At each round $t \in [n]$, the environment draws random outcomes $\boldsymbol{X}_t = (X_{t,1}, ...X_{t,m}) \sim \mathbb{D}_{\text{arm}}$. Note that for a fixed arm $i$, we assume outcomes $X_{t,i}, X_{t',i}$ are independent across different rounds $t \neq t'$. However, outcomes for different arms $X_{t,i}$ and $X_{t,j}$ for $i \neq j$ can be dependent within the same round $t$. We use $\boldsymbol{\mu} = (\mu_1, ..., \mu_m)$ to denote the unknown mean vector, where $\mu_i := \mathbb{E}_{\boldsymbol{X}_t \sim \mathbb{D}_{\text{arm}}}[X_{t,i}]$ for each base arm $i$.

**Combinatorial actions.** At each round $t \in [n]$, the learner selects a combinatorial action $S_t \in \mathcal{S}$, where $\mathcal{S}$ is the set of feasible actions. Typically, $S_t$ is a set of individual base arms $S \subseteq [m]$, which we refer to as a super arm. However, $S_t$ can be more general than the super arm, e.g., continuous arms are useful for applying CMAB to resource allocation (Zuo & Joe-Wong, 2021), which we emphasize as needed.

**Probabilistic arm triggering feedback.** Motivated by the properties of real-world applications that will be introduced in detail in Section 4, we consider a feedback process that involves scenarios where each base arm in a super arm $S_t$ does not always reveal its outcome, even probabilistically. For example, a user might leave the system randomly at some point before examining the entire recommended list $S_t$, resulting in unobserved feedback for the unexamined items. To handle such probabilistic feedback, we assume that after the action $S_t$ is selected, the base arms in a random set $\tau_t \sim \mathbb{D}_{\text{trig}}(S_t, \boldsymbol{X}_t)$ are triggered depending on the outcome $\boldsymbol{X}_t$, where $\mathbb{D}_{\text{trig}}(S, \boldsymbol{X})$ is an unknown probabilistic distribution over the subsets $2^{[m]}$ given $S$ and $\boldsymbol{X}$. This means that the outcomes of the arms in $\tau_t$, i.e., $(X_{t,i})_{i \in \tau_t}$, are revealed as feedback to the learner, which could also be involved in determining the reward of action $S_t$ as we introduce later. To allow the algorithm to estimate the mean $\mu_i$ directly from samples, we assume the outcome does not depend on whether the arm $i$ is triggered, i.e., $\mathbb{E}_{\boldsymbol{X} \sim \mathbb{D}_{\text{arm}}, \tau \sim \mathbb{D}_{\text{trig}}(S, \boldsymbol{X})}[X_i | i \in \tau] = \mathbb{E}_{\boldsymbol{X} \sim \mathbb{D}_{\text{arm}}}[X_i]$. We use $p_i^{\mathbb{D}_{\text{arm}}, S}$ to denote the probability that base arm $i$ is triggered when the action is $S$ and the mean vector is $\boldsymbol{\mu}$.

**Reward function.** At the end of round $t \in [n]$, the learner receives a nonnegative reward $R_t = R(S_t, \boldsymbol{X}_t, \tau_t)$, determined by action $S_t$, outcome $\boldsymbol{X}_t$, and triggered arm set $\tau_t$. Similarly to (Wang & Chen, 2017), we assume the expected reward to be $r(S_t; \boldsymbol{\mu}_t) := \mathbb{E}[R(S_t, \boldsymbol{X}_t, \tau)]$, a function of the unknown mean vector $\boldsymbol{\mu}$, where the expectation is taken over the randomness of $\boldsymbol{X}_t$ and $\tau_t \sim \mathbb{D}_{\text{trig}}(S_t, \boldsymbol{X}_t)$.

**Reward conditions.** Owing to the nonlinearity of the reward and the combinatorial structure of the action, it is essential to give some conditions for the reward function to achieve any meaningful theoretical guarantee (Wang & Chen, 2017). We consider the following conditions:

**Condition 1** (Monotonicity, Wang & Chen (2017)). *We say that a CMAB-T problem satisfies the monotonicity condition, if for any action $S \in \mathcal{S}$, for any two distributions $\mathbb{D}_{arm}, \mathbb{D}'_{arm} \in \mathbb{D}$ with mean vectors $\boldsymbol{\mu}, \boldsymbol{\mu}' \in [0, 1]^m$ such that $\mu_i \leq \mu'_i$ for all $i \in [m]$, we have $r(S; \boldsymbol{\mu}) \leq r(S; \boldsymbol{\mu}')$.*

**Condition 2** (1-norm TPM Bounded Smoothness, Wang & Chen (2017)). *We say that a CMAB-T problem satisfies the 1-norm triggering probability modulated (TPM) bounded smoothness condition with coefficient $B_1$, if there exists coefficient $B_1 > 0$ (referred to as smoothness coefficient), if for any two distributions $\mathbb{D}_{arm}, \mathbb{D}'_{arm} \in \mathbb{D}$ with mean*

*vectors $\boldsymbol{\mu}, \boldsymbol{\mu}' \in [0, 1]^m$, and for any action $S \in \mathcal{S}$, we have $|r(S; \boldsymbol{\mu}') - r(S; \boldsymbol{\mu})| \leq B_1 \sum_{i \in [m]} p_i^{\mathbb{D}_{arm}, S} |\mu_i - \mu'_i|$.*

**Remark 1** (Intuitions of Condition 1 and Condition 2)**.** Condition 1 indicates the reward is monotonically increasing when the parameter $\boldsymbol{\mu}$ increases. Condition 2 bounds the reward smoothness/sensitivity, i.e., the amount of the reward change caused by the parameter change from $\boldsymbol{\mu}$ to $\boldsymbol{\mu}'$. In the learning to rank (Section 4.1), for example, these conditions upper bounds the difference in total number of purchases when the purchase probability for the items changes from $\boldsymbol{\mu}$ to $\boldsymbol{\mu}'$. For Condition 2, the key feature is that the parameter change in each base arm $i$ is modulated by the triggering probability $p_i^{\boldsymbol{\mu}, S}$, saving a $p_{\min}$ factor in (Chen et al., 2016) where $p_{\min}$ is the minimum positive triggering probability. Intuitively, for base arm $i$ that is unlikely to be triggered/observed (small $p_i^{\boldsymbol{\mu}, S}$), Condition 2 ensures that a large change in $\mu_i$ (due to insufficient observation) only causes a small change (multiplied by $p_i^{\boldsymbol{\mu}, S}$) in reward, saving a $p_{\min}$ factor in (Wang & Chen, 2017) where $p_{\min}$ is the minimum positive triggering probability. In learning to rank application, for example, since users will never purchase an item if it is not examined, increasing or decreasing the purchase probability of an item that is unlikely to be examined (i.e., with small $p_i^{\boldsymbol{\mu}, S}$) does not significantly affect the total number of purchases.

## 2.2. Offline Data Collection and Performance Metric

**Offline dataset.** Fix any CMAB-T problem $\mathcal{I}$ together with its underlying distribution $\mathbb{D}_{\text{arm}}$. We consider the offline learning setting, that is, the learner only has access to a dataset $\mathcal{D}$ consisting of $n$ feedback data $\mathcal{D} := \{(S_t, \tau_t, (X_{t,i})_{i \in \tau_t})\}_{t=1}^n$ collected a priori by an experimenter. Here, we assume the experimenter takes an *unknown* data collecting distribution $\mathbb{D}_{\mathcal{S}}$ over feasible actions $\mathcal{S}$, such that $S_t$ is generated i.i.d. from $S_t \sim \mathbb{D}_{\mathcal{S}}$ for any offline data $t \in [n]$. After $S_t$ is sampled, the environment generates outcome $\boldsymbol{X}_t \sim \mathbb{D}_{\text{arm}}$. Then $\tau_t \sim \mathbb{D}_{\text{trig}}(S_t, \boldsymbol{X}_t)$ are triggered, whose outcome are recorded as $(X_{t,i})_{i \in \tau_t}$. To this end, we use $p_i^{\mathbb{D}_{\text{arm}}, \mathbb{D}_{\mathcal{S}}}$ to denote the data triggering probability, i.e., $p_i^{\mathbb{D}_{\text{arm}}, \mathbb{D}_{\mathcal{S}}} = \mathbb{E}_{S \sim \mathbb{D}_{\mathcal{S}}, \boldsymbol{X} \sim \mathbb{D}_{\text{arm}}, \tau \sim \mathbb{D}_{\text{trig}}(S, \boldsymbol{X})}[\mathbb{I}\{i \in \tau\}]$, which indicates the frequency of observing arm $i \in [m]$.

**Approximation oracle and $\alpha$ approximate suboptimality gap.** The goal of the offline learning problem for CMAB-T is to identify the optimal combinatorial action that maximizes the expected reward. Correspondingly, the performance of an offline learning algorithm $A$ is measured by the *suboptimality-gap*, defined as the difference in the expected reward between the optimal action $S^* := \operatorname{argmax}_{S' \in \mathcal{S}} r(S'; \boldsymbol{\mu})$ and the action $\hat{S}$ chosen by algorithm $A$ with dataset $\mathcal{D}$ as input. For many reward functions, it is NP-hard to compute the exact $S^*$ even when $\boldsymbol{\mu}$ is known, so similar to (Chen et al., 2013; Wang & Chen,

2017; Liu et al., 2022; 2024a), we assume that algorithm $A$ has access to an offline $\alpha$-approximation ORACLE, which takes any mean vector $\boldsymbol{\mu} \in [0,1]^m$ as input, and outputs an $\alpha$-approximate solution $S \in \mathcal{S}$, i.e., $S = \text{ORACLE}(\boldsymbol{\mu})$ satisfies

$$r(S; \boldsymbol{\mu}) \geq \alpha \cdot \max_{S' \in \mathcal{S}} r(S'; \boldsymbol{\mu}) \tag{1}$$

Given any action $\hat{S} \in \mathcal{S}$, the $\alpha$-approximate suboptimality gap over the CMAB-T instance $\mathcal{I}$ with unknown base arm mean $\boldsymbol{\mu}$ is defined as

$$\text{SubOpt}(\hat{S}; \alpha, \mathcal{I}) := \alpha \cdot r(S^*; \boldsymbol{\mu}) - r(\hat{S}; \boldsymbol{\mu}), \tag{2}$$

Our objective is to design an algorithm $A$ such that $\text{SubOpt}(\hat{S}; \alpha, \mathcal{I})$ is minimized with high probability $1 - \delta$, where the randomness is taken over the $(\mathbb{D}_{\mathcal{S}}, \mathbb{D}_{\text{arm}}, \mathbb{D}_{\text{trig}})$.

### 2.3. Data Coverage Conditions: Quality of the Dataset

Since the offline learning performance is closely related to the quality of the dataset $\mathcal{D}$, we consider the following conditions about the offline dataset:

**Condition 3** (Infinity-norm TPM Data Coverage). *For a CMAB-T instance $\mathcal{I}$ with unknown distribution $\mathbb{D}_{arm}$ and mean vector $\boldsymbol{\mu}$, let $S^* = \arg\max_{S \in \mathcal{S}} r(S; \boldsymbol{\mu})$.[2] We say that the data collecting distribution $\mathbb{D}_{\mathcal{S}}$ satisfies the infinity-norm triggering probability modulated (TPM) data coverage condition, if there exists a coefficient $C_\infty^* > 0$ (referred to as coverage coefficient), such that*

$$\max_{i \in [m]} \frac{p_i^{\mathbb{D}_{arm}, S^*}}{p_i^{\mathbb{D}_{arm}, \mathbb{D}_{\mathcal{S}}}} \leq C_\infty^*. \tag{3}$$

**Condition 4** (1-norm TPM Data Coverage). *For a CMAB-T instance $\mathcal{I}$ with unknown distribution $\mathbb{D}_{arm}$ and mean vector $\boldsymbol{\mu}$, let $S^* = \arg\max_{S \in \mathcal{S}} r(S; \boldsymbol{\mu})$. We say that the data collecting distribution $\mathbb{D}_{\mathcal{S}}$ satisfies the 1-norm triggering probability modulated (TPM) data coverage condition, if there exists a coefficient $C_1^* > 0$, such that*

$$\sum_{i \in [m]} \frac{p_i^{\mathbb{D}_{arm}, S^*}}{p_i^{\mathbb{D}_{arm}, \mathbb{D}_{\mathcal{S}}}} \leq C_1^*. \tag{4}$$

**Remark 2** (Intuition of Condition 3 and Condition 4). Both Condition 3 and Condition 4 evaluate the quality of the dataset $\mathcal{D}$, which directly impacts the amount of data required to accurately estimate the expected reward of the optimal $S^*$. The denominator $p_i^{\mathbb{D}_{arm}, \mathbb{D}_{\mathcal{S}}}$ represents the data generation rate for arm $i$, and $\frac{1}{p_i^{\mathbb{D}_{arm}, \mathbb{D}_{\mathcal{S}}}}$ corresponds to the expected number of samples needed to observe one instance

---

[2]Note that for simplicity, we choose an arbitrary optimal solution $S^*$, and in practice, we can choose one that leads to the smallest coverage coefficient.

of arm $i$. Incorporating similar triggering probability modulation as in Condition 2, we use $p_i^{\mathbb{D}_{\text{arm}}, S^*}$ to reweight the importance of each arm $i$, and when $p_i^{\mathbb{D}_{\text{arm}}, S^*}$ is small, the uncertainty associated with arm $i$ has small impact on the estimation. Consequently, a large amount of data is not required for learning about arm $i$. Notably, because we compare against the optimal super arm $S^*$, we only require the weight $p_i^{\mathbb{D}_{\text{arm}}, S^*}$ of the optimal action $S^*$ as the modulation. This is less restrictive than uniform coverage conditions that require adequate data for all possible actions, as used by Chen & Jiang (2019b); Jiang (2019).

The primary difference between Condition 3 and Condition 4 lies in the computation of the total expected data requirements for all arms. Condition 3 adopts a worst-case perspective using the max operator, whereas Condition 4 considers the total summation over $i \in [m]$. Generally, the relationship $C_1^* \leq K^* C_\infty^*$ holds. Depending on the application, different conditions may be preferable, offering varying guarantees for the suboptimality gap. Detailed discussion is provided in Remark 4.

**Remark 3** (Extension to handle out-of-distribution $\mathbb{D}_{\mathcal{S}}$). Note that Condition 3 and Condition 4 are restrictions on the base arm level. Hence, our framework is flexible and can accommodate any data collection distribution $\mathbb{D}_{\mathcal{S}}$, including distributions over actions $\mathcal{S}'$ that may assign zero probability to the optimal action $S^*$ or even extend beyond the feasible action set $\mathcal{S}$. For example, in the LLM cache problem (Section 4.2), the experimenter might ensure arm feedback by using an empty cache in each round, leveraging cache misses to collect feedback. In this case, the distribution $\mathbb{D}_{\mathcal{S}}$ assigns zero probability to the optimal cache configuration as well as any reasonable cache configurations.

## 3. CLCB Algorithm and Theoretical Analysis

In this section, we first introduce the Combinatorial Lower Confidence Bound (CLCB) algorithm (Algorithm 1) and analyze its performance in Section 3. We then derive a lower bound on the suboptimality gap, and we show that our gap upper bound matches this lower bound up to logarithmic factors.

The CLCB algorithm first computes high-probability lower confidence bounds (LCBs) for each base arm (line 5). These LCB estimates are then used as inputs to a combinatorial oracle to select an action $\hat{S}$ that approximately maximizes the worst-case reward function $r(S^*; \boldsymbol{\mu})$ (line 7). The key part of Algorithm 1 is to conservatively use the LCB, penalizing each base arm by its confidence interval, $\sqrt{\log(\frac{4mn}{\delta})/2N_i}$. This approach, rooted in the pessimism principle (Jin et al., 2020a), mitigates the impact of high fluctuations in empirical estimates caused by limited observations, effectively addressing the uncertainty inherent in

**Algorithm 1** CLCB: Combinatorial Lower Confidence Bound Algorithm for Off-CMAB

---
1: **Input:** Dataset $\mathcal{D} = \{(S_t, \tau_t, (X_{t,i})_{i \in \tau_t})\}_{t=1}^n$, computation oracle ORACLE, probability $\delta$.
2: **for** arm $i \in [m]$ **do**
3:     Calculate counter $N_i = \sum_{t=1}^n \mathbb{I}\{i \in \tau_t\}$.
4:     Calculate empirical mean $\hat{\mu}_i = \frac{\sum_{t=1}^n \mathbb{I}\{i \in \tau_t\} X_{t,i}}{N_i}$.
5:     Calculate LCB $\underline{\mu}_i = \hat{\mu}_i - \sqrt{\frac{\log(\frac{4mn}{\delta})}{2N_i}}$.
6: **end for**
7: Call oracle $\hat{S} = \text{ORACLE}(\underline{\mu}_1, ..., \underline{\mu}_m)$.
8: **Return:** $\hat{S}$.
---

passively collected data.

**Theorem 1.** *Let $\mathcal{I}$ be a CMAB-T problem and $\mathcal{D}$ a dataset with $n$ data samples. Let $\hat{S}$ denote the action given by CLCB (Algorithm 1) using an $\alpha$-approximate oracle. If the problem $\mathcal{I}$ satisfies (a) monotonicity (Condition 1), (b) 1-norm TPM smoothness (Condition 2) with coefficient $B_1$, and (c) the infinity-norm TPM data coverage condition (Condition 3) with coefficient $C_\infty^*$; and the number of samples satisfies $n \geq \frac{8 \log(\frac{m}{\delta})}{\min_{i \in [m]: p_i^{\mathbb{D}_{arm}, S^*} > 0} p_i^{\mathbb{D}_{arm}, \mathbb{D}_S}}$, then, with probability at least $1 - \delta$ (the randomness is taken over the all distributions $\mathbb{D}_S, \mathbb{D}_{arm}, \mathbb{D}_{trig}$), the suboptimality gap satisfies:*

$$\text{SubOpt}(\hat{S}; \alpha, \mathcal{I}) \leq 2\alpha B_1 \bar{K}_2^* \sqrt{\frac{C_\infty^* \log(4mn/\delta)}{n}}, \quad (5)$$

*where $\bar{K}_2^* := \sum_{i \in [m]} \sqrt{p_i^{\mathbb{D}_{arm}, S^*}}$ is the $\ell_2$-action size of $S^*$. Further, if problem $\mathcal{I}$ satisfies the 1-norm TPM data coverage condition (Condition 4) with coefficient $C_1^*$, then, with probability at least $1 - \delta$, the suboptimality gap satisfies:*

$$\text{SubOpt}(\hat{S}; \alpha, \mathcal{I}) \leq 2\alpha B_1 \sqrt{\frac{\bar{K}^* C_1^* \log(4mn/\delta)}{n}}, \quad (6)$$

*where $\bar{K}^* := \sum_{i \in [m]} p_i^{\mathbb{D}_{arm}, S^*}$ is the action size of $S^*$.*

*Proof Idea.* The proof of Theorem 1 consists of three key steps: (1) express the suboptimality gap in terms of the uncertainty gap $r(S^*; \boldsymbol{\mu}) - r(S^*; \underline{\boldsymbol{\mu}})$ over the optimal action $S^*$, rather than the on-policy error over the chosen action $\hat{S}$ as in online CMAB, (2) leverage Condition 2 to relate the uncertainty gap to the per-arm estimation gap, and (3) utilize Condition 2 to deal with the arbitrary data collection probabilities and bound the per-arm estimation gap in terms of $n$. For a detailed proof, see Appendix D. ∎

**Remark 4** (Discussion of Theorem 1)**.** Looking at the suboptimality gap result, both Eq. (5) and Eq. (6) decrease at a rate of $\frac{1}{\sqrt{n}}$ with respect to the number of offline data samples $n$. Additionally, they scale linearly with the smoothness

coefficient $B_1$ and the approximation ratio $\alpha$. For problems satisfying Eq. (5), the gap scales linearly with the $\ell_2$-action size $\bar{K}_2^*$ and the the coverage coefficient $C_\infty^*$ in Eq. (5). For problems satisfying Eq. (6), the gap depends on the action size $\bar{K}^*$ and the 1-norm data coverage coefficient $C_1^*$. To output an action that is $\epsilon$-close to $S^*$, Eq. (5) and Eq. (6) need $\tilde{O}(B_1^2 \alpha^2 \bar{K}_2^{*2} C_\infty^*/\epsilon^2)$ and $\tilde{O}(B_1^2 \alpha^2 \bar{K}^* C_1^*/\epsilon^2)$ samples, respectively.

In general, we have $C_1^* \leq K^* C_\infty^*$ and $\bar{K}^* \geq \frac{(\bar{K}_2^*)^2}{K^*}$, indicating that neither Eq. (5) nor Eq. (6) strictly dominates the other. For instance, for CMAB with semi-bandit feedback where $p_i^{\mathbb{D}_{arm}, S^*} = p_j^{\mathbb{D}_{arm}, S^*} = 1$ for any $i, j \in S^*$ and 0 otherwise, Eq. (6) is tighter than Eq. (5) since $\bar{K}^* = \bar{K}_2^* = K^*$ and $C_1^* \leq K^* C_\infty^*$. Conversely, for the LLM cache to be introduced in Section 4.2, if the experimenter selects the empty cache each time, such that $\frac{p_i^{\mathbb{D}_{arm}, S^*}}{p_i^{\mathbb{D}_{arm}, \mathbb{D}_s}} = 1$ for $i \in S^*$, then we have $C_1^* = K^* C_\infty^*$. Since $\bar{K}^* \geq \frac{(\bar{K}_2^*)^2}{K^*}$ so Eq. (5) is tighter than Eq. (6).

**Lower bound result.** In this section, we establish the lower bound for a specific combinatorial multi-armed bandit (CMAB) problem: the stochastic $k$-path problem $\mathcal{I}$. This problem was first introduced in (Kveton et al., 2015c) to derive lower bounds for the online CMAB problem.

The $k$-path problem involves $m$ arms, representing path segments denoted as $[m] = 1, 2, \ldots, m$. Without loss of generality, we assume $m/k$ is an integer. The feasible combinatorial actions $\mathcal{S}$ consist of $m/k$ paths, each containing $k$ unique arms. Specifically, the $j$-th path for $j \in [m/k]$ includes the arms $(j-1)k+1, \ldots, jk$. We define k-path$(m, k, C_\infty^*)$ as the set of all possible outcome and data collection distribution pairs $(\mathbb{D}_{arm}, \mathbb{D}_S)$ satisfying the following conditions:

(1) The outcome distribution $\mathbb{D}_{arm}$ specifies that all arms in any path $j \in [m/k]$ are fully dependent Bernoulli random variables, i.e., $X_{t,(j-1)k+1} = X_{t,(j-1)k+2} = \cdots = X_{t,jk}$, all with the same expectation $\mu_j$.

(2) The pair $(\mathbb{D}_{arm}, \mathbb{D}_S)$ satisfies the infinity-norm TPM data coverage condition (Condition 3) with $C_\infty^*$, i.e., $\max_{i \in [m]} \frac{p_i^{\mathbb{D}_{arm}, S^*}}{p_i^{\mathbb{D}_{arm}, \mathbb{D}_S}} \leq C_\infty^*$.

The feedback of the $k$-path problem follows the classical semi-bandit feedback for any $S \in \mathcal{S}$, i.e., $p_i^{\mathbb{D}_{arm}, S} = 1$ if $i \in S$ and $p_i^{\mathbb{D}_{arm}, S} = 0$ otherwise. We use $\mathcal{D} = (S_t, (X_{t,i})_{i \in S_t})_{t=1}^n$ to denote a random offline $k$-path dataset of size $n$ and $\mathcal{D} \sim \mathbb{D}(\mathbb{D}_{arm}, \mathbb{D}_S)$ to indicate dataset $\mathcal{D}$ is generated under the data collecting distribution $\mathbb{D}_S$ with the underlying arm distribution $\mathbb{D}_{arm}$.

**Theorem 2.** *Let us denote $A(\mathcal{D}) \in \mathcal{S}$ as the action returned by any algorithm $A$ that takes a dataset $\mathcal{D}$ of $n$*

*samples as input. For any* $m, k \in \mathbb{Z}_+$, *such that* $m/k$ *is an integer, and any* $C_\infty^* \geq 2$, *the following lower bound holds:*

$$\inf_A \ \sup_{(\mathbb{D}_{arm}, \mathbb{D}_S) \in k\text{-}path(m,k,C_\infty^*)} \mathbb{E}_{\mathcal{D} \sim \mathbb{D}(\mathbb{D}_{arm}, \mathbb{D}_S)}[r(S^*; \boldsymbol{\mu}) - r(A(\mathcal{D}); \boldsymbol{\mu})] \geq k \min\left(1, \sqrt{\frac{C_\infty^*}{n}}\right).$$

Comparing this result to the upper bound established in Theorem 1 for the $k$-path problem, we can verify that this problem satisfies Condition 2 with $B_1 = 1$ and $\bar{K}_2^* = k$, meaning that our upper bound result matches the lower bound up to logarithmic factors.

# 4. Applications of the Off-CMAB Framework

In this section, we introduce three representative applications that can fit into our Off-CMAB-T framework with new/improved results, which are summarized in Table 1. We also provide empirical evaluations for the cascading bandit and the LLM cache in Section 5.

## 4.1. Offline Learning for Cascading Bandits

The cascading bandit problem (Kveton et al., 2015a; Li et al., 2016; Vial et al., 2022; Dai et al., 2025a) addresses the *online* learning to rank problem (Liu et al., 2009) under the cascade model (Craswell et al., 2008). The canonical cascading bandit problem considers a $T$-round sequential decision-making process. At each round $t \in [T]$, a user $t$ comes to the recommendation system (e.g., Amazon), and the learner aims to recommend a ranked list $S_t = (a_{t,1}, ..., a_{t,k}) \subseteq [m]$ of length $k$ (i.e., a super arm) from a total of $m$ candidate products (i.e., base arms). Each item $i \in S_t$ has an unknown probability $\mu_i$ of being satisfactory and purchased by user $t$, which without loss of generality, is assumed to be in descending order $\mu_1 \geq \mu_2 \geq ... \geq \mu_m$.

**Reward function and cascading feedback.** Given the ranked list $S_t$, the user examines the list from $a_{t,1}$ to $a_{t,k}$ until they purchase the first satisfactory item (and leave the system) or exhaust the list without finding a satisfactory item. If the user purchases an item (suppose the $j_t$-th item), the learner receives a reward of 1 and observes outcomes of the form $(X_{t,a_1}, ..., X_{t,a_{j_{t-1}}}, X_{t,a_{j_t}}, ..., X_{t,a_k}) = (0, ..., 0, 1, -, ..., -)$, meaning the first $j_t - 1$ items are unsatisfactory (denoted as 0), the $j_t$-th item is satisfactory (denoted as 1), and the outcomes of the remaining items are unobserved (denoted as $-$). Otherwise, the learner receives a reward of 0 and observes Bernoulli outcomes $(X_{t,a_1}, ..., X_{t,a_k}) = (0, 0, ..., 0)$. The expected reward is $r(S_t; \boldsymbol{\mu}) = \mathbb{E}[\{\exists i \in [k] : X_{t,a_i} = 1\}] = 1 - \prod_{i \in S_t}(1 - \mu_i)$. Since $\mu_1 \geq \mu_2... \geq \mu_m$, we know that the optimal ranked list is the top-$k$ items $S^* = (1, 2, ..., k)$. The goal of the cascading bandit problem is to maximize the expected number of user purchases by applying an online learning algorithm. For this setting, we can see that it follows the cascading

feedback and the triggered arms are $\tau_t = \{a_{t,1}, ..., a_{t,j_t}\}$ where $j_t = K$ if $(a_{t,1}, ... a_{t,k}) = (0, .., 0)$ or otherwise $j_t = \operatorname{argmin}\{i \in [k] : X_{t,a_{t,i}} = 1\}$.

**Learning from the offline dataset.** We consider the offline learning setting for cascading bandits, where we are given a pre-collected dataset $\mathcal{D} = (S_t, \tau_t, (X_{t,i})_{i \in \tau_t})_{t=1}^n$ consisting of $n$ ranked lists and the user feedback for these ranked lists, where each $S_t$ is sampled from the data collecting distribution $\mathbb{D}_S$. Let us use $q_{ij}$ to denote the probability that arm $i$ is sampled at the $j$-th position of the ranked list, for $i \in [m], j \in [k]$. Then we have $p_i^{\mathbb{D}_{arm}, \mathbb{D}_S} \geq \sum_{j=1}^k q_{ij}(1 - \mu_1)^{j-1}$ and $p_i^{\mathbb{D}_{arm}, S^*} = \prod_{j=1}^{i-1}(1 - \mu_j)$. Therefore, we can derive that the 1-norm data coverage coefficient in Condition 4 is $C_1^* = \sum_{i=1}^k \frac{\prod_{j=1}^{i-1}(1-\mu_j)}{\sum_{j=1}^k q_{ij}(1-\mu_1)^{j-1}}$.

**Algorithm and result.** This application fits into the CMAB-T framework, satisfying Condition 2 with coefficient $B_1 = 1$ as in (Wang & Chen, 2017). The oracle is essentially to find the top-$k$ items regarding LCB $\underline{\mu}_i$, which maximizes $r(S; \underline{\boldsymbol{\mu}})$ in $O(m \log k)$ time complexity using the max-heap. Plugging this oracle into line 7 of Algorithm 1 gives the algorithm, whose detail is in Algorithm 4 in Appendix F.

**Corollary 1.** *For cascading bandits with arms* $\mu_1 \geq \mu_2... \geq \mu_m$ *and a dataset* $\mathcal{D}$ *with* $n$ *data points, suppose* $n \geq \frac{8 \log(\frac{2mn}{\delta})}{\min_{i \in [k]} \sum_{j=1}^k q_{ij}(1-\mu_1)^{j-1}}$, *where* $q_{ij}$ *is the probability that item* $i$ *is sampled at the* $j$-th *position regarding* $\mathbb{D}_S$. *Letting* $\hat{S}$ *be the ranked list returned by Algorithm 4, then with probability at least* $1 - \delta$,

$$r(S^*; \boldsymbol{\mu}) - r\left(\hat{S}; \boldsymbol{\mu}\right)$$
$$\leq 2\sqrt{\frac{k \log(\frac{4mn}{\delta})}{n} \sum_{i=1}^k \frac{\prod_{j=1}^{i-1}(1 - \mu_j)}{\sum_{j=1}^k q_{ij}(1-\mu_1)^{j-1}}}, \quad (7)$$

*If* $\mathbb{D}_S$ *is a uniform distribution so that* $q_{ij} = \frac{1}{m}$, *it holds that* $C_1^* \leq \frac{\mu_1 \cdot m}{\mu_k}$ *and*

$$r(S^*; \boldsymbol{\mu}) - r\left(\hat{S}; \boldsymbol{\mu}\right) \leq 2\sqrt{\frac{k \log(\frac{4mn}{\delta})}{n} \cdot \frac{m\mu_1}{\mu_k}}. \quad (8)$$

## 4.2. Offline Learning for LLM Cache

The LLM cache is a system designed to store and retrieve outputs of Large Language Models (LLMs), aiming to enhance efficiency and reduce redundant computations during inference (Pope et al., 2022; Bang, 2023; Zhu et al., 2023; Dai et al., 2025b).

In the LLM cache bandit (Zhu et al., 2023), which is a $T$-round sequential learning problem, we consider a finite set of $m$ distinct queries $\mathcal{Q} = \{q_1, ..., q_m\}$. Each query $q \in \mathcal{Q}$ is associated with an unknown expected cost $c(q) \in [0, 1]$

*Table 1.* Summary of the results of applying the Off-CMAB framework to various applications.

| Application | Smoothness | Data Coverage | Suboptimality Gap | Improvements |
|---|---|---|---|---|
| Learning to Rank (Section 4.1) | $B_1 = 1$ | $C_1^* = \frac{\mu_1 \cdot m}{\mu_k}$ | $\tilde{O}\left(\sqrt{\frac{k}{n} \cdot \frac{m\mu_1}{\mu_k}}\right)^*$ | – |
| LLM Cache (Section 4.2) | $B_1 = 1$ | $C_1^* = m$ | $\tilde{O}\left(\sqrt{\frac{m}{n}}\right)$ | $\tilde{O}\left(\sqrt{\frac{k^2}{C_1}}\right)^\dagger$ |
| Social Influence Maximization (Section 4.3) | $B_1 = V$ | Assumption 1** | $\tilde{O}\left(\sqrt{\frac{V^2 d_{\max}^2 \sigma^2(S^*;G)}{\eta \cdot \gamma^3 \cdot n}}\right)$ | $\tilde{O}\left(\sqrt{\frac{V^4}{k^2 d_{\max}^2 \eta}}\right)^\ddagger$ |

* $m, k, \mu_1, \mu_k$ denote the number of items, the length of the ranked list, and click probability for 1-st and $k$-th items, respectively;
† $m, k, C_1$ denote the number of LLM queries, the size of cache, and the lower bound of the query cost, respectively;
** Similar to Chen et al. (2021), we depend directly on assumption for seed sampling probability bound $\gamma$ and the activation probability bound $\eta$;
‡ $V, d_{\max}, \sigma(S^*; G), k$ denote the number of nodes, the max out-degree, optimal influence spread, and the number of seed nodes, respectively.

and unknown probability $p(q) \in [0,1]$, for a total of $2m$ base arms. When query $q$ is input to the LLM system, the LLM processes it and returns a corresponding response (i.e., answer) $r(q)$. Every round $t$ when the LLM processes $q$, it will incur a random cost $C_t(q)$ with mean $c(q)$, representing floating point operations (FLOPs) or the price for API calls (Zhu et al., 2023). We assume that $C_t(q) = c(q) + \epsilon_t(q)$, where $\epsilon_t(q)$ is a sub-Gaussian noise that captures the uncertainties in the cost, with $\mathbb{E}[\epsilon_t(q)] = 0$.

The goal of LLM cache bandit is to find the optimal cache $\mathcal{M}^*$ storing the query-response pairs that are both likely to be reused and associated with high costs.

**Expected cost function and cache feedback.** In each round $t$, a user comes to the system with query $q_t$, which is sampled from $\mathcal{Q}$ according to a fixed unknown distribution $\{p(q)\}_{q \in \mathcal{Q}}$ with $\sum_{q \in \mathcal{Q}} p(q) = 1$. To save the cost of repeatedly processing the queries, the LLM system maintains a cache $\mathcal{M}_t \subseteq \mathcal{Q}$, storing a small subset of queries of size $k \geq 0$ with their corresponding results. After $q_t$ is sampled, the agent will first check the current cache $\mathcal{M}_t$. If the query $q_t$ is found in the cache, i.e., $q_t \in \mathcal{M}_t$, we say the query *hits* the cache. In this case, the result of $q_t$ is directly returned without further processing by the LLM. The cost of processing this query is 0 and will save a potential cost $C_t(q_t)$, which is *unobserved* to the agent. If query $q_t$ does not hit the cache, the system processes the query, incurring a cost $C_t(q_t)$ which is observed by the learner, and returns the result $r(q_t)$. Let us denote $\boldsymbol{c} = (c(q))_{q \in \mathcal{Q}}$ and $\boldsymbol{p} = (p(q))_{q \in \mathcal{Q}}$ for convenience. Given any cache $\mathcal{M}$ and the query $q$, the random cost saved in round $t$ is $C_t(\mathcal{M}, q) = \mathbb{I}\{q \notin \mathcal{M}\} C_t(q)$. Thus the expected cost incurred is

$$c(\mathcal{M}; \boldsymbol{c}, \boldsymbol{p}) = \mathbb{E}\left[\sum_{q \in \mathcal{Q}} C_t(\mathcal{M}, q)\right] = \sum_{q \notin \mathcal{M}} p(q)c(q). \quad (9)$$

From our CMAB-T point of view, selecting any cache $\mathcal{M}$ of size $k$ can be regarded as selecting the super arm $S = \mathcal{Q} - \mathcal{M}$ with size $m - k$, which is the complement of $\mathcal{M}$. Thus, our super arms represent the queries not entered into

the cache. In this context, the expected cost function can be rewritten as: $c(S; \boldsymbol{c}, \boldsymbol{p}) = \sum_{q \in S} p(q)c(q)$. The goal of LLM cache bandit is to find the optimal cache $\mathcal{M}^*$ storing the query-response pairs that are both likely to be reused and associated with high costs, i.e., to find out the optimal super arm $S^* = \arg\min_{S \subseteq \mathcal{Q}: |S| \geq m-k} c(S; \boldsymbol{c}, \boldsymbol{p})$.

We separately consider the cache feedback for $\boldsymbol{c}$ and $\boldsymbol{p}$. For unknown costs $\boldsymbol{c}$, we can see that $\tau_{t,c} = q_t$ if $q_t \in \mathcal{M}_t$ and $\tau_{t,c} = \emptyset$ otherwise. For unknown probability distribution $\boldsymbol{p}$, we observe full feedback $\tau_{t,p} = \mathcal{Q}$ since $q_t$ means $q_t$ arrives and all other queries do not arrive. For the triggering probability, we have that, for any $S \in \mathcal{S}$, the triggering probability for unknown costs $p_{q,c}^{\mathbb{D}_{\text{arm}}, S} = p(q)$ for $q \in S$ and 0 otherwise. The triggering probability for unknown arrival probability $p_{q,p}^{\mathbb{D}_{\text{arm}}, S} = 1$ for all $q \in \mathcal{Q}$.

**Learning from the offline dataset.** We consider the offline learning setting for the LLM cache, where we are given a pre-collected dataset $\mathcal{D} = (\mathcal{M}_t, q_t, C_t)_{t=1}^n$ consisting of the selected cache $\mathcal{M}_t$, the arrived query $q_t$, and their cost feedback $C_t = C_t(q_t)$ if $q_t \notin \mathcal{M}_t$ or $C_t = \emptyset$ is unobserved otherwise, where each $\mathcal{M}_t$ is sampled from the data collecting distribution $\mathbb{D}_{\mathcal{S}}$. Let $\nu(q) = \Pr_{\mathcal{M} \sim \mathbb{D}_{\mathcal{S}}}[q \notin \mathcal{M}]$ be the probability that $q$ is not sampled in the experimenter's cache. Then we can derive that the 1-norm data coverage coefficient in Condition 4 is $C_1^* = \sum_{q \in S^*} \frac{1}{\nu(q)} + m$.

**Algorithm and result.** This application fits into the CMAB-T framework, satisfying the 1-norm TPM smoothness condition with coefficient $B_1 = 1$ (see Appendix G.1 for the detailed proof). Since we are minimizing the cost rather than maximizing the reward, we use the UCB $\bar{p}(q)\bar{c}(q)$. The oracle is essentially to find the top-$k$ queries regarding UCB $\bar{p}(q)\bar{c}(q)$, which minimizes $c(S; \bar{\boldsymbol{c}}, \bar{\boldsymbol{p}})$ in $O(m \log k)$ time complexity using the max-heap. The detailed algorithm and its result is provided in Algorithm 5.

Since the arrival probabilities $\boldsymbol{p}$ are full-feedback categorical random variables, we can further improve our result by a factor of $\sqrt{m}$ by directly using the empirical mean of $\hat{\boldsymbol{p}}$ instead of UCB $\bar{\boldsymbol{p}}$. The improved algorithm is shown in Algorithm 2 with its theoretical suboptimality guarantee:

**Algorithm 2** `CLCB-LLM-C`: Combinatorial Lower Confidence Bound Algorithm for LLM Cache

---

1: **Input:** Dataset $\mathcal{D} = \{(\mathcal{M}_t, q_t, C_t)\}_{t=1}^n$, queries $\mathcal{Q}$, solver `Top-k`, probability $\delta$.
2: **for** query $q \in \mathcal{Q}$ **do**
3:      Calculate counters $N(q) = \sum_{t=1}^n \mathbb{I}\{q = q_t\}$ and $N_c(q) = \sum_{t=1}^n \mathbb{I}\{q = q_t \text{ and } q_t \notin \mathcal{M}_t\}$.
4:      Calculate empirical means $\hat{p}(q) = N(q)/n$ and $\hat{c}(q) = \sum_{t\in[n]} \mathbb{I}\{q = q_t \text{ and } q_t \notin \mathcal{M}_t\}C_t/N_c(q)$.
5:      Calculate UCB $\bar{c}(q) = \hat{c}(q) + \sqrt{\frac{\log(\frac{6mn}{\delta})}{2N_c(q)}}$.
6: **end for**
7: Call $\hat{\mathcal{M}} = \texttt{Top-k}\left(\hat{p}(q_1)\bar{c}(q_1), ..., \hat{p}(q_m)\bar{c}(q_m)\right)$.
8: **Return:** $\hat{\mathcal{M}}$.

---

**Theorem 3.** *For LLM cache bandit with a dataset $\mathcal{D}$ of $n$ data samples, suppose $n \geq \frac{8\log(\frac{1}{\delta})}{\min_{q\in\mathcal{Q}-\mathcal{M}^*} p(q)\nu(q)}$, where $\nu(q)$ is the probability that query $q$ is not included in each offline sampled cache. Letting $\hat{\mathcal{M}}$ be the cache returned by algorithm Algorithm 5, then with probability at least $1 - \delta$,*

$$c(\mathcal{M}^*; \boldsymbol{c}, \boldsymbol{p}) - c\left(\hat{\mathcal{M}}; \boldsymbol{c}, \boldsymbol{p}\right) \tag{10}$$

$$\leq 2\sqrt{\frac{2\sum_{q\in\mathcal{Q}-\mathcal{M}^*}\frac{1}{\nu(q)}\log(\frac{6mn}{\delta})}{n}} + 2\sqrt{\frac{2m\log(\frac{3}{\delta})}{n}},$$

*If the experimenter samples empty cache $\mathcal{M}_t = \emptyset$ in each round as in (Zhu et al., 2023) so that $\nu(q) = 1$, it holds that*

$$c(\mathcal{M}^*; \boldsymbol{c}, \boldsymbol{p}) - c\left(\hat{\mathcal{M}}; \boldsymbol{c}, \boldsymbol{p}\right) \leq 4\sqrt{\frac{2m\log(\frac{6mn}{\delta})}{n}}. \tag{11}$$

**Remark 5** (Discussion of Theorem 3). Looking at Eq. (11), our result improves upon the state-of-the-art result (Zhu et al., 2023) $\tilde{O}(k\sqrt{\frac{m}{C_1 n}})$ by a factor of $\tilde{O}(\sqrt{\frac{k^2}{C_1}})$, where $C_1 > 0$ is assumed to be an lower bound of $c(q)$ for $q \in \mathcal{Q}$. This improvement comes from our tight analysis to deal with the triggering probability (the $1/C_1$ can be thought of as minimum triggering probability in their analysis) and from the way that we deal with full-feedback arm $\boldsymbol{p}$. Furthermore, since we use the UCB $\bar{c}$ while they use LCB $\underline{c}$, our algorithm in principle can be generalized to more complex distributions as long as the optimal queries are sufficiently covered. As a by-product, we also consider the online streaming LLM cache bandit setting as in (Zhu et al., 2023) from our CMAB-T point of view, which improves their result $\tilde{O}(\frac{km\sqrt{T}}{C_1})$ by a factor of $\tilde{O}(\frac{k\sqrt{m}}{C_1})$. We defer the details to Appendix G.3.

### 4.3. Offline Learning for Influence Maximization with Extension to Node-level Feedback

Influence maximization (IM) is the task of selecting a small number of seed nodes $S$ in a social network $G(\mathcal{V}, \mathcal{E}, p)$ to maximize the influence spread $\sigma(S; G)$ from these nodes. IM has been intensively studied over the past two decades under various diffusion models, such as the independent cascade (IC) model (Kempe et al., 2003b), the linear threshold (LT) model (Chen et al., 2010), as well as different feedback such as edge-level and node-level feedback models. For the edge-level feedback model, IM smoothly fits into our framework by viewing each edge weight $p_{uv}$ for $(u, v) \in \mathcal{E}$ as the base arm. In this section, we consider a more realistic yet challenging setting where we can only obverse the node-level feedback (Chen et al., 2020; 2021), showing that our framework still applies as long as we can construct a high probability lower bound (LCB) for each base arm (edge). Due to space constraints, we present only our main result here. The detailed setting, algorithm, and theoretical analysis can be found in Appendix C.

**Theorem 4.** *Under Assumption 1, suppose the number of data $n \geq \frac{392\log(\frac{12nE}{\delta})}{\eta \cdot \gamma}$. Letting $\hat{S}$ be the seed set returned by Algorithm 3, then it holds with probability at least $1 - \delta$,*

$$\alpha\sigma(S^*; G) - \sigma\left(\hat{S}; G\right) \tag{12}$$

$$\leq 48\sqrt{6}\sqrt{\frac{V^2 d_{\max}^2 \sigma^2(S^*; G) \cdot \log(\frac{12nE}{\delta})}{\eta \cdot \gamma^3 \cdot n}},$$

*where $d_{\max}$ is the maximum out-degree, $\gamma, \eta$ are lower bounds for seed sampling probability and the activation probability, respectively, as in Assumption 1.*

**Remark 6** (Discussion). To find out an action $\hat{S}$ such that $\sigma(\hat{S}; G) \geq (\alpha - \epsilon)\sigma(S^*; G)$, our algorithm requires that $n \geq \tilde{O}\left(\frac{V^2 d_{\max}^2}{\epsilon^2 \eta\gamma^3}\right)$, which improves the existing result by a factor of $\tilde{O}\left(\frac{V^4}{k^2 d_{\max}^2 \eta}\right)$, owing to our variance-adaptive LCB construction and the tight CMAB-T analysis. We also relax the assumption of Chen et al. (2021) for the seed sampling probability and activation probability, owing to the usage of LCBs, rather than directly using the empirical mean of edge weight $p_{uv}$. See Appendix C for the detailed discussion.

## 5. Experiments

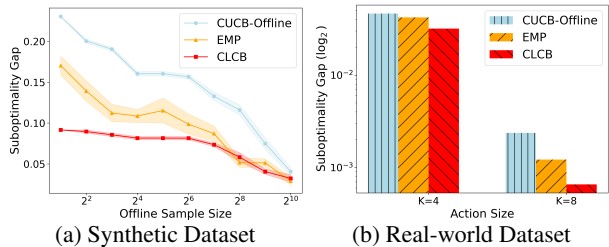

(a) Synthetic Dataset       (b) Real-world Dataset

*Figure 1.* Suboptimality gaps for cascading bandit application.

We now present experimental results on both synthetic and real-world datasets. Each experiment was conducted over 20

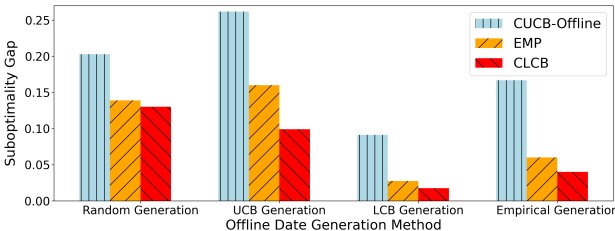

*Figure 2.* Comparison of different offline data generation methods.

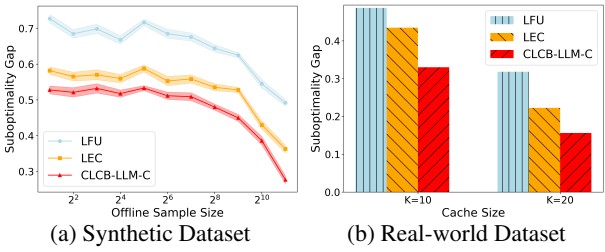

(a) Synthetic Dataset      (b) Real-world Dataset

*Figure 3.* Suboptimality gaps for LLM cache application.

independent trials. For cascading bandits on the application of learning to rank, in the synthetic setting, item parameters $\mu_i$ are drawn from $U[0, 1]$, and in each round $t$, a ranked list $S_t$ of $K$ items is randomly sampled. Fig. 1a shows that compared to CUCB-Offline (Chen et al., 2016), which is an offline adaptation of CUCB for our setting, and EMP (Liu et al., 2021), which always selects the action based on the empirical mean of rewards, CLCB (Algorithm 1) reduces suboptimality gaps by 47.75% and 20.02%, respectively. For real-world evaluation, we use the Yelp dataset[3], where users rate businesses (Dai et al., 2024c). We randomly select $m = 200$ rated items per user and recommend up to $K$ items to maximize the probability of user engagement. The unknown probability $\mu_i$ is derived from Yelp, and cascading feedback is collected. Fig. 1b compares suboptimality gaps over $n = 100$ rounds for $K = 4, 8$, with a logarithmic scale on the y-axis. Note that as $K$ increases, the expected reward also changes, thus reducing suboptimality gaps. CLCB consistently achieves the lowest suboptimality gap.

Moreover, we generate offline datasets $\mathcal{D}$ with $n = 100$ in four different ways: random sampling, UCB-based generation, LCB-based generation, and empirical-based generation. For the UCB-based, LCB-based, and empirical-based data generation methods, we select the top $K = 5$ arms with the largest UCB, LCB, and empirical reward means, respectively. It can be observed in Fig. 2 that our method consistently maintains the smallest suboptimality gap. When using the UCB data generation method, our algorithm performs significantly better than the CUCB-Offline and EMP baselines, which aligns with our theoretical results.

---

[3]https://www.yelp.com/dataset

Similarly, we conduct experiments in the LLM cache setting. In the synthetic setup, we simulate 100 distinct queries with a cache size of 40, following a power-law frequency distribution ($\alpha = 0.9$) as in (Zhu et al., 2023). As shown in Fig. 3a, our CLCB-LLM-C algorithm outperforms LFU (which evicts the least frequently accessed items to optimize cache usage) (Zhu et al., 2023) and LEC (which minimizes inference cost by evicting items with the lowest estimated expected cost) (Zhu et al., 2023), achieving at least $1.32\times$ improvement. For real-world evaluation, we use the SciQ dataset (Welbl et al., 2017). We evaluate GPT-4-o with the "o200k_base" encoding with cache sizes $K = 10$ and $K = 20$, where cost is defined by OpenAI's API pricing with the `tiktoken` library (OpenAI, 2025). Fig. 3b shows that CLCB-LLM-C (Algorithm 2) reduces costs by up to 36.01% and 20.70%, compared to LFU and LEC. Moreover, a larger $K$ shows a lower suboptimality gap, which is consistent with Theorem 3. Further details on experimental setups, results, and additional comparisons can be found in Appendix J.

## 6. Conclusion and Future Directions

In this paper, we introduce Off-CMAB, the first offline learning framework for CMAB. We propose two novel data coverage conditions and develop a provably sample-efficient CLCB algorithm, matching the lower bound up to logarithmic factors. We show the practical usefulness of our framework via three diverse applications—learning to rank, LLM caching, and social influence maximization—achieving new or improved theoretical results. These results are further validated through extensive experiments on both synthetic and real-world datasets. Looking ahead, an exciting direction is to develop variance-adaptive algorithms to further improve our theoretical guarantees. Additionally, extending our framework to offline RL with combinatorial action spaces is another promising direction for future research.

## Acknowledgement

The work is supported by NSF CNS-2103024 and the Office of Naval Research under grant N000142412073. The work of John C.S. Lui was supported in part by the RGC GRF-14202923. The work of Jinhang Zuo was supported by CityUHK 9610706.

## Impact Statement

This paper presents a theoretical study on multi-armed bandits and reinforcement learning. There are many potential societal consequences of our work, none of which we feel must be specifically highlighted here.

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

## A. Extended Related Works

### A.1. Combinatorial Multi-armed Bandits

The combinatorial multi-armed bandit (CMAB) problem has been extensively studied over the past decade, covering domains such as stochastic CMAB (Gai et al., 2012; Kveton et al., 2015c; Combes et al., 2015; Chen et al., 2016; Wang & Chen, 2017; Merlis & Mannor, 2019; Saha & Gopalan, 2019; Agrawal et al., 2019; Liu et al., 2022; 2024b), adversarial CMAB (György et al., 2007; Uchiya et al., 2010; Cesa-Bianchi & Lugosi, 2012; Bubeck et al., 2012; Audibert et al., 2014; Neu, 2015; Han et al., 2021), and hybrid best-of-both-worlds settings (Zimmert et al., 2019; Ito, 2021; Tsuchiya et al., 2023). Contextual extensions with linear or nonlinear function approximation have also been explored (Qin et al., 2014; Takemura et al., 2021; Liu et al., 2023a; Chen et al., 2018; Nika et al., 2020; Choi et al., 2024; Hwang et al., 2023; Liu et al., 2024b).

Our work falls within the stochastic CMAB with semi-bandit feedback domain, first introduced by Gai et al. (2012), with a specific focus on CMAB with probabilistically triggered arms (CMAB-T). Chen et al. (2016) introduced the concept of arm triggering processes for applications like cascading bandits and influence maximization, proposing the CUCB algorithm with a regret bound of $O(B_1\sqrt{mKT\log T}/p_{\min})$ regret bound under the 1-norm smoothness condition with coefficient $B_1$. Subsequently, Wang & Chen (2017) refined this result, proposed a stronger 1-norm triggering probability modulated

(TPM) $B_1$ smoothness condition, and employed triggering group analysis to eliminate the $1/p_{\min}$ factor from the previous regret bound. More recently, Liu et al. (2022) leveraged the variance-adaptive principle to propose the BCUCB-T algorithm, which further reduces the regret's dependency on action-size from $O(K)$ to $O(\log K)$ under the new variance and triggering probability modulated (TPVM) condition. While inspired by these works, our study diverges by addressing the offline CMAB setting, where online exploration is unavailable, and the focus is on minimizing the suboptimality gap rather than regret.

Another notable line of work considers CMAB with full bandit feedback (György et al., 2007; Cesa-Bianchi & Lugosi, 2012; Niazadeh et al., 2021; Fourati et al., 2023; Nie et al., 2023; Fourati et al., 2024b;a; Sun et al., 2025). In their setting, the feedback only provides aggregate rewards for the entire super arm, often resulting in higher regret (e.g., $O(T^{2/3})$) and requiring fundamentally different oracle designs. In contrast, our setting (CMAB with semi-bandit feedback) assumes semi-bandit feedback, where the learner observes individual arm-level feedback for selected arms (i.e., components of the super arm). This enables more informative learning and allows us to construct accurate base-arm estimators for use in our oracles, leading to an $O(T^{1/2})$ regret bound. Moreover, prior full-bandit approaches often rely on additional structural assumptions such as submodularity to achieve these bounds. Similarly, our approach leverages smoothness assumptions on the reward function to ensure statistical efficiency in the offline regime.

### A.2. Offline Bandit and Reinforcement Learning

Offline reinforcement learning (RL), also known as "batch RL", focuses on learning from pre-collected datasets to make sequential decisions without online exploration. Initially studied in the early 2000s (Ernst et al., 2005; Riedmiller, 2005; Lange et al., 2012), offline RL has gained renewed interest in recent years (Levine et al., 2020).

From an empirical standpoint, offline RL has achieved impressive results across diverse domains, including robotics (Singh et al., 2021), healthcare (Liu et al., 2020), recommendation systems (Chen et al., 2023), autonomous driving (Kiran et al., 2020), and large language model fine-tuning and alignment (Casper et al., 2023). Algorithmically, offline RL approaches can be broadly categorized into policy constraint methods (Fujimoto et al., 2018; Kumar et al., 2019), pessimistic value/policy regularization (Haarnoja et al., 2018; Kumar et al., 2020), uncertainty estimation (Agarwal et al., 2019), importance sampling (Jiang & Li, 2015; Nachum et al., 2019), imitation learning (Fujimoto & Gu, 2021; Chen et al., 2019), and model-based methods (Kidambi et al., 2020; Yu et al., 2021).

Theoretically, early offline RL studies relied on strong uniform data coverage assumptions (Szepesvari & Munos, 2005; Chen & Jiang, 2019a; Wang et al., 2019; Xie et al., 2021a). Recent works have relaxed these assumptions to partial coverage for tabular Markov Decision Processes (MDPs) (Rashidinejad et al., 2021; Yin et al., 2021; Shi et al., 2022; Li et al., 2022b), linear MDPs (Jin et al., 2020b; Chang et al., 2021; Bai et al., 2022), and general function approximation settings (Rashidinejad et al., 2022; Zanette et al., 2021; Xie et al., 2021b; Zanette & Wainwright, 2022).

Offline bandit learning has also been explored in multi-armed bandits (MAB) (Rashidinejad et al., 2021), contextual MABs (Rashidinejad et al., 2021; Jin et al., 2020b; Li et al., 2022a), and neural contextual bandits (Nguyen-Tang et al., 2021b;a).

While our work leverages the pessimism principle and focuses on partial coverage settings, none of the aforementioned offline bandit or RL studies address the combinatorial action space, which is the central focus of our work. Conversely, recent work in the CMAB framework demonstrates that episodic tabular RL can be viewed as a special case of CMAB (Liu et al., 2024b). Building on this connection, our proposed framework can potentially extend to certain offline RL problems, offering a unified approach to tackle both combinatorial action spaces and offline learning.

### A.3. Related Offline Learning Applications.

Cascading bandits, a classical online learning-to-rank framework, have been extensively studied in the literature (Kveton et al., 2015a;b; Li et al., 2016; Wang & Chen, 2018; Vial et al., 2022; Zhong et al., 2021; Liu et al., 2022; Wang et al., 2023; 2024). Offline cascading bandits, on the other hand, focus primarily on reducing bias in learning settings (Joachims, 2002; Wang et al., 2018; 2016; Keane & O'Brien, 2006; Zhang et al., 2023). Unlike these prior works, our study tackles the unbiased setting where data coverage is insufficient. Moreover, we are the first to provide a theoretically guaranteed solution using a CMAB-based approach.

LLM caching is a memory management technique aimed at mitigating memory footprints and access overhead during training and inference. Previous studies have investigated LLM caching at various levels, including attention-level (KV-cache) (Pope et al., 2022; Kwon et al., 2023; Sheng et al., 2023; Bang, 2023), query-level (Gim et al., 2023; Zhu et al., 2023),

and model/API-level (Qu et al., 2024; Dai et al., 2024a; Feng et al., 2024). Among these, the closest related work is the LLM cache bandit framework proposed by Zhu et al. (2023). However, their approach is ad hoc, whereas our CMAB-based framework systematically tackles the same problem and achieves improved results in both offline and online settings.

Influence Maximization (IM) was initially formulated as an algorithmic problem by Richardson & Domingos (2002) and has since been studied using greedy approximation algorithms (Kempe et al., 2003b; Chen et al., 2009). The online IM problem has also received significant attention (Wen et al., 2017; Vaswani et al., 2017a; Wu et al., 2019; Vaswani et al., 2017b; 2015; Wang & Chen, 2017). In the offline IM domain, our work aligns closely with the optimization-from-samples (OPS) framework (Balkanski et al., 2015; 2016; Chen et al., 2020; 2021), originally proposed by Balkanski et al. (2015). Specifically, our work falls under the subdomain of optimization-from-structured-samples (OPSS) (Chen et al., 2020; 2021), where samples include detailed diffusion step information $(S_0, ..., S_{V-1})$ instead of only the final influence spread $\sigma(S_0; G)$ in the standard OPS. Compared to Chen et al. (2021), which selects the best seed set using empirical means, our approach employs a variance-adaptive pessimistic LCB, improving the suboptimal gap under relaxed assumptions.

## B. More Justification of Studying Offline CMAB

While online bandits are a natural choice when online data is readily available and inexpensive, many real-world applications restrict access to only offline data as follows, which motivates the study of offline CMAB.

For instance, consider the cascading bandit model in recommendation systems (Kveton et al., 2015a). Online CMAB learning requires a tight feedback loop where the platform (i.e., the learner) updates its recommendation policy after every user interaction. However, in many practical scenarios, such fine-grained online feedback is unavailable as the platform cannot afford to update at such a high frequency. Instead, data is collected in batches (e.g., over a week), logged, and then used to update the policy in a single offline training phase. This workflow aligns precisely with our offline CMAB setting.

Another motivating scenario involves outsourced system design. For example, if OpenAI or Anthropic outsources the design of an LLM caching system, the consultant (i.e., the learner) typically receives only anonymized user logs. They must learn user behavior and design the system purely based on this private offline dataset and cannot reach out for direct interaction with the users, which fits naturally into the offline CMAB framework.

Moreover, our work on CMAB also mirrors the development trajectory in reinforcement learning (RL). RL began with a focus on online learning (Mnih et al., 2013); then, around 2020, concerns over the cost and availability of online interactions led to a growing emphasis on offline RL—learning solely from logged data (Levine et al., 2020). More recently, hybrid approaches (Lee et al., 2022) combining offline pretraining with online fine-tuning have emerged. Similarly, after establishing foundational results in online CMAB, we now focus on the offline setting as a crucial step toward enabling future hybrid CMAB approaches.

## C. Offline Learning for Influence Maximization with Extension to Node-level Feedback

Influence maximization (IM) is the task of selecting a small number of seed nodes in a social network to maximize the influence spread from these nodes, which has been applied in various important applications such as viral marketing, epidemic control, and political campaigning (Richardson & Domingos, 2002; Kempe et al., 2003b; Chen et al., 2009). IM has been intensively studied over the past two decades under various diffusion models, such as the independent cascade (IC) model (Kempe et al., 2003b), the linear threshold (LT) model (Chen et al., 2010), and the voter model (Narasimhan et al., 2015)], as well as different feedback such as edge-level and node-level feedback models (Chen et al., 2020). For the edge-level feedback model, IM smoothly fits into our framework by viewing each edge as the base arm, which can obtain the theoretical result similar to our previous two applications. In this section, we consider a more realistic yet challenging setting where we can only obverse the node-level feedback, showing that our framework still applies as long as we can construct a high probability lower bound (LCB) for each base arm (edge).

**Influence maximization under the independent cascade diffusion model.** We consider a weighted digraph $G(\mathcal{V}, \mathcal{E}, p)$ to model the social network, where $\mathcal{V}$ is the set of nodes and $\mathcal{E}$ is the set of edges, with cardinality $V = |\mathcal{V}|$ and $E = |\mathcal{E}|$, respectively. For each edge $(u, v) \in \mathcal{E}$, it is associated with a weight or probability $p_{uv} \in [0, 1]$. We use $N(v) = N^{\text{in}}(v)$ to denote the in-neighbors of node $v \in \mathcal{V}$.

The diffusion model describes how the information propagates, which is detailed as follows. Denote $S_0 \subseteq \mathcal{V}$ as the seed nodes and $S_h \subseteq \mathcal{V}$ as the set of active nodes at time steps $h \geq 1$. By default, we let $S_{-1} = \emptyset$. In the IC model,

at time step $h \geq 1$, for each node $v \notin S_{h-1}$, each newly activated node in the last step, $u \in N(v) \cup (S_{h-1} \backslash S_{h-2})$, will try to active $v$ independently with probability $p_{uv}$. This indicates that $v$ will become activated with probability $1 - \prod_{u \in N(v) \cup (S_{h-1} \backslash S_{h-2})} (1 - p_{uv})$. Once activated, $v$ will be added into $S_h$. The propagation ends at step $h$ when $S_h = S_{h-1}$. It is obvious that the propagation process proceeds in at most $V - 1$ time steps, so we use $(S_0, S_1, ..., S_{V-1})$ to denote the random sequence of the active nodes, which we refer to as *influence cascade*. Let $\Phi(S_0) = S_{V-1}$ be the final active node set given the seed nodes $S_0$. The influence maximization problem aims to select at most $k$ seed nodes so as to maximize the expected number of active nodes $\sigma(S_0; G) := \mathbb{E}[|\Phi(S_0)|]$, which we often refer to as the *influence spread* of $S_0$ given the graph $G$. Formally, the IM problem aims to solve $S^* = \operatorname{argmax}_{S \subseteq \mathcal{V}} \sigma(S; G)$.

**Offline dataset and learning from the node-level feedback.** We consider the offline learning setting for IM where the underlying graph $G$ is unknown. To find the optimal seed set $S^*$, we are given a pre-collected dataset consisting of $n$ influence cascades $\mathcal{D} = (S_{t,0}, S_{t,1}, ..., S_{t,V-1})_{t=1}^n$ and a probability $\delta$. Our goal is to output a seed node set $\hat{S}(\mathcal{D}, \delta)$, whose influence spread is as large as possible with high probability $1 - \delta$.

Similar to (Chen et al., 2021), we assume these $n$ influence cascades are generated independently from a seed set distribution $S_{t,0} \sim \mathbb{D}_\mathcal{S}$, and given $S_{t,0}$, the cascades are generated according to the IC diffusion process. For each node $v \in \mathcal{V}$, we use $q_v = \Pr[v \in S_{t,0}]$ to denote the probability that the node $v$ is selected by the experimenter in the seed set $S_{t,0}$. We use $p_G(\bar{v}) := \Pr[v \notin S_{t,1}]$ to denote the probability that the node $v$ is not activated in one time step when the graph is $G$. For any two nodes $u, v \in \mathcal{V}$, we use $p_G(\bar{v}|u) := \Pr[v \notin S_{t,1}|u \in S_{t,0}]$ and $p_G(\bar{v}|\bar{u}) := \Pr[v \notin S_{t,1}|u \notin S_{t,0}]$ to denote the probability that the node $v$ is not activated in one time step conditioned on whether the node $u$ is in the seed set $S_{t,0}$ or not, respectively. We also assume $\mathbb{D}_\mathcal{S}$ is a product distribution, i.e., each node $u \in \mathcal{V}$ is selected as a seed node in $S_{t,0}$ *independently*. Similar to (Chen et al., 2021), we also need an additional Assumption 1.

**Assumption 1** (Bounded seed node sampling probability and bounded activation probability). *Let $\tilde{\mathcal{E}}(S^*) \subseteq \mathcal{E}$ be the set of edges that can be triggered by the optimal seed set $S^*$. There exist parameters $\eta \in (0, 1]$ and $\alpha \in (0, 1/2]$ such that for any $(u, v) \in \tilde{\mathcal{E}}(S^*)$, we have $q_u \in [\gamma, 1 - \gamma]$ and $p_G(\bar{v}) \geq \eta$.*

**Algorithm that constructs variance-adaptive LCB using the node-level feedback.** Note that in this setting, we cannot obtain edge-level feedback about which node influences which node in the dataset. It is an extension which cannot be directly handled by Algorithm 1 since one cannot directly estimate the edge weight from the node-level feedback. However, as long as we can obtain a high probability LCB for each arm $(u, v) \in \mathcal{E}$ and replace the line 5 of Algorithm 1 with this new LCB, we can still follow a similar analysis to bound its suboptimality gap. Our algorithm is presented in Algorithm 3.

Inspired by (Chen et al., 2021), for each node-level feedback data $t \in [n]$, we only use the seed set $S_{t,0}$ and the active nodes in the first diffusion step $S_{t,1}$ to construct the LCB.

Since each node $u$ is independently selected in $S_{t,0}$ with probability $q_u$, and we consider only one step activation for any node $v$, the event $\{v$ is activated by $u\}$ and the event $\{v$ is activated by other nodes $G - \{u\}\}$ are independent. Thus, we have $p_G(\bar{v}) = (1 - q_u p_{uv}) \cdot p_{G \backslash \{u\}}(\bar{v}) = (1 - q_u p_{uv}) \cdot p_G(\bar{v}|\bar{u})$. Rearranging terms, we have:

$$p_{uv} = \frac{1}{q_u} \left( 1 - \frac{p_G(\bar{v})}{p_G(\bar{v}|\bar{u})} \right). \tag{13}$$

Let us omit the graph $G$ in the subscript of $p_G(\bar{v})$ and $p_G(\bar{v}|\bar{u})$ when the context is clear.

We can observe that $p_{uv}$ is monotonically decreasing when $q_u$ or $p(\bar{v})$ increases and when $p(\bar{v}|\bar{u})$ decreases. Therefore, we separately construct intermediate UCB for $q_u, p(\bar{v})$ and LCB for $p(\bar{v}|\bar{u})$ and plug into Eq. (13) to construct an overall LCB $\underline{p}_{uv}$ for each arm $p_{uv}$ as in line 7. Based on the LCB for each edge, we construct the LCB graph $\underline{G}$ and call IM oracle over $\underline{G}$. Also note that for each intermediate UCB/LCB, we use variance-adaptive confidence intervals to further reduce the estimation bias.

**Theorem 5.** *Under Assumption 1, suppose the number of data $n \geq \frac{392 \log(\frac{12nE}{\delta})}{\eta \cdot \gamma}$. Let $\hat{S}$ be the seed set returned by Algorithm 3, then it holds with probability at least $1 - \delta$ that*

$$\alpha\sigma(S^*; G) - \sigma\left(\hat{S}; G\right) \leq 48\sqrt{6}\sqrt{\frac{V^2 d_{\max}^2 \sigma^2(S^*; G) \cdot \log(\frac{12nE}{\delta})}{\eta \cdot \gamma^3 \cdot n}}, \tag{14}$$

*where $d_{\max}$ is the maximum out-degree of the graph $G$.*

---

**Algorithm 3** `CLCB-IM-N`: Combinatorial Lower Confidence Bound Algorithm for Influence Maximization with Node-level Feedback

1: **Input:** Dataset $\mathcal{D} = \{(S_{t,0}, S_{t,1}, ..., S_{t,V-1})\}_{t=1}^n$, nodes $\mathcal{V}$, edges $\mathcal{E}$, cardinality $k$, influence maximization solver `IM`, probability $\delta$.
2: **for** edge $(u,v) \in \mathcal{E}$ **do**
3:     Calculate counters $n_{0,u} = |\{i \in [n] : u \in S_{i,0}\}|, n_{1,\bar{v}} = |\{i \in [n] : v \notin S_{i,1}\}|, n_{1,\bar{u},\bar{v}} = |\{i \in [n] : u \notin S_{i,0} \text{ and } v \notin S_{i,1}\}|$;
4:     Calculate empirical means $\hat{q}_u = n_{0,u}/n, \hat{p}(\bar{v}) = n_{1,\bar{v}}/n, \hat{p}(\bar{v}|\bar{u}) = n_{1,\bar{u},\bar{v}}/n_{1,\bar{v}}$;
5:     Calculate variance-adaptive intervals $\rho_u = \sqrt{\frac{6(1-\hat{q}_u)\hat{q}_u \log(\frac{12nE}{\delta})}{n}} + \frac{9\log(\frac{12nE}{\delta})}{n}, \rho(\bar{v}) = \sqrt{\frac{6(1-\hat{p}(\bar{v}))\hat{p}(\bar{v}) \log(\frac{12nE}{\delta})}{n}} + \frac{9\log(\frac{12nE}{\delta})}{n}, \rho(\bar{v}|\bar{u}) = \sqrt{\frac{6(1-\hat{p}(\bar{v}|\bar{u}))\hat{p}(\bar{v}|\bar{u}) \log(\frac{12nE}{\delta})}{n_{0,\bar{u}}}} + \frac{9\log(\frac{12nE}{\delta})}{n_{0,\bar{u}}}$;
6:     Compute intermediate UCB/LCB $\bar{q}_u = \min\{\hat{q}_u + \rho_u, 1\}, \bar{p}(\bar{v}) = \min\{\hat{p}(\bar{v}) + \rho(\bar{v}), 1\}, \underline{p}(\bar{v}|\bar{u}) = \max\{\hat{p}(\bar{v}|\bar{u}) - \rho(\bar{v}|\bar{u}), 0\}$;
7:     Compute edge-level LCB $\underline{p}_{uv} = \min\left\{1, \max\left\{0, \frac{1}{\bar{q}_u}\left(1 - \frac{\bar{p}(\bar{v})}{\underline{p}(\bar{v}|\bar{u})}\right)\right\}\right\}$ for $(u,v) \in \mathcal{E}$.
8: **end for**
9: Construct LCB graph $\underline{G} = (\mathcal{V}, \mathcal{E}, \underline{p})$ with edge-level LCB $\underline{p} = (\underline{p}_{uv})_{(u,v)\in\mathcal{E}}$.
10: Call IM sovler $\hat{S} = \text{IM}(\underline{G}, k)$.
11: **Return:** $\hat{S}$.

---

**Remark 7** (Discussion). To find out an action $\widehat{S}$ such that $\sigma(\widehat{S}; G) \geq (\alpha - \epsilon)\sigma(S^*; G)$, our algorithm requires that $n \geq \tilde{O}\left(\frac{V^2 d_{\max}^2}{\epsilon^2 \eta\gamma^3}\right)$, which improves the existing result by at least a factor of $\tilde{O}\left(\frac{V^4}{k^2 d_{\max}^2 \eta}\right)$, owing to our variance-adaptive LCB construction and the tight CMAB-T analysis. We also relax the assumption regarding Assumption 1, where we require bounded $q_u, p(\bar{v})$ only for $(u,v) \in \mathcal{E}(s^*)$, since we use LCB $\underline{p}_{uv}$. Chen et al. (2021), instead, needs bounded $q_u, p(\bar{v})$ for all $(u,v) \in \mathcal{E}$ as they directly use the empirical mean of $p_{uv}$.

## D. Proof for the Upper Bound Result

*Proof of Theorem 1.* We first show the regret bound under the infinity-norm TPM data coverage condition (Condition 3):

Let $N_i(\mathcal{D})$ be the counter for arm $i$ as defined in line 3 of Algorithm 1, given the dataset $\mathcal{D}$ and the failure probability $\delta$.

Let $\hat{\boldsymbol{\mu}}(\mathcal{D}) = (\hat{\mu}_1(\mathcal{D}, \delta), ..., \hat{\mu}_m(\mathcal{D}, \delta))$ be the empirical mean defined in line 4 of Algorithm 1.

Let $\boldsymbol{\mu}(\mathcal{D}, \delta) = (\mu_1(\mathcal{D}, \delta), ..., \mu_m(\mathcal{D}, \delta))$ be the LCB vector defined in line 5 of Algorithm 1.

Let $\hat{S}(\mathcal{D}, \delta)$ be the action returned by Algorithm 1 in line 7.

Let $p_i^{\mathbb{D}_{\text{arm}}, \mathbb{D}_{\mathcal{S}}}$ be the data collecting probability that for arm $i$, i.e., the probability of observing arm $i$ in each offline data.

Let $\tilde{S}^* = \{i \in [m] : p_i^{\text{arm}, S^*} > 0\}$ be the arms that can be triggered by the optimal action $S^*$ and $p^* = \min_{i \in \tilde{S}^*} p_i^{\mathbb{D}_{\text{arm}}, \mathbb{D}_{\mathcal{S}}}$ be the minimum data collection probability.

We first define the events $\mathcal{E}_{\text{arm}}$ and $\mathcal{E}_{\text{counter}}$ as follows.

$$\mathcal{E}_{\text{arm}} := \left\{|\hat{\mu}_i(\mathcal{D}) - \mu_i| \leq \sqrt{\frac{\log(\frac{2mn}{\delta})}{2N_i(\mathcal{D})}} \text{ for any } i \in [m]\right\} \tag{15}$$

$$\mathcal{E}_{\text{counter}} := \left\{N_i(\mathcal{D}) \geq \frac{n \cdot p_i^{\mathbb{D}_{\text{arm}}, \mathbb{D}_{\mathcal{S}}}}{2} \text{ for any } i \in \tilde{S}^* \,\middle|\, n \geq \frac{8\log\frac{m}{\delta}}{p^*}\right\} \tag{16}$$

When $n \geq \frac{8\log\frac{m}{\delta}}{p^*}$ and under the events $\mathcal{E}_{\text{arm}}$ and $\mathcal{E}_{\text{counter}}$, we have the following gap decomposition:

$$\alpha r(S^*; \boldsymbol{\mu}) - r\left(\hat{S}(\mathcal{D}, \delta); \boldsymbol{\mu}\right) \tag{17}$$

$$\overset{(a)}{=} \underbrace{\alpha r(S^*; \boldsymbol{\mu}) - \alpha r\left(S^*; \underline{\boldsymbol{\mu}}(\mathcal{D}, \delta)\right)}_{\text{uncertainty gap}} \tag{18}$$

$$+ \underbrace{\alpha r\left(S^*; \underline{\boldsymbol{\mu}}(\mathcal{D}, \delta)\right) - r\left(\hat{S}(\mathcal{D}, \delta); \underline{\boldsymbol{\mu}}(\mathcal{D}, \delta)\right)}_{\text{oracle gap}} + \underbrace{r\left(\hat{S}(\mathcal{D}, \delta); \underline{\boldsymbol{\mu}}(\mathcal{D}, \delta)\right) - r\left(\hat{S}(\mathcal{D}, \delta); \boldsymbol{\mu}\right)}_{\text{pessimism gap}}$$

$$\overset{(b)}{\leq} \alpha \left(r(S^*; \boldsymbol{\mu}) - r\left(S^*; \underline{\boldsymbol{\mu}}(\mathcal{D}, \delta)\right)\right) \tag{19}$$

$$\overset{(c)}{\leq} \alpha B_1 \sum_{i \in [m]} p_i^{\mathbb{D}_{\text{arm}}, S^*} \left(\mu_i - \underline{\mu}_i(\mathcal{D}, \delta)\right) \tag{20}$$

$$\overset{(d)}{\leq} 2\alpha B_1 \sum_{i \in [m]} p_i^{\mathbb{D}_{\text{arm}}, S^*} \sqrt{\frac{\log(\frac{2mn}{\delta})}{2 N_i(\mathcal{D})}} \tag{21}$$

$$\overset{(e)}{\leq} 2\alpha B_1 \sum_{i \in [m]} p_i^{\mathbb{D}_{\text{arm}}, S^*} \sqrt{\frac{\log(\frac{2mn}{\delta})}{n \cdot p_i^{\mathbb{D}_{\text{arm}}, \mathbb{D}_{\mathcal{S}}}}} \tag{22}$$

$$= 2\alpha B_1 \sum_{i \in [m]} \sqrt{p_i^{\mathbb{D}_{\text{arm}}, S^*}} \sqrt{\frac{\log(\frac{2mn}{\delta}) \cdot p_i^{\mathbb{D}_{\text{arm}}, S^*}}{n \cdot p_i^{\mathbb{D}_{\text{arm}}, \mathbb{D}_{\mathcal{S}}}}} \tag{23}$$

$$\overset{(f)}{\leq} 2\alpha B_1 \bar{K}_2^* \sqrt{\frac{\log(\frac{2mn}{\delta}) \cdot C_\infty^*}{n}}, \tag{24}$$

where inequality (a) is due to adding and subtracting $\alpha r\left(S^*; \underline{\boldsymbol{\mu}}(\mathcal{D}, \delta)\right)$ and $r\left(\hat{S}(\mathcal{D}, \delta); \underline{\boldsymbol{\mu}}(\mathcal{D}, \delta)\right)$, inequality (b) is due to oracle gap $\leq 0$ by Eq. (1) as well as pessimism gap $\leq 0$ by monotonicity (Condition 1) and Lemma 5, inequality (c) is due to 1-norm TPM smoothness condition (Condition 2), inequality (d) is due to Lemma 5, inequality (e) is due to event $\mathcal{E}_{counter}$, inequality (f) is due to infinity-norm TPM data coverage condition (Condition 3).

Next, we show the regret bound under the 1-nrom TPM data coverage condition (Condition 4).

When $n \geq \frac{8 \log \frac{m}{\delta}}{p^*}$ and under the events $\mathcal{E}_{\text{arm}}$ and $\mathcal{E}_{\text{counter}}$, we follow the proof from Eq. (17) to Eq. (23) and proceed as:

$$\alpha r(S^*; \boldsymbol{\mu}) - r\left(\hat{S}(\mathcal{D}, \delta); \boldsymbol{\mu}\right) \tag{25}$$

$$= \underbrace{\alpha r(S^*; \boldsymbol{\mu}) - \alpha r\left(S^*; \underline{\boldsymbol{\mu}}(\mathcal{D}, \delta)\right)}_{\text{uncertainty gap}} \tag{26}$$

$$+ \underbrace{\alpha r\left(S^*; \underline{\boldsymbol{\mu}}(\mathcal{D}, \delta)\right) - r\left(\hat{S}(\mathcal{D}, \delta); \underline{\boldsymbol{\mu}}(\mathcal{D}, \delta)\right)}_{\text{oracle gap}} + \underbrace{r\left(\hat{S}(\mathcal{D}, \delta); \underline{\boldsymbol{\mu}}(\mathcal{D}, \delta)\right) - r\left(\hat{S}(\mathcal{D}, \delta); \boldsymbol{\mu}\right)}_{\text{pessimism gap}}$$

$$\leq \alpha \left(r(S^*; \boldsymbol{\mu}) - r\left(S^*; \underline{\boldsymbol{\mu}}(\mathcal{D}, \delta)\right)\right) \tag{27}$$

$$\leq \alpha B_1 \sum_{i \in [m]} p_i^{\mathbb{D}_{\text{arm}}, S^*} \left(\mu_i - \underline{\mu}_i(\mathcal{D}, \delta)\right) \tag{28}$$

$$\leq 2\alpha B_1 \sum_{i \in [m]} \sqrt{p_i^{\mathbb{D}_{\text{arm}}, S^*}} \sqrt{\frac{\log(\frac{2mn}{\delta}) \cdot p_i^{\mathbb{D}_{\text{arm}}, S^*}}{n \cdot p_i^{\mathbb{D}_{\text{arm}}, \mathbb{D}_{\mathcal{S}}}}} \tag{29}$$

$$\overset{(a)}{\leq} 2\alpha B_1 \sqrt{\sum_{i \in [m]} p_i^{\mathbb{D}_{\text{arm}}, S^*}} \sqrt{\sum_{i \in [m]} \frac{\log(\frac{2mn}{\delta}) \cdot p_i^{\mathbb{D}_{\text{arm}}, S^*}}{n \cdot p_i^{\mathbb{D}_{\text{arm}}, \mathbb{D}_{\mathcal{S}}}}} \tag{30}$$

$$\overset{(b)}{\leq} 2\alpha B_1 \sqrt{\frac{\bar{K}^* C_1^* \log(\frac{2mn}{\delta})}{n}}, \tag{31}$$

where inequality (a) is due to Cauchy Schwarz inequality, and inequality (b) is due to 1-norm TPM data coverage condition

Condition 4.

The final step is to show event $\mathcal{E}_{\text{arm}}$ and $\mathcal{E}_{\text{counter}}$ hold with high probability. By Lemma 5 and Lemma 6, event $\mathcal{E}_{\text{arm}}$ and $\mathcal{E}_{\text{counter}}$ both hold with probability at least with $1 - \delta$. By setting $\delta' = \delta/2$ concludes the proof.

∎

# E. Proof for the Lower Bound Result

*Proof of Theorem 2.* Let $\Delta \in [0, \frac{1}{4}]$ be a gap to be tuned later and let $C_\infty^* \geq 2$. We consider a $k$-path problem with two problem instances $\mathcal{P}_1$ and $\mathcal{P}_2$, where $m/k$ path's mean vectors are $\boldsymbol{\mu}_1 = (\frac{1}{2}, \frac{1}{2} - \Delta, 0, ..., 0) \in \mathbb{R}^{m/k}$ and $\boldsymbol{\mu}_2 = (\frac{1}{2}, \frac{1}{2} + \Delta, 0, ..., 0) \in \mathbb{R}^{m/k}$, respectively. For the data collecting distribution, $\mathbb{D}_\mathcal{S}$ follows $\boldsymbol{p} = (\frac{1}{C_\infty^*}, 1 - \frac{1}{C_\infty^*}, 0, ..., 0)$ for both $\mathcal{P}_1$ and $\mathcal{P}_2$. We have that the optimal action $S_1^* = (1, 2, ..., k)$ for $\mathcal{P}_1$ and $S_2^* = (k+1, k+2, ..., 2k)$ for $\mathcal{P}_2$.

For the triggering probability, $p_i^{\mathcal{P}_1, S^*} = 1$ for $i = 1, ..., k$ and $0$ otherwise and $p_i^{\mathcal{P}_2, S^*} = 1$ for $i = k+1, ..., 2k$ and $0$ otherwise.

We then show that both problem instances $\mathcal{P}_1, \mathcal{P}_2$ satisfy Condition 3. For $\mathcal{P}_1$ and $\mathcal{P}_2$, we have

$$\max_{i \in [m]} \frac{p_i^{\mathcal{P}_1, S^*}}{p_i^{\mathcal{P}_1, \mathcal{D}_\mathcal{S}}} = \frac{1}{\frac{1}{C_\infty^*}} = C_\infty^*, \tag{32}$$

$$\max_{i \in [m]} \frac{p_i^{\mathcal{P}_2, S^*}}{p_i^{\mathcal{P}_2, \mathcal{D}_\mathcal{S}}} = \frac{1}{1 - \frac{1}{C_\infty^*}} \overset{(a)}{\leq} C_\infty^*, \tag{33}$$

where inequality $(a)$ is due to $C_\infty^* \geq 2$.

Let us define suboptimality gap of any action $\hat{S}$ as:

$$g(\hat{S}; \boldsymbol{\mu}) := r(S^*(\boldsymbol{\mu}); \boldsymbol{\mu}) - r(\hat{S}; \boldsymbol{\mu}) \tag{34}$$

where $S^*(\boldsymbol{\mu})$ is the optimal super arm under $\boldsymbol{\mu}$.

For any action $\hat{S} \in \mathcal{S}$, we have

$$g(\hat{S}; \boldsymbol{\mu}_1) + g(\hat{S}; \boldsymbol{\mu}_2) \geq k\Delta \tag{35}$$

Recall that $A(\mathcal{D})$ is the action returned by algorithm $A$ and we use the Le Cam's method (Le Cam, 2012):

$$\inf_A \sup_{(\mathbb{D}_{\text{arm}}, \mathbb{D}_\mathcal{S}) \in \text{k-path}(m,k,C_\infty^*)} \mathbb{E}_{\mathcal{D} \sim \mathbb{D}(\mathbb{D}_{\text{arm}}, \mathbb{D}_\mathcal{S})}[r(S^*; \mu) - r(A(\mathcal{D}); \mu)] \tag{36}$$

$$\geq \inf_A \sup_{\boldsymbol{\mu} \in \boldsymbol{\mu}_1, \boldsymbol{\mu}_2} \mathbb{E}_\mathcal{D}[g(A(\mathcal{D}); \boldsymbol{\mu})] \tag{37}$$

$$\overset{(a)}{\geq} \inf_A \frac{1}{2} \left( \mathbb{E}_{\boldsymbol{p} \otimes \boldsymbol{\mu}_1}[A(\mathcal{D}); \boldsymbol{\mu}_1)] + \mathbb{E}_{\boldsymbol{p} \otimes \boldsymbol{\mu}_2}[g(A(\mathcal{D}); \boldsymbol{\mu}_2)] \right) \tag{38}$$

$$\overset{(b)}{\geq} \frac{k\Delta}{4} \exp\left( -\text{KL}\left( \mathbb{P}_{\boldsymbol{p} \otimes \boldsymbol{\mu}_1} || \mathbb{P}_{\boldsymbol{p} \otimes \boldsymbol{\mu}_2} \right) \right), \tag{39}$$

where inequality (a) is due to $\max a, b \geq (a+b)/2$, and inequality (b) is due to the following derivation:

Let event $\mathcal{E} = \{g(A(\mathcal{D}); \boldsymbol{\mu}_1) \leq \frac{k\Delta}{2}\}$. On $\neg\mathcal{E}$ it holds that $g(A(\mathcal{D}); \boldsymbol{\mu}_1) \geq \frac{k\Delta}{2}$ and on $\mathcal{E}$ it holds that $g(A(\mathcal{D}); \boldsymbol{\mu}_2) \geq k\Delta - g(A(\mathcal{D}); \boldsymbol{\mu}_1) \geq \frac{k\Delta}{2}$. Thus, we have:

$$\inf_A \frac{1}{2} \left( \mathbb{E}_{\boldsymbol{p} \otimes \boldsymbol{\mu}_1}[g(A(\mathcal{D}); \boldsymbol{\mu}_1)] + \mathbb{E}_{\boldsymbol{p} \otimes \boldsymbol{\mu}_2}[g(A(\mathcal{D}); \boldsymbol{\mu}_2)] \right) \tag{40}$$

$$\geq \frac{k\Delta}{4} \left( \mathbb{P}_{\boldsymbol{p} \otimes \boldsymbol{\mu}_1}(\neg\mathcal{E}) + \mathbb{P}_{\boldsymbol{p} \otimes \boldsymbol{\mu}_2}(\mathcal{E}) \right) \tag{41}$$

$$\overset{(a)}{\geq} \frac{k\Delta}{8} \exp\left( -\text{KL}\left( \mathbb{P}_{\boldsymbol{p} \otimes \boldsymbol{\mu}_1} || \mathbb{P}_{\boldsymbol{p} \otimes \boldsymbol{\mu}_2} \right) \right) \tag{42}$$

$$\overset{(b)}{\geq} \frac{k}{8e} \min\left(\frac{1}{4}, \sqrt{\frac{C_\infty^\star}{20n}}\right) \tag{43}$$

$$\tag{44}$$

where inequality (a) is due to Lemma 8 and inequality (b) comes from: $\mathrm{KL}\left(\mathbb{P}_{\boldsymbol{p}\otimes\boldsymbol{\mu}_1}\|\mathbb{P}_{\boldsymbol{p}\otimes\boldsymbol{\mu}_2}\right) \leq \frac{n\mathrm{KL}\left(\mathbb{P}_{\boldsymbol{\mu}_1}\|\mathbb{P}_{\boldsymbol{\mu}_2}\right)}{C_\infty^\star} \leq \frac{n(2\Delta)^2}{C_\infty^\star(1/4-\Delta^2)} \leq 20n\Delta^2/C_\infty^\star$. Here we use the fact that each arm in the path are fully dependent Bernoulli random variables, $\mathrm{KL}\left(\mathrm{Bern}(p)\|\mathrm{Bern}(q)\right) \leq \frac{(p-q)^2}{q(1-q)}$ and that $\Delta \in [0, \frac{1}{4}]$. By taking $\Delta = \min\left(\frac{1}{4}, \sqrt{\frac{C_\infty^\star}{20n}}\right)$ concludes Theorem 2. ∎

## F. Proof for the Application of Offline Learning for Cascading Bandits

*Proof of Corollary 1.* For the cascading bandit application, we need to prove how it satisfies the monotonicity condition (Condition 1), the 1-norm TPM condition (Condition 2), the 1-norm data coverage condition (Condition 4), and then settle down the corresponding smoothness factor $B_1$, data coverage coefficient $C_1^*$, action size $\bar{K}^*$.

---

**Algorithm 4** `CLCB-Cascade`: Combinatorial Lower Confidence Bound Algorithm for Cascading Bandits

---

1: **Input:** Dataset $\mathcal{D} = \{(S_t, \tau_t, (X_{t,i})_{i\in\tau_t})\}_{t=1}^n$, cardinality $k > 0$, solver `Top-k`, probability $\delta$.
2: **for** arm $i \in [m]$ **do**
3:     Calculate counter $N_i = \sum_{t=1}^n \mathbb{I}\{i \in \tau_t\}$;
4:     Calculate empirical mean $\hat{\mu}_i = \frac{\sum_{t=1}^n \mathbb{I}\{i\in\tau_t\}X_{t,i}}{N_i}$;
5:     Calculate LCB $\underline{\mu}_i = \hat{\mu}_i - \sqrt{\frac{\log(\frac{4mn}{\delta})}{2N_i}}$.
6: **end for**
7: Call oracle $\hat{S} = \text{Top-k}(\underline{\mu}_1, ..., \underline{\mu}_m)$.
8: **Return:** $\hat{S}$.

---

For the monotonicity condition (Condition 1), the 1-norm TPM condition (Condition 2), Lemma 1 in Wang & Chen (2017) yields $B_1 = 1$.

For the 1-norm data coverage condition (Condition 4), recall that we assume the arm means are in descending order $\mu_1 \geq \mu_2 \geq ... \geq \mu_m$, therefore we have $S^* = (1, 2, ..., k)$, and

$$p_i^{\mathbb{D}_{\text{arm}}, S^*} = \begin{cases} \prod_{j=1}^{i-1}(1-\mu_j), & \text{if } i \leq k, \\ 0, & \text{else if } i \geq k+1. \end{cases} \tag{45}$$

As for $p_i^{\mathbb{D}_{\text{arm}}, \mathbb{D}_\mathcal{S}}$, we have

$$p_i^{\mathbb{D}_{\text{arm}}, \mathbb{D}_\mathcal{S}} \geq \sum_{j=1}^k q_{ij}(1-\mu_1)^{j-1}, \tag{46}$$

where $q_{ij}$ is the probability that arm $i$ is sampled at the $j$-th position of the random ranked list sampled by the experimenter. By math calculation, we have

$$C_1^* = \sum_{i\in[m]} \frac{p_i^{\mathbb{D}_{\text{arm}}, S^*}}{p_i^{\mathbb{D}_{\text{arm}}, \mathbb{D}_\mathcal{S}}} \leq \sum_{i=1}^k \frac{\prod_{j=1}^{i-1}(1-\mu_j)}{\sum_{j=1}^k q_{ij}(1-\mu_1)^{j-1}} \tag{47}$$

and

$$\bar{K}^* = \sum_{i\in[m]} p_i^{\mathbb{D}_{\text{arm}}, S^*} \leq \sum_{i\in[k]} 1 = k \tag{48}$$

Plugging $B_1 = 1, C_1^* = \sum_{i=1}^{k} \frac{\prod_{j=1}^{i-1}(1-\mu_j)}{\sum_{j=1}^{k} q_{ij}(1-\mu_1)^{j-1}}, \bar{K}^* = k$ into our general result Theorem 1 yields the general result of Corollary 1.

When we assume that the data collecting distribution $\mathbb{D}_\mathcal{S}$ follows the *uniform* distribution from all possible ordered lists $\mathcal{S} = \{(a_1, ..., a_k) : a_i \in [m]$ for all $i \in [m]$, and $a_i \neq a_j$ for all $i \neq j\}$. Then we have $q_{i,j} = \frac{1}{m}$, and using Eq. (46) we have

$$p_i^{\mathbb{D}_{arm}, \mathbb{D}_\mathcal{S}} \geq \sum_{j=1}^{k} \frac{(1-\mu_1)^{j-1}}{m} = \frac{1 - (1-\mu_1)^k}{\mu_1 \cdot m} \tag{49}$$

We can use Eq. (47) and Eq. (45) to bound

$$C_1^* = \sum_{i \in [m]} \frac{p_i^{\mathbb{D}_{arm}, S^*}}{p_i^{\mathbb{D}_{arm}, \mathbb{D}_\mathcal{S}}} \leq \sum_{i \in [m]} \frac{p_i^{\mathbb{D}_{arm}, S^*}}{\frac{1-(1-\mu_1)^k}{\mu_1 \cdot m}} \tag{50}$$

$$\leq \frac{\sum_{j=1}^{k}(1-\mu_k)^j}{\frac{1-(1-\mu_1)^k}{\mu_1 \cdot m}} \tag{51}$$

$$= \frac{\frac{1-(1-\mu_k)^k}{\mu_k}}{\frac{1-(1-\mu_1)^k}{\mu_1 \cdot m}} \tag{52}$$

$$\leq \frac{\mu_1 \cdot m}{\mu_k}. \tag{53}$$

Plugging $B_1 = 1, C_1^* = \frac{\mu_1 \cdot m}{\mu_k}, \bar{K}^* = k$ into our general result Theorem 1 concludes Corollary 1.

$\blacksquare$

# G. Algorithm and Proof for the LLM Cache Application

### G.1. Offline Learning for the LLM Cache under the Standard CMAB-T View

---
**Algorithm 5** `CLCB-LLM-C`: Combinatorial Lower Confidence Bound Algorithm for LLM Cache
---
1: **Input:** Dataset $\mathcal{D} = \{(\mathcal{M}_t, q_t, c_t)\}_{t=1}^{n}$, queries $\mathcal{Q}$, solver `Top-k`, probability $\delta$.
2: **for** arm $q \in \mathcal{Q}$ **do**
3:     Calculate counter $N(q) = \sum_{t=1}^{n} \mathbb{I}\{q = q_t\}$ and $N_c(q) = \sum_{t=1}^{n} \mathbb{I}\{q = q_t$ and $q_t \notin \mathcal{M}_t\}$;
4:     Calculate empirical probability $\hat{p}(q) = N(q)/n, \hat{c}(q) = \sum_{t \in [n]} \mathbb{I}\{q = q_t$ and $q_t \notin \mathcal{M}_t\}c_t/N_c(q)$;
5:     Calculate UCB of the cost $\bar{c}(q) = \hat{c}(q) + \sqrt{\frac{\log(\frac{6mn}{\delta})}{2N_c(q)}}$, and UCB of the arrival probability $\bar{p}(q) = \hat{p}(q) + \sqrt{\frac{\log(\frac{6mn}{\delta})}{2n}}$.
6: **end for**
7: Call $\hat{\mathcal{M}} = $ `Top-k` $(\bar{p}(q_1)\bar{c}(q_1), ..., \bar{p}(q_m)\bar{c}(q_m))$.
8: **Return:** $\hat{\mathcal{M}}$.
---

For this LLM cache problem, we first show the corresponding base arms, super arm, and triggering probability. Then we prove this problem satisfies the 1-norm TPM smoothness condition (Condition 2) and 1-norm TPM data coverage condition (Condition 4). Finally, we give the upper bound result by using Theorem 1.

From the CMAB-T point of view, we have $2m$ base arms: the first $m$ arms correspond to the unknown costs $c(q) \in [0, 1]$ for $q \in \mathcal{Q}$, and the last $m$ arms corresponds to the arrival probability $p(q) \in [0, 1]$ for $q \in \mathcal{Q}$.

Let us denote $\boldsymbol{c} = (c(q))_{q \in \mathcal{Q}}$ and $\boldsymbol{p} = (p(q))_{q \in \mathcal{Q}}$ for convenience.

Recall that we treat the queries $S \in \mathcal{S}$ outside the cache $\mathcal{M}$ as the *super arm*, where $\mathcal{S} = \{\mathcal{Q} - \mathcal{M} : \mathcal{M} \subseteq \mathcal{Q}, |\mathcal{M}| \leq k\}$.

We can write the expected cost for each super arm $S \in \mathcal{S}$ as $c(S; \boldsymbol{c}, \boldsymbol{p}) = \sum_{q \in S} p(q)c(q)$.

Then we know that $S^* = \operatorname{argmax}_{S \in \mathcal{S}} c(S; \boldsymbol{c}, \boldsymbol{p})$, which contains the top $m - k$ queries regarding $p(q)c(q)$.

For the triggering probability, we have that, for any $S \in \mathcal{S}$, the triggering probability for unknown costs $p_{q,c}^{\mathbb{D}_{\text{arm}},S} = p(q)$ for $q \in S$ and 0 otherwise. The triggering probability for unknown arrival probability $p_{q,p}^{\mathbb{D}_{\text{arm}},S} = 1$ for all $q \in \mathcal{Q}$.

Now we can prove that this problem satisfies the 1-norm TPM smoothness condition (Condition 2) with $B_1 = 1$. That is, for any $S \in \mathcal{S}$, any $\boldsymbol{p}, \boldsymbol{p}', \boldsymbol{c}, \boldsymbol{c}' \in [0,1]^m$, we have

$$|c(S; \boldsymbol{c}, \boldsymbol{p}) - c(S; \boldsymbol{c}', \boldsymbol{p}')| = |c(S; \boldsymbol{c}, \boldsymbol{p}) - c(S; \boldsymbol{c}', \boldsymbol{p}) + c(S; \boldsymbol{c}', \boldsymbol{p}) - c(S; \boldsymbol{c}', \boldsymbol{p}')| \tag{54}$$

$$\leq |c(S; \boldsymbol{c}, \boldsymbol{p}) - c(S; \boldsymbol{c}', \boldsymbol{p})| + |c(S; \boldsymbol{c}', \boldsymbol{p}) - c(S; \boldsymbol{c}', \boldsymbol{p}')| \tag{55}$$

$$= \left| \sum_{q \in S} p(q)c(q) - p(q)c'(q) \right| + \left| \sum_{q \in S} p(q)c'(q) - p'(q)c'(q) \right| \tag{56}$$

$$\leq \sum_{q \in S} p(q) |c(q) - c'(q)| + \sum_{q \in S} c'(q) |p(q) - p'(q)| \tag{57}$$

$$\leq \sum_{q \in S} p(q) |c(q) - c'(q)| + \sum_{q \in \mathcal{Q}} |p(q) - p'(q)| \tag{58}$$

Next, we prove that this problem satisfies the 1-norm TPM data coverage condition (Condition 3).

Let $\nu(q) = \Pr_{\mathcal{M} \sim \mathbb{D}_{\mathcal{S}}}[q \notin \mathcal{M}]$ be the probability that $q$ is not sampled in the experimenter's cache $\mathcal{M} \sim \mathbb{D}_{\mathcal{S}}$. Then the data collecting probability for unknown costs $p_{q,c}^{\mathbb{D}_{\text{arm}}, \mathbb{D}_{\mathcal{S}}} = p(q)\nu(q)$ and $p_{q,p}^{\mathbb{D}_{\text{arm}}, \mathbb{D}_{\mathcal{S}}} = 1$ for unknown arrival probability, for $q \in \mathcal{Q}$. We can prove that the LLM cache satisfies Condition 4 by

$$\sum_{q \in \mathcal{Q}} \left( \frac{p_{q,c}^{\mathbb{D}_{\text{arm}}, S^*}}{p_{q,c}^{\mathbb{D}_{\text{arm}}, \mathbb{D}_{\mathcal{S}}}} + \frac{p_{q,p}^{\mathbb{D}_{\text{arm}}, S^*}}{p_{q,p}^{\mathbb{D}_{\text{arm}}, \mathbb{D}_{\mathcal{S}}}} \right) = \sum_{q \in \mathcal{Q}} \left( \frac{p(q)\mathbb{I}\{q \in S^*\}}{p(q)\nu(q)} + 1 \right) \leq \sum_{q \in S^*} \frac{1}{\nu(q)} + m = C_1^*. \tag{59}$$

Finally, we have $\bar{K}^* = \sum_{q \in \mathcal{Q}} \left( p_{q,c}^{\mathbb{D}_{\text{arm}}, S^*} + p_{q,p}^{\mathbb{D}_{\text{arm}}, S^*} \right) = \sum_{q \in \mathcal{Q}} p(q)\mathbb{I}\{q \in S^*\} + \sum_{q \in \mathcal{Q}} 1 \leq 1 + m$.

Plugging into Theorem 1 with $B_1 = 1$, $C_1^* = \sum_{q \in S^*} \frac{1}{\nu(q)} + m$, $\bar{K}^* = 1 + m$, we have the following suboptimality upper bound.

**Lemma 1** (Standard Upper Bound for LLM Cache). *For LLM cache bandit with a dataset $\mathcal{D}$ of $n$ data samples, let $\mathcal{M}^*$ be the optimal cache and suppose $n \geq \frac{8 \log(\frac{1}{\delta})}{\min_{q \in \mathcal{Q} - \mathcal{M}^*} p(q)\nu(q)}$, where $\nu(q)$ is the probability that query $q$ is not included in each offline sampled cache. Let $\hat{M}$ be the cache returned by algorithm Algorithm 5, then it holds with probability at least $1 - \delta$ that*

$$\text{SubOpt}(\hat{\mathcal{M}}; \alpha, \boldsymbol{c}, \boldsymbol{p}) := c(\mathcal{M}^*; \boldsymbol{c}, \boldsymbol{p}) - c\left(\hat{\mathcal{M}}; \boldsymbol{c}, \boldsymbol{p}\right) \tag{60}$$

$$\leq 2\sqrt{\frac{(m+1)\left(\sum_{q \in \mathcal{Q} - \mathcal{M}^*} \frac{1}{\nu(q)} + m\right) \log(\frac{6mn}{\delta})}{n}}, \tag{61}$$

*if the experimenter samples empty cache in each round as in (Zhu et al., 2023) so that $\nu(q) = 1$, it holds that $C_1^* \leq 2m$ and*

$$\text{SubOpt}(\hat{\mathcal{M}}; \alpha, \boldsymbol{\mu}) := c(\mathcal{M}^*; \boldsymbol{p}, \boldsymbol{c}) - c\left(\hat{\mathcal{M}}; \boldsymbol{p}, \boldsymbol{c}\right) \tag{62}$$

$$\leq 2\sqrt{\frac{2m(m+1) \log(\frac{6mn}{\delta})}{n}}. \tag{63}$$

## G.2. Improved Offline Learning for the LLM Cache by Leveraging the Full-feedback Property and the Vector-valued Concentration Inequality

**Theorem 6** (Improved Upper Bound for LLM Cache). *For LLM cache bandit with a dataset $\mathcal{D}$ of $n$ data samples, let $\mathcal{M}^*$ be the optimal cache and suppose $n \geq \frac{8 \log(\frac{1}{\delta})}{\min_{q \in \mathcal{Q} - \mathcal{M}^*} p(q)\nu(q)}$, where $\nu(q)$ is the probability that query $q$ is not included in*

*each offline sampled cache. Let $\hat{M}$ be the cache returned by algorithm Algorithm 5, then it holds with probability at least $1 - \delta$ that*

$$\text{SubOpt}(\hat{\mathcal{M}}; \alpha, \boldsymbol{\mu}) := c(\mathcal{M}^*; \boldsymbol{p}, \boldsymbol{c}) - c\left(\hat{\mathcal{M}}; \boldsymbol{p}, \boldsymbol{c}\right) \tag{64}$$

$$\leq 2\sqrt{\frac{2 \sum_{q \in \mathcal{Q} - \mathcal{M}^*} \frac{1}{\nu(q)} \log(\frac{6mn}{\delta})}{n}} + 2\sqrt{\frac{2m \log(\frac{3}{\delta})}{n}}, \tag{65}$$

*if the experimenter samples empty cache in each round as in (Zhu et al., 2023) so that $\nu(q) = 1$, it holds that $C_1^* \leq m$ and*

$$\text{SubOpt}(\hat{\mathcal{M}}; \alpha, \boldsymbol{\mu}) := c(\mathcal{M}^*; \boldsymbol{p}, \boldsymbol{c}) - c\left(\hat{\mathcal{M}}; \boldsymbol{p}, \boldsymbol{c}\right) \tag{66}$$

$$\leq 4\sqrt{\frac{2m \log(\frac{6mn}{\delta})}{n}}. \tag{67}$$

In this section, we use an improved CMAB-T view by clustering $m$ arrival probabilities $p(q)$ as a vector-valued arm, which is fully observed in each data sample (observing $q$ means observing one hot vector $\boldsymbol{e}_q \in \{0, 1\}^m$ with 1 at the $q$-th entry and 0 elsewhere).

Specifically, we have $m + 1$ base arms: the first $m$ arms correspond to the unknown costs $c(q)$ for $q \in \mathcal{Q}$, and the last (vector-valued) arm corresponds to the arrival probability vector $(p(q))_{q \in \mathcal{Q}}$.

Let us denote $\boldsymbol{c} = (c(q))_{q \in \mathcal{Q}}$ and $\boldsymbol{p} = (p(q))_{q \in \mathcal{Q}}$ for convenience.

For the triggering probability, for any action $S$, we only consider unknown costs $p_{q,c}^{\mathbb{D}_{\text{arm}}, S} = p(q)$ for $q \in S$ and 0 otherwise.

For the 1-norm TPM smoothness condition (Condition 2), directly following Eq. (58), we have that for any $S \in \mathcal{S}$, any $\boldsymbol{p}, \boldsymbol{p}', \boldsymbol{c}, \boldsymbol{c}' \in [0, 1]^m$,

$$|c(S; \boldsymbol{c}, \boldsymbol{p}) - c(S; \boldsymbol{c}', \boldsymbol{p}')| \leq \sum_{q \in S} p(q) |c(q) - c'(q)| + \sum_{q \in \mathcal{Q}} |p(q) - p'(q)| = \sum_{q \in S} p(q) |c(q) - c'(q)| + \|\boldsymbol{p} - \boldsymbol{p}'\|_1 \tag{68}$$

Recall that the empirical arrival probability vector is $\hat{\boldsymbol{p}}$ and the UCB of the cost is $\bar{\boldsymbol{c}}$.

Recall that $\hat{\mathcal{M}} = \arg\max_{|\mathcal{M}| = k} \sum_{q \in \mathcal{M}} \hat{p}(q) \bar{c}(q)$ given by line 7 in Algorithm 2, which from the CMAB-T view, corresponds to the super arm $\hat{S} = \mathcal{Q} - \hat{\mathcal{M}} = \arg\min_{S \in \mathcal{S}} c(S; \bar{\boldsymbol{c}}, \hat{\boldsymbol{p}})$.

Since we treat $\boldsymbol{p}$ as a single vector-valued base arm and consider the cost function (and minimizing the cost) instead of the reward function (and maximizing the reward), we need a slight adaptation of the proof of Eq. (17) as follows:

Recall that the dataset $\mathcal{D} = \{(\mathcal{M}_t, q_t, c_t)\}_{t=1}^n$.

Recall that $N_c(q) = \sum_{t=1}^n \mathbb{I}\{q = q_t \text{ and } q_t \notin \mathcal{M}_t\}$ is the number of times that $q$ is not in cache $\mathcal{M}_t$.

Recall that $\nu(q)$ is the probability that query $q$ is not included in each offline sampled cache $\mathcal{M}_t$.

Let $p^* = \min_{q \in \mathcal{Q} - \mathcal{M}^*} p(q) \nu(q)$ be the minimum data collecting probability.

First, we need a new concentration event for the vector-valued $\mathcal{E}_{arv}$ and two previous events as follows.

$$\mathcal{E}_{\text{arv}} := \left\{ \|\hat{\boldsymbol{p}} - \boldsymbol{p}\|_1 \leq \sqrt{\frac{2m \log(\frac{2}{\delta})}{n}} \right\} \tag{69}$$

$$\mathcal{E}_{\text{arm}} := \left\{ |\hat{c}(q) - c(q)| \leq \sqrt{\frac{\log(\frac{2mn}{\delta})}{2N_c(q)}} \text{ for any } q \in \mathcal{Q} \right\} \tag{70}$$

$$\mathcal{E}_{\text{counter}} := \left\{ N_c(q) \geq \frac{n \cdot p_{q,c}^{\mathbb{D}_{\text{arm}}, \mathbb{D}_{\mathcal{S}}}}{2} \text{ for any } q \in \mathcal{Q} - \mathcal{M}^* \,\middle|\, n \geq \frac{8 \log \frac{m}{\delta}}{p^*} \right\} \tag{71}$$

Following the derivation of Eq. (17), we have:

$$c(\hat{S}; \boldsymbol{c}, \boldsymbol{p}) - c(S^*; \boldsymbol{c}, \boldsymbol{p}) \tag{72}$$

$$\overset{(a)}{=} \underbrace{c(S^*; \bar{\boldsymbol{c}}, \hat{\boldsymbol{p}}) - c(S^*; \boldsymbol{c}, \boldsymbol{p})}_{\text{uncertainty gap}} + \underbrace{c\left(\hat{S}; \bar{\boldsymbol{c}}, \hat{\boldsymbol{p}}\right) - c(S^*; \bar{\boldsymbol{c}}, \hat{\boldsymbol{p}})}_{\text{oracle gap}} + \underbrace{c\left(\hat{S}; \boldsymbol{c}, \boldsymbol{p}\right) - c\left(\hat{S}; \bar{\boldsymbol{c}}, \hat{\boldsymbol{p}}\right)}_{\text{pessimism gap}} \tag{73}$$

$$\overset{(b)}{\leq} c(S^*; \bar{\boldsymbol{c}}, \hat{\boldsymbol{p}}) - c(S^*; \boldsymbol{c}, \boldsymbol{p}) + c\left(\hat{S}; \boldsymbol{c}, \boldsymbol{p}\right) - c\left(\hat{S}; \bar{\boldsymbol{c}}, \hat{\boldsymbol{p}}\right) \tag{74}$$

$$\overset{(c)}{\leq} c(S^*; \bar{\boldsymbol{c}}, \hat{\boldsymbol{p}}) - c(S^*; \boldsymbol{c}, \boldsymbol{p}) + c\left(\hat{S}; \bar{\boldsymbol{c}}, \boldsymbol{p}\right) - c\left(\hat{S}; \bar{\boldsymbol{c}}, \hat{\boldsymbol{p}}\right) \tag{75}$$

$$\overset{(d)}{\leq} c(S^*; \bar{\boldsymbol{c}}, \hat{\boldsymbol{p}}) - c(S^*; \boldsymbol{c}, \boldsymbol{p}) + \|\hat{\boldsymbol{p}} - \boldsymbol{p}\|_1 \tag{76}$$

$$\overset{(e)}{\leq} \sum_{q \in S^*} p(q) |\bar{c}(q) - c(q)| + 2 \|\hat{\boldsymbol{p}} - \boldsymbol{p}\|_1 \tag{77}$$

$$\overset{(f)}{\leq} \sum_{q \in S^*} p(q) |\bar{c}(q) - c(q)| + 2 \sqrt{\frac{2m \log(\frac{1}{\delta})}{n}}, \tag{78}$$

where inequality (a) is due to adding and subtracting $c\left(S^*; \bar{\boldsymbol{c}}, \hat{\boldsymbol{p}}\right)$ and $c\left(\hat{S}; \bar{\boldsymbol{c}}, \hat{\boldsymbol{p}}\right)$, inequality (b) is due to oracle gap $\leq 0$ by line 7 of Algorithm 2, inequality (c) is due to the monotonicity, inequality (d) is due to Eq. (58), inequality (e) is also due to Eq. (58), inequality (f) is due to the event $\mathcal{E}_{arv}$.

Then for the first term of Eq. (78), we follow Eq. (28) to Eq. (31):

$$\sum_{q \in S^*} p(q) |\bar{c}(q) - c(q)| \leq 2\alpha B_1 \sqrt{\frac{2\bar{K}^* C_1^* \log(\frac{2mn}{\delta})}{n}} + 2\sqrt{\frac{2m \log(\frac{1}{\delta})}{n}} \tag{79}$$

$$\leq 2\alpha B_1 \sqrt{\frac{2\bar{K}^* C_1^* \log(\frac{2mn}{\delta})}{n}} \tag{80}$$

$$\leq 2\sqrt{\frac{2 \sum_{q \in \mathcal{Q} - \mathcal{M}^*} \frac{1}{\nu(q)} \log(\frac{2mn}{\delta})}{n}} \tag{81}$$

where the last inequality is plugging in $B_1 = 1$, $\alpha = 1$, $\bar{K}^* = \sum_{q \in \mathcal{Q}} p_{q,c}^{\mathbb{D}_{\text{arm}}, S^*} = \sum_{q \in \mathcal{Q}} p(q)\mathbb{I}\{q \in S^*\} \leq 1$, and $C_1^* = \sum_{q \in S^*} \frac{1}{\nu(q)}$.

Putting together Eq. (81) and Eq. (78), we have

$$c(\hat{S}; \boldsymbol{c}, \boldsymbol{p}) - c(S^*; \boldsymbol{c}, \boldsymbol{p}) \leq 2\sqrt{\frac{2 \sum_{q \in \mathcal{Q} - \mathcal{M}^*} \frac{1}{\nu(q)} \log(\frac{2mn}{\delta})}{n}} + 2\sqrt{\frac{2m \log(\frac{1}{\delta})}{n}} \tag{82}$$

Finally, by Lemma 5, Lemma 6, and Lemma 7 we can show that event $\mathcal{E}_{\text{arm}}$, $\mathcal{E}_{\text{counter}}$, $\mathcal{E}_{arv}$ all hold with probability at least with $1 - \delta$. Setting $\delta' = \frac{1}{3\delta}$ concludes the theorem.

### G.3. Online learning for LLM Cache

For the online setting, we consider a $T$-round online learning game between the environment and the learner. In each round $t$, there will be a query $q_t$ coming to the system. Our goal is to select a cache $\mathcal{M}_t$ (or equivalently the complement set $\mathcal{Q} - \mathcal{M}_t$ in each round $t \in [T]$) so as to minimize the regret:

$$\text{Reg}(T) = \sum_{t=1}^{T} \mathbb{E}\left[c(\mathcal{M}_t; \boldsymbol{c}, \boldsymbol{p}) - c(\mathcal{M}^*; \boldsymbol{c}, \boldsymbol{p})\right]. \tag{83}$$

Similar to Zhu et al. (2023), we consider the streaming setting where the cache of size $k$ is the only space we can save the query's response. That is, after we receive query $q_t$ each round, if the cache misses the current cache $\mathcal{M}_t$, then we can choose to update the cache $\mathcal{M}_t$ by adding the current query and response to the cache, and replacing the one of the existing cached items if the cache $\mathcal{M}_t$ is full. This means that the feasible set $\mathcal{Q}_{t+1}$ needs to be a subset of the $\mathcal{M}_t \cup q_t$, for $t \in [T]$.

For this setting, we propose the CUCB-LLM-S algorithm (Algorithm 6).

---

**Algorithm 6** `CUCB-LLM-S`: Combinatorial Upper Confidence Bound Algorithm for Online Streaming LLM Cache

1: **Input:** Queries $\mathcal{Q}$, cache size $k$, probability $\delta$.
2: **Initialize:** Counter, empirical mean, LCB for unknown costs $N_{c,0}(q) = 0, \hat{c}_0(q) = 0, \underline{c}_0(q) = 0$. Empirical mean for arrival probability $\hat{p}_0(q) = 0$, for all $q \in \mathcal{Q}$. Initial cache $\mathcal{M}_1 = \emptyset$.
3: **for** $t = 1, 2, ..., T$ **do**
4:     User $t$ arrives with query $q_t$.
5:     **if** $q_t \in \mathcal{M}_t$ **then**
6:         Incur cost $C_t = 0$ but does not receive any feedback.
7:         Update $\hat{p}_t(q_t) = \frac{(t-1) \cdot \hat{p}_{t-1}(q_t) + 1}{t}$, and $\hat{p}_t(q_t) = \frac{(t-1) \cdot \hat{p}_{t-1}(q_t) + 0}{t}$ for $q \neq q_t$.
8:         Keep $N_{c,t}(q) = N_{c,t-1}(q), \hat{c}_t(q) = \hat{c}_{t-1}(q)$ for $q \in \mathcal{Q}$.
9:         Keep $\mathcal{M}_{t+1} = \mathcal{M}_t$.
10:    **else**
11:        The Cache misses and the system pay random cost $C_t$ with mean $c(q_t)$ to compute the response of $q_t$.
12:        Update $\hat{p}_t(q_t) = \frac{(t-1) \cdot \hat{p}_{t-1}(q_t) + 1}{t}$, and $\hat{p}_t(q_t) = \frac{(t-1) \cdot \hat{p}_{t-1}(q_t) + 0}{t}$ for $q \neq q_t$.
13:        Update $N_{c,t}(q_t) = N_{c,t-1}(q_t) + 1, \hat{c}_t(q_t) = \frac{N_{c,t-1}(q_t) \cdot \hat{c}_{t-1}(q_t) + C_t}{N_{c,t-1}(q_t) + 1}$.
14:        Keep $N_{c,t}(q) = N_{c,t-1}(q), \hat{c}_t(q) = \hat{c}_{t-1}(q)$ for $q \neq q_t$.
15:        Compute $\underline{c}_t(q) = \max \left\{ \hat{c}_t(q) - \sqrt{\frac{\log(2mT^2)}{2N_{c,t}(q)}}, 0 \right\}$ for all $q \in \mathcal{Q}$.
16:        **if** $|\mathcal{M}_t| < k$ **then**
17:            Add $q_t$'s response into $\mathcal{M}_t$ so that $\mathcal{M}_{t+1} = \mathcal{M}_t \cup q_t$.
18:        **else if** $\min_{q \in \mathcal{M}_t} \hat{p}_t(q) \underline{c}_t(q) \leq \hat{p}_t(q_t) \underline{c}_t(q_t)$ **then**
19:            Replace $q_{t,\min}$'s response with $q_t$'s, i.e., $\mathcal{M}_{t+1} = \mathcal{M}_t - q_{t,\min} + q_t$, where $q_{t,\min} = \text{argmin}_{q \in \mathcal{M}_t} \hat{p}_t(q) \underline{c}_t(q)$.
20:        **else**
21:            Keep $\mathcal{M}_{t+1} = \mathcal{M}_t$.
22:        **end if**
23:    **end if**
24: **end for**

---

The key difference from the traditional CUCB algorithm, where any super arm $S \in \mathcal{S}$ can be selected, is that the feasible future cache $\mathcal{M}_{t+1}$ in round $t + 1$ is restricted to $\mathcal{M}_{t+1} \subseteq \mathcal{M}_t \bigcup q_t$, where $\mathcal{M}_t$ is the current cache and the query that comes to the system. This means that we cannot directly utilize the top-$k$ oracle as in line 7 of Algorithm 2 and other online CMAB-T works (Wang & Chen, 2017; Liu et al., 2023b) due to the restricted feasible action set. To tackle this challenge, we design a new streaming procedure (lines 16-22), which leverages the previous cache $\mathcal{M}_t$ and newly coming $q_t$ to get the top-$k$ queries regarding $\hat{p}_t(q) \underline{c}_t(q)$.

We can prove the following lemma:

**Lemma 2** (Streaming procedure yields the global top-$k$ queries)**.** *Let $\mathcal{M}_t$ be the cache selected by Algorithm 6 in each round, then we have $\mathcal{M}_t = \text{argmax}_{\mathcal{M} \subseteq \mathcal{Q}: |\mathcal{M}| \leq k} \sum_{q \in \mathcal{M}} \hat{p}_{t-1}(q) \underline{c}_{t-1}(q)$.*

**Proof.** We prove this lemma by induction.

Base case when $t = 1$:

Since $\underline{c}_0(q) = \hat{p}_0(q) = 0$ for any $q \in \mathcal{Q}$, we have $\mathcal{M}_1 = \text{argmax}_{\mathcal{M} \subseteq \mathcal{Q}: |\mathcal{M}| \leq k} \hat{p}_0(q) \underline{c}_0(q) = \emptyset$.

For $t \geq 2$:

Suppose $\mathcal{M}_t = \text{argmax}_{\mathcal{M} \subseteq \mathcal{Q}: |\mathcal{M}| \leq k} \sum_{q \in \mathcal{M}} \hat{p}_{t-1}(q) \underline{c}_{t-1}(q)$.

Then we prove that $\mathcal{M}_{t+1} = \text{argmax}_{\mathcal{M} \subseteq \mathcal{Q}: |\mathcal{M}| \leq k} \sum_{q \in \mathcal{M}} \hat{p}_t(q) \underline{c}_t(q)$ as follows:

Case 1 (line 5): If $q_t \in \mathcal{M}_t$, then $\underline{c}_t(q) = \underline{c}_{t-1}(q)$ remain unchanged for $q \in \mathcal{Q}$. For the arrival probability, $\hat{p}_t(q_t) \geq \hat{p}_{t-1}(q_t)$ is increased, and $\hat{p}_t(q) = \frac{(t-1)\cdot\hat{p}_{t-1}(q)}{t}$ are scaled with an equal ratio of $\frac{t-1}{t}$ for $q \neq q_t$. Therefore, the *relative order* of queries $q \in \mathcal{Q} - q_t$ remain unchanged regarding $\hat{p}_{t-1}(q)\underline{c}_{t-1}(q)$ and $\hat{p}_t(q)\underline{c}_t(q)$. Moreover, $\hat{p}_{t-1}(q_t)\underline{c}_{t-1}(q_t) \leq \hat{p}_t(q_t)\underline{c}_t(q_t)$ is increased while other queries are decreased, so $q_t$ remains in the top-$|\mathcal{M}_t|$ queries. Thus, $\mathcal{M}_{t+1} = \mathcal{M}_t$ remains the top-$|\mathcal{M}_t|$ queries.

Case 2 (line 16): If $q_t \notin \mathcal{M}_t$ and $|\mathcal{M}_t| < k$, then we know that all the queries $q \notin (\mathcal{M}_t + q_t)$ never arrives, and $\bar{c}_t(q) = 0$. Therefore, $\hat{p}_t(q_t)\underline{c}_t(q_t) \geq \hat{p}_t(q)\underline{c}_t(q) = 0$ for any $q \notin (\mathcal{M}_t + q_t)$, and $\mathcal{M}_t + q_t$ are top-$|\mathcal{M}_t + 1|$ queries.

Case 3 (line 18): If $q_t \notin \mathcal{M}_t$ and $|\mathcal{M}_t| = k$, then $\underline{c}_t(q) = \underline{c}_{t-1}(q)$ remain unchanged for $q \in \mathcal{Q} - q_t$ and $\hat{p}_t(q) = \frac{(t-1)\cdot\hat{p}_{t-1}(q)}{t}$ are scaled with an equal ratio of $\frac{t-1}{t}$ for $q \neq q_t$, so the *relative order* of queries $q \in \mathcal{Q} - q_t$ remain unchanged regarding $\hat{p}_{t-1}(q)\underline{c}_{t-1}(q)$ and $\hat{p}_t(q)\underline{c}_t(q)$. The only changed query is the $q_t$, so we only need to replace the minimum query $q_{t,\min} = \arg\min_{q\in\mathcal{M}_t} \hat{p}_t(q)\underline{c}_t(q)$ with $q_t$, if $\hat{p}_t(q_{t,\min})\underline{c}_t(q_{t,\min}) \leq \hat{p}_t(q_t)\underline{c}_t(q_t)$, which is exactly the line 19. This guarantees that $\mathcal{M}_{t+1}$ are top-$k$ queries regarding $\hat{p}_t(q)\underline{c}_t(q)$, concluding our induction. $\blacksquare$

Now we go back to the CMAB-T view by using $S_t = \mathcal{Q} - \mathcal{M}_t$, and by the above Lemma 2, we have $S_t = \arg\min_{S\subseteq\mathcal{Q}:|S|\geq k} \sum_{q\in S} \hat{p}_{t-1}(q)\underline{c}_{t-1}(q)$.

Then we have the following theorem.

**Theorem 7.** *For the online streaming LLM cache problem, the regret of Algorithm 6 is upper bounded by* $O\left(\sqrt{mT\log(\frac{mT}{\delta})}\right)$ *with probability at least* $1 - \delta$.

**Proof.** We also define two high-probability events:

$$\mathcal{E}_{\mathrm{arv}} := \left\{ \|\hat{\boldsymbol{p}}_t - \boldsymbol{p}\|_1 \leq \sqrt{\frac{2m\log(\frac{2T}{\delta})}{t}} \text{ for any } t \in [T] \right\} \tag{84}$$

$$\mathcal{E}_{\mathrm{arm}} := \left\{ |\hat{c}_t(q) - c(q)| \leq \sqrt{\frac{\log(\frac{2mT^2}{\delta})}{2N_{c,t}(q)}} \text{ for any } q \in \mathcal{Q}, t \in [T] \right\} \tag{85}$$

Now we can have the following regret decomposition under $\mathcal{E}_{\mathrm{arv}}$ and $\mathcal{E}_{\mathrm{arm}}$:

$$\mathrm{Reg}(T) = \mathbb{E}\left[\sum_{t=1}^T (c(S_t; \boldsymbol{c}, \boldsymbol{p}) - c(S^*; \boldsymbol{c}, \boldsymbol{p}))\right] \tag{86}$$

$$\overset{(a)}{=} \mathbb{E}\left[\sum_{t=1}^T \left( \underbrace{c(S_t; \boldsymbol{c}, \boldsymbol{p}) - c(S_t; \underline{\boldsymbol{c}}_{t-1}, \hat{\boldsymbol{p}}_{t-1})}_{\text{uncertainty gap}} \right.\right.$$

$$\left.\left. + \underbrace{c(S_t; \underline{\boldsymbol{c}}_{t-1}, \hat{\boldsymbol{p}}_{t-1}) - c(S^*; \underline{\boldsymbol{c}}_{t-1}, \hat{\boldsymbol{p}}_{t-1})}_{\text{oracle gap}} + \underbrace{c(S^*; \underline{\boldsymbol{c}}_{t-1}, \hat{\boldsymbol{p}}_{t-1}) - c(S^*; \boldsymbol{c}, \boldsymbol{p})}_{\text{optimistic gap}} \right)\right] \tag{87}$$

$$\overset{(b)}{\leq} \mathbb{E}\left[\sum_{t=1}^T (c(S_t; \boldsymbol{c}, \boldsymbol{p}) - c(S_t; \underline{\boldsymbol{c}}_{t-1}, \hat{\boldsymbol{p}}_{t-1}) + c(S^*; \underline{\boldsymbol{c}}_{t-1}, \hat{\boldsymbol{p}}_{t-1}) - c(S^*; \boldsymbol{c}, \boldsymbol{p}))\right] \tag{88}$$

$$\overset{(c)}{\leq} \mathbb{E}\left[\sum_{t=1}^T (c(S_t; \boldsymbol{c}, \boldsymbol{p}) - c(S_t; \underline{\boldsymbol{c}}_{t-1}, \hat{\boldsymbol{p}}_{t-1}) + c(S^*; \boldsymbol{c}, \hat{\boldsymbol{p}}_{t-1}) - c(S^*; \boldsymbol{c}, \boldsymbol{p}))\right] \tag{89}$$

$$\overset{(d)}{\leq} \mathbb{E}\left[\sum_{t=1}^T \left( c(S_t; \boldsymbol{c}, \boldsymbol{p}) - c(S_t; \underline{\boldsymbol{c}}_{t-1}, \hat{\boldsymbol{p}}_{t-1}) + \sqrt{\frac{2m\log(\frac{2T}{\delta})}{t}} \right)\right] \tag{90}$$

$$\overset{(e)}{\leq} \mathbb{E}\left[\sum_{t=1}^T \left( \sum_{q\in S_t} p(q)|\underline{c}_{t-1}(q) - c(q)| + 2\sqrt{\frac{2m\log(\frac{2T}{\delta})}{t}} \right)\right] \tag{91}$$

$$\stackrel{(f)}{\leq} \mathbb{E}\left[\sum_{t=1}^{T}\left(\sum_{q\in S_t} 2p(q)\sqrt{\frac{\log(\frac{2mT}{\delta})}{2N_{c,t-1}(q)}} + 2\sqrt{\frac{2m\log(\frac{2T}{\delta})}{t}}\right)\right] \tag{92}$$

$$\leq \mathbb{E}\left[\sum_{t=1}^{T}\sum_{q\in S_t} p(q)\sqrt{\frac{2\log(\frac{2mT}{\delta})}{N_{c,t-1}(q)}}\right] + 4\sqrt{2mT\log(\frac{2T}{\delta})} \tag{93}$$

$$\stackrel{(g)}{\leq} 14\sqrt{2m\bar{K}^*T\log(\frac{2mT}{\delta})} + 2m + 4\sqrt{2mT\log(\frac{2T}{\delta})} \tag{94}$$

$$\stackrel{(f)}{\leq} 18\sqrt{2mT\log(\frac{2mT}{\delta})} + 2m \tag{95}$$

where inequality (a) is due to adding and subtracting terms, inequality (b) is due to oracle gap $\leq 0$ by $S_t = \mathrm{argmin}_{S\subseteq\mathcal{Q}:|S|\geq k}\sum_{q\in S}\hat{p}_{t-1}(q)\mathfrak{c}_{t-1}(q)$ (indicated by Lemma 2), inequality (c) is due to the monotonicity, inequality (d) is due to Eq. (58) and event $\mathcal{E}_{arv}$, inequality (e) is also due to Eq. (58), inequality (f) is due to the event $\mathcal{E}_{arm}$, and inequality (g) is by the same derivation of Appendix C.1 starting from inequality (50) in Liu et al. (2023a) by recognizing $p_i^{D,S_t} = p(q), \bar{\mu}_{t,i} = \mathfrak{c}_{t-1}(q), \mu_i = c(q)$, inequality (f) is due to $\bar{K}^* = \sum_{q\in\mathcal{Q}} p_{q,c}^{\mathbb{D}_{arm},S^*} = \sum_{q\in\mathcal{Q}} p(q)\mathbb{I}\{q\in S^*\}\leq 1$. Letting $\delta = 1/T$ and considering the failure of the events $\mathcal{E}_{arv}, \mathcal{E}_{arm}$ that produces $O(1)$ regret concludes the theorem. ∎

## H. Proof for the Influence Maximization Application under the Node-level feedback

*Proof for Theorem 4.* Recall that the underlying graph is $G(\mathcal{V},\mathcal{E},p)$ and our offline dataset is $\mathcal{D} = \{(S_{t,0}, S_{t,1}, ..., S_{t,V-1})\}_{t=1}^{n}$.

For each node-level feedback data $t\in[n]$, recall that we only use the seed set $S_{t,0}$ and the active nodes in the first diffusion step $S_{t,1}$ to construct the LCB.

We use $q_v = \Pr[v\in S_{t,0}]$ to denote the probability that the node $v$ is selected by the experimenter in the seed set $S_{t,0}$.

We use $p(\bar{v}) := \Pr[v\notin S_{t,1}]$ to denote the probability that the node $v$ is not activated in one time step.

We use $p(\bar{v}|u) := \Pr[v\notin S_{t,1}|u\in S_{t,0}]$ and $p(\bar{v}|\bar{u}) := \Pr[v\notin S_{t,1}|u\notin S_{t,0}]$ to denote the probability that the node $v$ is not activated in one time step conditioned on whether the node $u$ is in the seed set $S_{t,0}$ or not, respectively.

Recall that we use the following notations to denote the set of counters, which are helpful in constructing the unbiased estimator and the high probability confidence interval of the above probabilities $q_v, p(\bar{v})$ and $p(\bar{v}|\bar{u})$:

$$n_{0,u} = |\{t\in[n]: u\in S_{t,0}\}|, \tag{96}$$

$$n_{0,\bar{u}} = |\{t\in[n]: u\notin S_{t,0}\}|, \tag{97}$$

$$n_{1,\bar{v}} = |\{t\in[n]: v\notin S_{t,1}\}|, \tag{98}$$

$$n_{1,\bar{u},\bar{v}} = |\{t\in[n]: u\notin S_{t,0} \text{ and } v\notin S_{t,1}\}| \tag{99}$$

Recall that for any $u,v\in\mathcal{V}$ and given probability $\delta$, we construct the UCB $\bar{q}_u, \bar{p}(\bar{v})$, and LCB $\underline{p}(\bar{v}|\bar{u})$ as follows.

$$\bar{q}_u = \min\{\hat{q}_u + \rho_u, 1\}, \tag{100}$$

$$\bar{p}(\bar{v}) = \min\{\hat{p}(\bar{v}) + \rho(\bar{v}), 1\}, \tag{101}$$

$$\underline{p}(\bar{v}|\bar{u}) = \max\{\hat{p}(\bar{v}|\bar{u}) - \rho(\bar{v}|\bar{u}), 0\} \tag{102}$$

where the unbiased estimators are:

$$\hat{q}_u = n_{0,u}/n, \tag{103}$$

$$\hat{p}(\bar{v}) = n_{1,\bar{v}}/n, \tag{104}$$

$$\hat{p}(\bar{v}|\bar{u}) = n_{1,\bar{u},\bar{v}}/n_{0,\bar{u}} \tag{105}$$

and the variance-adaptive confidence intervals are:

$$\rho_u = \sqrt{\frac{6(1-\hat{q}_u)\hat{q}_u\log(\frac{1}{\delta})}{n}} + \frac{9\log(\frac{1}{\delta})}{n}, \tag{106}$$

$$\rho(\bar{v}) = \sqrt{\frac{6(1 - \widehat{p}(\bar{v}))\widehat{p}(\bar{v})\log(\frac{1}{\delta})}{n}} + \frac{9\log(\frac{1}{\delta})}{n} \tag{107}$$

$$\rho(\bar{v}|\bar{u}) = \sqrt{\frac{6\left(1 - \widehat{p}(\bar{v}|\bar{u})\right)\widehat{p}(\bar{v}|\bar{u})\log(\frac{1}{\delta})}{n_{0,\bar{u}}}} + \frac{9\log(\frac{1}{\delta})}{n_{0,\bar{u}}} \tag{108}$$

Based on the above unbiased estimators and confidence intervals, we define the following events to bound the difference between the true parameter and their UCB/LCBs:

$$\mathcal{E}_{arm,1}(u) := \left\{ q_u \leq \bar{q}_u \leq \min\left\{ q_u + 4\sqrt{3}\sqrt{\frac{q_u(1 - q_u)\log(\frac{1}{\delta})}{n}} + 28 \cdot \frac{\log(\frac{1}{\delta})}{n}, 1 \right\} \right\} \tag{109}$$

$$\mathcal{E}_{arm,2}(\bar{v}) := \left\{ p(\bar{v}) \leq \bar{p}(\bar{v}) \leq \min\left\{ p(\bar{v}) + 4\sqrt{3}\sqrt{\frac{p(\bar{v})(1 - p(\bar{v}))\log(\frac{1}{\delta})}{n}} + 28 \cdot \frac{\log(\frac{1}{\delta})}{n}, 1 \right\} \right\} \tag{110}$$

$$\mathcal{E}_{arm,3}(\bar{u}, \bar{v}) := \left\{ \max\left\{ p(\bar{v}|\bar{u}) - 4\sqrt{3}\sqrt{\frac{p(\bar{v}|\bar{u})(1 - p(\bar{v}|\bar{u}))\log(\frac{1}{\delta})}{n_{0,\bar{u}}}} - 28 \cdot \frac{\log(\frac{1}{\delta})}{n_{0,\bar{u}}}, 0 \right\} \leq \underline{p}(\bar{v}|\bar{u}) \leq p(\bar{v}|\bar{u}) \right\} \tag{111}$$

$$\mathcal{E}_{counter}(u) := \left\{ n_{0,\bar{u}} \geq \frac{n(1 - q_u)}{2} \, \middle| \, n \geq \frac{8\log\frac{1}{\delta}}{1 - q_u} \right\}. \tag{112}$$

$$\mathcal{E}_{emp,1}(\bar{v}) := \left\{ \hat{p}(\bar{v}) \leq 2p(\bar{v}) \, \middle| \, n \geq \frac{8\log\frac{1}{\delta}}{p(\bar{v})} \right\} \tag{113}$$

$$\mathcal{E}_{emp,2}(\bar{u}, \bar{v}) := \left\{ \hat{p}(\bar{v}|\bar{u}) \geq p(\bar{v}|\bar{u})/2 \, \middle| \, n_{0,u} \geq \frac{8\log\frac{1}{\delta}}{p(\bar{v}|\bar{u})} \right\} \tag{114}$$

Recall that the relationship between $p_{uv}$ and $q_u, p(\bar{v})$ and $p(\bar{v}|\bar{u})$ is:

$$p_{uv} = \frac{1}{q_u}\left(1 - \frac{p(\bar{v})}{p(\bar{v}|\bar{u})}\right) \tag{115}$$

Then we construct the LCB $\underline{p}_{uv}$ based on the above intermediate UCB $\bar{q}_u, \bar{p}(\bar{v})$, and LCB $\underline{p}(\bar{v}|\bar{u})$:

$$\underline{p}_{uv} = \min\left\{ 1, \max\left\{ 0, \frac{1}{\bar{q}_u}\left(1 - \frac{\bar{p}(\bar{v})}{\underline{p}(\bar{v}|\bar{u})}\right) \right\} \right\}. \tag{116}$$

(1) It is obvious that $\underline{p}_{u,v}$ is a lower bound of $p_{uv}$, i.e., $\underline{p}_{u,v} \leq p_{uv}$, since $q_u \leq \bar{q}_u, p(\bar{v}) \leq \bar{p}(\bar{v}), \underline{p}(\bar{v}|\bar{u}) \leq p(\bar{v}|\bar{u})$ under event $\mathcal{E}_{arm,1}(u), \mathcal{E}_{arm,2}(\bar{v}), \mathcal{E}_{arm,3}(\bar{u}, \bar{v})$.

(2) Our next key step is to show that the difference between $\underline{p}_{u,v}$ and $p_{uv}$ is very small and decreases as the number of data samples $n$ increases:

Fix any two nodes $u, v$, we define two intermediate LCBs for $p_{uv}$ where only one parameter changes at a time:

$$\underline{p}_{1,uv} = \frac{1}{\bar{q}_u}\left(1 - \frac{p(\bar{v})}{p(\bar{v}|\bar{u})}\right) \tag{117}$$

$$\underline{p}_{2,uv} = \frac{1}{\bar{q}_u}\left(1 - \frac{\bar{p}(\bar{v})}{p(\bar{v}|\bar{u})}\right) \tag{118}$$

Suppose $\gamma \leq q_u \leq 1 - \gamma, p(\bar{v}) \geq \eta$, and $n \geq \frac{392\log(\frac{1}{\delta})}{\eta \cdot \gamma}$,

We can bound each term under event $\mathcal{E}_{arm,1}(u), \mathcal{E}_{arm,2}(\bar{v}), \mathcal{E}_{arm,3}(\bar{u}, \bar{v})$ by:

$$p_{uv} - \underline{p}_{1,uv} = \frac{1}{q_u}\left(1 - \frac{p(\bar{v})}{p(\bar{v}|\bar{u})}\right) - \frac{1}{\bar{q}_u}\left(1 - \frac{p(\bar{v})}{p(\bar{v}|\bar{u})}\right) \tag{119}$$

$$= \frac{\bar{q}_u - q_u}{\bar{q}_u q_u}\left(1 - \frac{p(\bar{v})}{p(\bar{v}|\bar{u})}\right) \tag{120}$$

$$\overset{(a)}{=} \frac{\bar{q}_u - q_u}{\bar{q}_u} \cdot p_{uv} \tag{121}$$

$$\overset{(b)}{\leq} \frac{4\sqrt{3}\sqrt{\frac{q_u \log(\frac{1}{\delta})}{n}} + 28\frac{\log(\frac{1}{\delta})}{n}}{q_u} \cdot p_{uv} \tag{122}$$

$$\overset{(c)}{\leq} \frac{8\sqrt{3}\sqrt{\frac{q_u \log(\frac{1}{\delta})}{n}}}{q_u} \cdot p_{uv} \tag{123}$$

$$= 8\sqrt{3}\sqrt{\frac{\log(\frac{1}{\delta})}{nq_u}} \cdot p_{uv} \tag{124}$$

$$\overset{(d)}{\leq} 8\sqrt{3}\sqrt{\frac{\log(\frac{1}{\delta})}{\gamma \cdot n}} \cdot p_{uv}, \tag{125}$$

where equality (a) is due to Eq. (115), inequality (b) is due to the event $\mathcal{E}_{arm,1}(u)$, inequality (c) is due to $28\frac{\log(\frac{1}{\delta})}{n} \leq 4\sqrt{3}\sqrt{\frac{q_u \log(\frac{1}{\delta})}{n}}$ when $n \geq \frac{392 \log(\frac{1}{\delta})}{\eta \cdot \gamma} > \frac{49}{3}\frac{\log(\frac{1}{\delta})}{q_u}$, inequality (d) is due to $q_u \geq \gamma$.

$$\underline{p}_{1,uv} - \underline{p}_{2,uv} = \frac{1}{\bar{q}_u}\left(1 - \frac{p(\bar{v})}{p(\bar{v}|\bar{u})}\right) - \frac{1}{\bar{q}_u}\left(1 - \frac{\bar{p}(\bar{v})}{p(\bar{v}|\bar{u})}\right) \tag{126}$$

$$= \frac{1}{\bar{q}_u}\left(\frac{\bar{p}(\bar{v}) - p(\bar{v})}{p(\bar{v}|\bar{u})}\right) \tag{127}$$

$$\overset{(a)}{\leq} \frac{1}{\gamma}\left(\frac{\bar{p}(\bar{v}) - p(\bar{v})}{p(\bar{v}|\bar{u})}\right) \tag{128}$$

$$\overset{(b)}{\leq} \frac{1}{\gamma}\frac{4\sqrt{3}\sqrt{\frac{p(\bar{v}) \log(\frac{1}{\delta})}{n}} + 28\frac{\log(\frac{1}{\delta})}{n}}{p(\bar{v}|\bar{u})} \tag{129}$$

$$\overset{(c)}{\leq} \frac{1}{\gamma}\frac{8\sqrt{3}\sqrt{\frac{p(\bar{v}) \log(\frac{1}{\delta})}{n}}}{p(\bar{v}|\bar{u})} \tag{130}$$

$$\overset{(d)}{\leq} \frac{8\sqrt{3}}{\gamma}\sqrt{\frac{\log(\frac{1}{\delta})}{p(\bar{v}) \cdot n}} \tag{131}$$

$$\overset{(e)}{\leq} \frac{8\sqrt{3}}{\gamma}\sqrt{\frac{\log(\frac{1}{\delta})}{\eta \cdot n}}, \tag{132}$$

where inequality (a) is due to $\bar{q}_t \geq q_u \geq \gamma$, inequality (b) is due to the event $\mathcal{E}_{arm,2}(\bar{v})$, inequality (c) is due to $28\frac{\log(\frac{1}{\delta})}{n} \leq 4\sqrt{3}\sqrt{\frac{p(\bar{v}) \log(\frac{1}{\delta})}{n}}$ when $n \geq \frac{392 \log(\frac{1}{\delta})}{\eta \cdot \gamma} > \frac{49}{3}\frac{\log(\frac{1}{\delta})}{p(\bar{v})}$, inequality (d) is due to $p(\bar{v}) \leq p(\bar{v}|\bar{u})$, inequality (e) is due to $p(\bar{v}) \geq \eta$.

Before we bound $\underline{p}_{2,uv} - p_{uv}$, we first show that $\underline{p}(\bar{v}|\bar{u}) \geq \frac{1}{2}p((\bar{v}|\bar{u})) > 0$ for any $(u,v) \in \mathcal{E}$. That is:

$$\underline{p}(\bar{v}|\bar{u}) \overset{(a)}{\geq} p(\bar{v}|\bar{u}) - 4\sqrt{3}\sqrt{\frac{p(\bar{v}|\bar{u}) \log(\frac{1}{\delta})}{n_{0,\bar{u}}}} - \frac{28 \log(\frac{1}{\delta})}{n_{0,\bar{u}}} \tag{133}$$

$$\overset{(b)}{\geq} p(\bar{v}|\bar{u}) - 8\sqrt{3}\sqrt{\frac{p(\bar{v}|\bar{u})\log(\frac{1}{\delta})}{n_{0,\bar{u}}}} \tag{134}$$

$$\overset{(c)}{\geq} \frac{p(\bar{v}|\bar{u})}{2} > 0 \tag{135}$$

where inequality (a) is due to event $\mathcal{E}_{arm,3}(\bar{u},\bar{v})$, inequality (b) is due to $4\sqrt{3}\sqrt{\frac{p(\bar{v}|\bar{u})\log(\frac{1}{\delta})}{n_{0,\bar{u}}}} \geq \frac{28\log(\frac{1}{\delta})}{n_{0,\bar{u}}}$ when $n_{0,u} > \frac{49}{3}\frac{\log(\frac{1}{\delta})}{p(\bar{v}|\bar{u})}$, which is guaranteed when $n \geq \frac{98\log(\frac{1}{\delta})}{3\eta\cdot\gamma}$ under the event $\mathcal{E}_{counter}(u)$ and $1-q_u \geq \gamma$ (i.e., $n_{0,u} \geq \frac{n\gamma}{2}$), inequality (c) is due to $4\sqrt{3}\sqrt{\frac{p(\bar{v}|\bar{u})\log(\frac{1}{\delta})}{n_{0,\bar{u}}}} \leq \frac{p(\bar{v}|\bar{u})}{2}$ when $n_{0,u} > \frac{196\log(\frac{1}{\delta})}{p(\bar{v}|\bar{u})}$, which is guaranteed when $n \geq \frac{392\log(\frac{1}{\delta})}{\eta\cdot\gamma}$ under the event $\mathcal{E}_{counter}(u)$ and $1-q_u \geq \gamma$ (i.e., $n_{0,u} \geq \frac{n\gamma}{2}$).

When $\min\left\{1, \max\left\{0, \frac{1}{\bar{q}_u}\left(1 - \frac{\bar{p}(\bar{v})}{\underline{p}(\bar{v}|\bar{u})}\right)\right\}\right\} = 1$, we have

$$\underline{p}_{2,uv} - \min\left\{1, \max\left\{0, \frac{1}{\bar{q}_u}\left(1 - \frac{\bar{p}(\bar{v})}{\underline{p}(\bar{v}|\bar{u})}\right)\right\}\right\} \leq 0. \tag{136}$$

When $\min\left\{1, \max\left\{0, \frac{1}{\bar{q}_u}\left(1 - \frac{\bar{p}(\bar{v})}{\underline{p}(\bar{v}|\bar{u})}\right)\right\}\right\} < 1$, we have

$$\underline{p}_{2,uv} - \min\left\{1, \max\left\{0, \frac{1}{\bar{q}_u}\left(1 - \frac{\bar{p}(\bar{v})}{\underline{p}(\bar{v}|\bar{u})}\right)\right\}\right\} \tag{137}$$

$$= \underline{p}_{2,uv} - \max\left\{0, \frac{1}{\bar{q}_u}\left(1 - \frac{\bar{p}(\bar{v})}{\underline{p}(\bar{v}|\bar{u})}\right)\right\} \tag{138}$$

$$\leq \underline{p}_{2,uv} - \frac{1}{\bar{q}_u}\left(1 - \frac{\bar{p}(\bar{v})}{\underline{p}(\bar{v}|\bar{u})}\right) \tag{139}$$

$$= \frac{1}{\bar{q}_u}\left(1 - \frac{\bar{p}(\bar{v})}{p(\bar{v}|\bar{u})}\right) - \frac{1}{\bar{q}_u}\left(1 - \frac{\bar{p}(\bar{v})}{\underline{p}(\bar{v}|\bar{u})}\right) \tag{140}$$

$$= \frac{\bar{p}(\bar{v})}{\bar{q}_u}\left(\frac{p(\bar{v}|\bar{u}) - \underline{p}(\bar{v}|\bar{u})}{\underline{p}(\bar{v}|\bar{u})p(\bar{v}|\bar{u})}\right) \tag{141}$$

$$\overset{(a)}{\leq} \frac{\bar{p}(\bar{v})}{\bar{q}_u}\left(\frac{4\sqrt{3}\sqrt{\frac{p(\bar{v}|\bar{u})\log(\frac{1}{\delta})}{n_{0,\bar{u}}}} + \frac{28\log(\frac{1}{\delta})}{n_{0,\bar{u}}}}{\underline{p}(\bar{v}|\bar{u})p(\bar{v}|\bar{u})}\right) \tag{142}$$

$$\overset{(b)}{\leq} \frac{\bar{p}(\bar{v})}{\bar{q}_u}\left(\frac{8\sqrt{3}\sqrt{\frac{p(\bar{v}|\bar{u})\log(\frac{1}{\delta})}{n_{0,\bar{u}}}}}{\underline{p}(\bar{v}|\bar{u})p(\bar{v}|\bar{u})}\right) \tag{143}$$

$$\overset{(c)}{\leq} \frac{\bar{p}(\bar{v})}{\bar{q}_u p(\bar{v}|\bar{u})}\left(\frac{8\sqrt{3}\sqrt{\frac{p(\bar{v}|\bar{u})\log(\frac{1}{\delta})}{n_{0,\bar{u}}}}}{\frac{1}{2}p(\bar{v}|\bar{u})}\right) \tag{144}$$

$$\overset{(d)}{\leq} \frac{p(\bar{v}) + 4\sqrt{3}\sqrt{\frac{p(\bar{v})\log(\frac{1}{\delta})}{n}} + 28\frac{\log(\frac{1}{\delta})}{n}}{\bar{q}_u p(\bar{v}|\bar{u})}\left(\frac{8\sqrt{3}\sqrt{\frac{p(\bar{v}|\bar{u})\log(\frac{1}{\delta})}{n_{0,\bar{u}}}}}{\frac{1}{2}p(\bar{v}|\bar{u})}\right) \tag{145}$$

$$\overset{(e)}{\leq} \frac{p(\bar{v}) + 8\sqrt{3}\sqrt{\frac{p(\bar{v})\log(\frac{1}{\delta})}{n}}}{\bar{q}_u p(\bar{v}|\bar{u})}\left(\frac{8\sqrt{3}\sqrt{\frac{p(\bar{v}|\bar{u})\log(\frac{1}{\delta})}{n_{0,\bar{u}}}}}{\frac{1}{2}p(\bar{v}|\bar{u})}\right) \tag{146}$$

$$\overset{(f)}{\leq} \frac{2p(\bar{v})}{\bar{q}_u p(\bar{v}|\bar{u})} \left( \frac{8\sqrt{3}\sqrt{\frac{p(\bar{v}|\bar{u})\log(\frac{1}{\delta})}{n_{0,\bar{u}}}}}{\frac{1}{2}p(\bar{v}|\bar{u})} \right) \tag{147}$$

$$= \frac{32\sqrt{3} \cdot p(\bar{v})}{\bar{q}_u \cdot p(\bar{v}|\bar{u})} \sqrt{\frac{\log(\frac{1}{\delta})}{p(\bar{v}|\bar{u}) \cdot n_{0,\bar{u}}}} \tag{148}$$

$$\overset{(g)}{\leq} \frac{32\sqrt{3}}{\bar{q}_u} \sqrt{\frac{\log(\frac{1}{\delta})}{p(\bar{v}) \cdot n_{0,\bar{u}}}} \tag{149}$$

$$\overset{(h)}{\leq} \frac{32\sqrt{6}}{\bar{q}_u} \sqrt{\frac{\log(\frac{1}{\delta})}{\eta \cdot n \cdot (1 - q_u)}} \tag{150}$$

$$\overset{(i)}{\leq} 32\sqrt{6}\sqrt{\frac{\log(\frac{1}{\delta})}{\eta \cdot \gamma^3 \cdot n}} \tag{151}$$

where inequality (a) is due to event $\mathcal{E}_{arm,3}(\bar{u}, \bar{v})$, inequality (b) is due to $4\sqrt{3}\sqrt{\frac{p(\bar{v}|\bar{u})\log(\frac{1}{\delta})}{n_{0,\bar{u}}}} \geq \frac{28\log(\frac{1}{\delta})}{n_{0,\bar{u}}}$ when $n_{0,u} > \frac{49}{3}\frac{\log(\frac{1}{\delta})}{p(\bar{v}|\bar{u})}$, which is guaranteed when $n \geq \frac{98\log(\frac{1}{\delta})}{3\eta \cdot \gamma}$ under the event $\mathcal{E}_{counter}(u)$ and $1 - q_u \geq \gamma$ (i.e., $n_{0,u} \geq \frac{n\gamma}{2}$), inequality (c) is due to Eq. (135), inequality (d) is due to the event $\mathcal{E}_{arm,2}(\bar{v})$, inequality (e) is due to $4\sqrt{3}\sqrt{\frac{p(\bar{v})\log(\frac{1}{\delta})}{n}} \geq \frac{28\log(\frac{1}{\delta})}{n}$ when $n > \frac{49}{3}\frac{\log(\frac{1}{\delta})}{\eta}$, inequality (f) is due to $p(\bar{v}) \geq 8\sqrt{3}\sqrt{\frac{p(\bar{v})\log(\frac{1}{\delta})}{n}}$ when $n \geq \frac{392\log(\frac{1}{\delta})}{\gamma}$, inequality (g) is due to $p(\bar{v}) \geq p(\bar{v}|\bar{u})$, inequality (h) is due to the event $\mathcal{E}_{counter}(u)$, inequality (i) is due to $\bar{q}_u \geq q_u \geq \gamma, 1 - q_u \geq \gamma$.

Combining Eq. (136) and Eq. (151), we have

$$p_{2,uv} - \underline{p}_{uv} \leq 32\sqrt{6}\sqrt{\frac{\log(\frac{1}{\delta})}{\eta \cdot \gamma^3 \cdot n}} \tag{152}$$

Combining Eq. (125), Eq. (132), Eq. (152), the difference then can be bounded by:

$$p_{uv} - \underline{p}_{uv} = p_{uv} - \underline{p}_{1,uv} + \underline{p}_{1,uv} - \underline{p}_{2,uv} + \underline{p}_{2,uv} - \underline{p}_{uv} \tag{153}$$

$$\leq 8\sqrt{3}\sqrt{\frac{\log(\frac{1}{\delta})}{\gamma \cdot n}} \cdot p_{uv} + 8\sqrt{3}\sqrt{\frac{\log(\frac{1}{\delta})}{\eta \cdot \gamma^2 \cdot n}} + 32\sqrt{6}\sqrt{\frac{\log(\frac{1}{\delta})}{\eta \cdot \gamma^3 \cdot n}} \tag{154}$$

$$\leq 48\sqrt{6}\sqrt{\frac{\log(\frac{1}{\delta})}{\eta \cdot \gamma^3 \cdot n}} \tag{155}$$

Combining the above inequality with Eq. (20) yields:

$$\alpha\sigma(S^*; G) - \sigma\left(\hat{S}(\mathcal{D}, \delta); G\right) \tag{156}$$

$$\overset{(a)}{\leq} \alpha B_1 \sum_{(u,v) \in \mathcal{E}} p_{uv}^{\mathbb{D}_{arm}, S^*} \left(p_{uv} - \underline{p}_{u,v}\right) \tag{157}$$

$$\overset{(b)}{\leq} 48\sqrt{6}\alpha B_1 \sqrt{\frac{\log(\frac{1}{\delta})}{\eta \cdot \gamma^3 \cdot n}} \sum_{(u,v) \in \mathcal{E}} p_{uv}^{\mathbb{D}_{arm}, S^*} \tag{158}$$

$$\overset{(c)}{\leq} 48\sqrt{6}\alpha V \sqrt{\frac{\log(\frac{1}{\delta})}{\eta \cdot \gamma^3 \cdot n}} \sum_{(u,v) \in \mathcal{E}} p_{uv}^{\mathbb{D}_{arm}, S^*} \tag{159}$$

$$\overset{(d)}{\leq} 48\sqrt{6} \cdot \alpha V \cdot \sqrt{\frac{\log(\frac{1}{\delta})}{\eta \cdot \gamma^3 \cdot n}} d_{\max}\sigma(S^*; G) \tag{160}$$

where inequality (a) is due to the same derivation of Eq. (20), inequality (b) is due to Eq. (155), inequality (c) is due to $B_1 \leq V$ by Lemma 2 of (Wang & Chen, 2017). For the last inequality (d), since $\sigma(S^*; G) = \sum_{u \in \mathcal{V}} p_u^*$ where $p_u^*$ is the probability node $u$ is triggered by the optimal action $S^*$ and $p_{u,v}^{\mathbb{D}_{\text{arm}}, S^*} = p_u^*$, we have $\sum_{(u,v) \in \mathcal{E}} p_{uv}^{\mathbb{D}_{\text{arm}}, S^*} \leq d_{\max} \sum_{u \in \mathcal{V}} p_u^* = d_{\max} \sigma(S^*; G)$, where $d_{\max}$ is the maximum out-degree.

Finally, we can set $\delta' = \frac{\delta}{12nE}$ so that events $\mathcal{E}_{arm,1}(u), \mathcal{E}_{arm,2}(\bar{v}), \mathcal{E}_{arm,3}(\bar{u}, \bar{v}), \mathcal{E}_{counter}(u), \mathcal{E}_{emp,1}(\bar{v}), \mathcal{E}_{emp,2}(\bar{u}, \bar{v})$ for any $(u, v) \in \tilde{S}^*$ hold with probability at least $1 - \delta'$, by Lemma 9 and taking union bound over all these events and $(u, v) \in \mathcal{E}$. ∎

# I. Auxiliary Lemmas

**Lemma 3** (Hoeffding's inequality). *Let $X_1, ..., X_n \in [0, 1]$ be independent and identically distributed random variables with common mean $\mu$. Let $X = \sum_{i=1}^{n} X_i$. Then, for any $a \geq 0$,*

$$\Pr[|X - n\mu| \geq a] \leq 2e^{-2a^2/n} \tag{161}$$

**Lemma 4** (Multiplicative Chernoff bound). *Let $X_1, X_2, \cdots, X_n$ be independent random variables in $\{0, 1\}$ with $\Pr[X_i = 1] = p_i$. Let $X = \sum_{i=1}^{n} X_i$ and $\mu = \sum_{i=1}^{n} p_i$. Then, for $0 < a < 1$,*

$$\Pr[X \geq (1 + a)\mu] \leq e^{-\mu a^2/3} \tag{162}$$

*and*

$$\Pr[X \leq (1 - a)\mu] \leq e^{-\mu a^2/2} \tag{163}$$

**Lemma 5** (Concentration of the base arm). *Recall that the event $\mathcal{E}_{arm} = \left\{ |\hat{\mu}_i(\mathcal{D}) - \mu_i| \leq \sqrt{\frac{\log(\frac{2mn}{\delta})}{2N_i(\mathcal{D})}} \text{ for any } i \in [m] \right\}$. Then it holds that $\Pr\{\mathcal{E}_{arm}\} \geq 1 - \delta$ with respect to the randomness of $\mathcal{D}$. And under $\mathcal{E}_{arm}$, we have*

$$\mu_i - 2\sqrt{\frac{\log(\frac{2mn}{\delta})}{2N_i(\mathcal{D})}} \leq \hat{\mu}_i(\mathcal{D}) - \sqrt{\frac{\log(\frac{2mn}{\delta})}{2N_i(\mathcal{D})}} \leq \mu_i \tag{164}$$

*for all $i \in [m]$.*

**Proof.**

$$\Pr\{\neg\mathcal{E}_{arm}\} = \Pr\left\{ \exists i \in [m], |\hat{\mu}_i(\mathcal{D}) - \mu_i| \geq \sqrt{\frac{\log(\frac{2mn}{\delta})}{2N_i(\mathcal{D})}} \right\} \tag{165}$$

$$\leq \sum_{i \in [m]} \Pr\left\{ |\hat{\mu}_i(\mathcal{D}) - \mu_i| \geq \sqrt{\frac{\log(\frac{2mn}{\delta})}{2N_i(\mathcal{D})}} \right\} \tag{166}$$

$$= \sum_{i \in [m]} \sum_{j \in [n]} \Pr\left\{ N_i = j, |\hat{\mu}_i(\mathcal{D}) - \mu_i| \geq \sqrt{\frac{\log(\frac{2mn}{\delta})}{2N_i(\mathcal{D})}} \right\} \tag{167}$$

Since $S_t$ are sampled from i.i.d. distribution $\mathbb{D}_\mathcal{S}$, $X_{i,1}, ..., X_{i,j}$ are i.i.d. random variables fixing $i$ and $N_i(\mathcal{D}) = j$. Then we use Lemma 3 to obtain:

$$\Pr\left\{ N_i(\mathcal{D}) = j, |\hat{\mu}_i(\mathcal{D}) - \mu_i| \geq \sqrt{\frac{\log(\frac{2mn}{\delta})}{2N_i(\mathcal{D})}} \right\} \leq 2e^{-2N_i(\mathcal{D})\frac{\log(\frac{2mn}{\delta})}{2N_i(\mathcal{D})}} \leq \frac{\delta}{mn} \tag{168}$$

Combining Eq. (167) gives $\Pr\{\mathcal{E}_{arm}\} \geq 1 - \delta$.

And under $\mathcal{E}_{arm}$, $\hat{\mu}_i(\mathcal{D}) - \sqrt{\frac{\log(\frac{2mn}{\delta})}{2N_i(\mathcal{D})}} \leq \mu_i \leq \hat{\mu}_i(\mathcal{D}) + \sqrt{\frac{\log(\frac{2mn}{\delta})}{2N_i(\mathcal{D})}}$, and rearranging terms gives Eq. (164). ∎

**Lemma 6** (Concentration of the base arm counter). *Recall that the event* $\mathcal{E}_{counter} =$ $\left\{ N_i(\mathcal{D}) \geq \frac{n \cdot p_i^{\mathbb{D}_{arm}, \mathbb{D}_{\mathcal{S}}}}{2} \text{ for any } i \in [m] \,\middle|\, n \geq \frac{8 \log \frac{m}{\delta}}{p^*} \right\}$. *Then it holds that* $\Pr[\mathcal{E}_{arm}] \geq 1 - \delta$ *with respect to the randomness of* $\mathcal{D}$.

**Proof.**

$$\Pr\{\neg \mathcal{E}_{\text{counter}}\} = \Pr\left\{ \exists i \in [m], N_i(\mathcal{D}) \geq \frac{n \cdot p_i^{\mathbb{D}_{\text{arm}}, \mathbb{D}_{\mathcal{S}}}}{2} \,\middle|\, n \geq \frac{8 \log \frac{m}{\delta}}{p^*} \right\} \tag{169}$$

$$\overset{(a)}{\leq} \sum_{i \in [m]} \Pr\left\{ N_i(\mathcal{D}) \geq \frac{n \cdot p_i^{\mathbb{D}_{\text{arm}}, \mathbb{D}_{\mathcal{S}}}}{2} \,\middle|\, n \geq \frac{8 \log \frac{m}{\delta}}{p^*} \right\} \tag{170}$$

$$\overset{(b)}{\leq} \sum_{i \in [m]} e^{-n p_i^{\mathbb{D}_{\text{out}}, \mathbb{D}_{\mathcal{S}}}/8} \tag{171}$$

$$\leq \sum_{i \in [m]} e^{-\frac{8 \log \frac{m}{\delta}}{p_i^{\mathbb{D}_{\text{arm}}, \mathbb{D}_{\mathcal{S}}}} p_i^{\mathbb{D}_{\text{out}}, \mathbb{D}_{\mathcal{S}}}/8} \tag{172}$$

$$\overset{(c)}{\leq} \delta, \tag{173}$$

∎

where inequality (a) is due to the union bound over $i \in [m]$, inequality (b) is due to Lemma 4 by setting $a = 1/2$ with the random $N_i(\mathcal{D})$ being the summation of $n$ i.i.d. Bernoulli random variables with mean $p_i^{\mathbb{D}_{\text{arm}}, \mathbb{D}_{\mathcal{S}}}$, inequality (c) is due to $n \geq \frac{8 \log \frac{m}{\delta}}{p^*} \geq \frac{8 \log \frac{m}{\delta}}{p_i^{\mathbb{D}_{\text{arm}}, \mathbb{D}_{\mathcal{S}}}}$ for any $i \in \tilde{S}^*$.

**Lemma 7** (Concentration of the vector-valued arrival probability). *Recall that the event* $\mathcal{E}_{arv} = \left\{ \|\hat{\boldsymbol{p}} - \boldsymbol{p}\|_1 \leq \sqrt{\frac{2m \log(\frac{2}{\delta})}{n}} \right\}$. *It holds that* $\Pr[\mathcal{E}_{arv}] \geq 1 - \delta$.

The following lemma is extracted from Theorem 14.2 in Lattimore & Szepesvári (2020).

**Lemma 8** (Hardness of testing). *Let $P$ and $Q$ be probability measures on the same measurable space $(\Omega, \mathcal{F})$ and let $A \in \mathcal{F}$ be an arbitrary event. Then,*

$$P(A) + Q(A^c) \geq \frac{1}{2} \exp(-\text{KL}(P, Q))$$

*where $A^c = \Omega \backslash A$ is the complement of $A$.*

**Lemma 9** (Variance-adaptive concentration of the UCBs, LCBs, and counters in the influence maximization application). *It holds that* $\Pr\{\mathcal{E}_{arm,1}(u)\} \geq 1 - 2\delta$, $\Pr\{\mathcal{E}_{arm,2}(\bar{v})\} \geq 1 - 2\delta$, $\Pr\{\mathcal{E}_{arm,3}(\bar{u}, \bar{v})\} \geq 1 - 2n\delta$, $\Pr\{\mathcal{E}_{counter}(u)\} \geq 1 - \delta$, $\Pr\{\mathcal{E}_{emp,1}(\bar{v})\} \geq 1 - \delta$, $\Pr\{\mathcal{E}_{emp,2}(\bar{u}, \bar{v})\} \geq 1 - \delta$, *where*

$$\mathcal{E}_{arm,1}(u) := \left\{ q_u \leq \bar{q}_u \leq \min\left\{ q_u + 4\sqrt{3}\sqrt{\frac{q_u(1 - q_u)\log(\frac{1}{\delta})}{n}} + 28 \cdot \frac{\log(\frac{1}{\delta})}{n}, 1 \right\} \right\} \tag{174}$$

$$\mathcal{E}_{arm,2}(\bar{v}) := \left\{ p(\bar{v}) \leq \bar{p}(\bar{v}) \leq \min\left\{ p(\bar{v}) + 4\sqrt{3}\sqrt{\frac{p(\bar{v})(1 - p(\bar{v}))\log(\frac{1}{\delta})}{n}} + 28 \cdot \frac{\log(\frac{1}{\delta})}{n}, 1 \right\} \right\} \tag{175}$$

$$\mathcal{E}_{arm,3}(\bar{u}, \bar{v}) := \left\{ \max\left\{ p(\bar{v}|\bar{u}) - 4\sqrt{3}\sqrt{\frac{p(\bar{v}|\bar{u})(1 - p(\bar{v}|\bar{u}))\log(\frac{1}{\delta})}{n_{0,\bar{u}}}} - 28 \cdot \frac{\log(\frac{1}{\delta})}{n_{0,\bar{u}}}, 0 \right\} \leq \underline{p}(\bar{v}|\bar{u}) \leq p(\bar{v}|\bar{u}) \right\} \tag{176}$$

$$\mathcal{E}_{counter}(u) := \left\{ n_{0,\bar{u}} \geq \frac{n(1 - q_u)}{2} \,\middle|\, n \geq \frac{8 \log \frac{1}{\delta}}{1 - q_u} \right\}. \tag{177}$$

$$\mathcal{E}_{emp,1}(\bar{v}) := \left\{ \hat{p}(\bar{v}) \leq 2p(\bar{v}) \,\middle|\, n \geq \frac{8 \log \frac{1}{\delta}}{p(\bar{v})} \right\} \tag{178}$$

$$\mathcal{E}_{emp,2}(\bar{u}, \bar{v}) := \left\{ \hat{p}(\bar{v}|\bar{u}) \geq p(\bar{v}|\bar{u})/2 \,\middle|\, n_{0,u} \geq \frac{8 \log \frac{1}{\delta}}{p(\bar{v}|\bar{u})} \right\} \tag{179}$$

**Proof.** For $\Pr\{\mathcal{E}_{arm,1}(u)\} \geq 1 - 2\delta$, $\Pr\{\mathcal{E}_{arm,2}(\bar{v})\} \geq 1 - 2\delta$ they are extracted from Lemma 8 from Liu et al. (2022) without taking union bound on $t, u, v$ as in Liu et al. (2022). For $\Pr\{\mathcal{E}_{arm,3}(\bar{u}, \bar{v})\} \geq 1 - 2n\delta$, it is extracted from Lemma 8 from Liu et al. (2022) by only taking union bound on $n_{0,\bar{u}}$. For $\Pr\{\mathcal{E}_{counter}(u)\} \geq 1 - \delta$, $\Pr\{\mathcal{E}_{emp,1}(\bar{v})\} \geq 1 - \delta$, $\Pr\{\mathcal{E}_{emp,2}(\bar{u}, \bar{v})\} \geq 1 - \delta$, they follow the proof of Lemma 6 without taking union bound on $i \in [m]$. ∎

## J. Detailed Experiments

In this section, we present experiments to assess the performance of our proposed algorithms using both synthetic and real-world datasets. Each experiment was conducted over 20 independent trials to ensure reliability. All tests were performed on a macOS system equipped with an Apple M3 Pro processor and 18 GB of RAM.

### J.1. Offline Learning for Cascading Bandits

We evaluate our algorithm (Algorithm 4) in the cascading bandit scenario by comparing it against the following baseline methods: 1. CUCB-Offline (Chen et al., 2016), an offline variant of the non-parametric CUCB algorithm, adapted for our setting. We refer to this modified version as CUCB-Offline. 2. EMP (Liu et al., 2021), which always selects the action based on the empirical mean of rewards.

**Synthetic Dataset.** We conduct experiments on cascading bandits for the online learning-to-rank application described in Section 4.1, where the objective is to select $K = 5$ items from a set of $m = 100$ to maximize the reward (Dai et al., 2025a). To simulate the unknown parameter $\mu_i$, we draw samples from a uniform distribution $U[0, 1]$ over the interval $[0, 1]$. In each round $t$ of the offline pre-collected dataset, a ranked list $S_t = (a_{t,1}, \dots, a_{t,K}) \subseteq [m]$ is randomly selected. The outcome $X_{t,i}$ for each $i \in S_t$ is generated from a Bernoulli distribution with mean $\mu_i$. The reward at round $t$ is set to 1 if there exists an item $a_{t,k}$ with index $k$ such that $X_{t,a_{t,k}} = 1$. In this case, the learner observes the outcomes for the first $k$ items of $S_t$. Otherwise, if no such item exists, the reward is 0, and the learner observes all $K$ item outcomes as $X_{t,i} = 0$ for $i \in S_t$. Fig. 1a presents the average suboptimality gaps of the algorithms across different ranked lists $n$. The proposed CLCB algorithm outperforms the baseline methods, achieving average reductions in suboptimality gaps of 47.75% and 20.02%, compared to CUCB-Offline and EMP algorithms, respectively. These results demonstrate the superior performance of CLCB in offline environments.

**Real-World Dataset.** We conduct experiments on a real-world recommendation dataset, the Yelp dataset[4], which is collected by Yelp (Dai et al., 2024b). On this platform, users contribute reviews and ratings for various businesses such as restaurants and shops. Our offline data collection process is as follows: we select a user and randomly draw 200 items (e.g., restaurants or shops) that the user has rated as candidates for recommendation. The agent (i.e., the recommender system) attempts to recommend at most $K$ items to the user to maximize the probability that the user is attracted to at least one item in the recommended list. Each item has an unknown probability $\mu_i$, derived from the Yelp dataset, indicating whether the user finds it attractive. Regarding feedback, the agent collects cascading user feedback offline, observing a subset of the chosen $K$ items until the first one is marked as attractive (feedback of 1). If the user finds none of the items in the recommended list $S_t$ attractive, the feedback is 0 for all items. Fig. 1b shows the average suboptimality gaps of different algorithms over $n = 100$ rounds across two action sizes ($K = 4, 8$). Notably, as $K$ changes, the optimal reward also adjusts according to the expected reward $r(S_t; \boldsymbol{\mu}) = 1 - \prod_{i \in S_t}(1 - \mu_i)$, which explains why the suboptimality gap for smaller $K$ tends to be larger compared to that for larger $K$. CLCB achieves the lowest suboptimality gap compared to CUCB and EMP algorithms, demonstrating its strong performance even on real-world data.

### J.2. Offline Learning for LLM Cache

In the LLM Cache scenario, we compare our algorithm (Algorithm 2) against two additional baselines: LFU (Least Frequently Used), which is a caching strategy that evicts the least frequently accessed items to optimize cache usage (Zhu

---

[4]https://www.yelp.com/dataset

et al., 2023), and Least Expected Cost (LEC), which is an advanced caching algorithm that minimizes inference cost by evicting items with the lowest estimated expected cost (Zhu et al., 2023).

**Synthetic Dataset.** For the LLM cache application described in Section 4.2, we simulate the scenario using 100 distinct queries and set the cache size to 40. Consistent with (Zhu et al., 2023), the frequency distribution follows a power law with $\alpha = 0.9$, and the ground truth cost for each query processed is drawn from a Bernoulli distribution with parameter 0.5. The simulation is repeated 20 times to ensure robustness, and we report the mean and standard deviation of the results across different dataset sizes $n = \{2, 4, 8, 16, 32, 64, 128, 256, 512, 1024, 2048\}$ in Fig. 3a. Our normalized results suggest that CLCB-LLM-C significantly outperforms the baseline algorithms, LFU and LEC, achieving an average improvement of $1.32\times$. These results highlight the effectiveness of CLCB-LLM-C in optimizing cache performance for LLM applications.

**Real-World Dataset.** We use the SciQ dataset (Welbl et al., 2017), which covers a variety of topics, including physics, chemistry, and biology, to evaluate the performance of our proposed CLCB-LLM-C algorithm using OpenAI's LLMs. The cost is defined as the price for API calls, based on OpenAI's official API pricing. Since the cost heavily depends on the token count of the input text, we utilize OpenAI's `tiktoken` library, designed to tokenize text for various GPT models. We consider two different LLMs with distinct encoding strategies. Specifically, we use GPT-4-o with the "o200k_base" encoding to present the main experimental results. Additionally, we experiment with another variant, GPT-4-turbo, which employs the "cl100k_base" encoding (OpenAI, 2025). For the evaluation, we work with 100 distinct prompts from the SciQ dataset in an offline setting, performing a total of 10,000 queries with cache sizes of $K = 10$ and

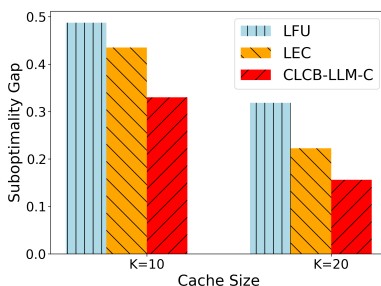

*Figure 4.* Algorithms on another LLM.

$K = 20$, respectively. Fig. 3b presents the normalized suboptimality gap of cost over $n = 100$ rounds. CLCB-LLM-C achieves 36.01% and 20.70%, less cost compared to LFU and LEC, respectively. Moreover, a larger $K$ shows a lower suboptimality gap, which is consistent with Theorem 3. In addition to the results presented in the main text using GPT-4-o with the "o200k_base" encoding, we experiment with another LLM, GPT-4-turbo with the "cl100k_base" encoding. Fig. 4 demonstrates the robustness of our algorithm across different LLMs.

