# OpenReview forum: "Offline Learning for Combinatorial Multi-armed Bandits"
_ICML.cc/2025/Conference — ICML 2025 poster_

### Official Review · Reviewer_qESa · 2025-03-06

**Overall Recommendation:** 1

**Summary:**

The authors study a problem within the combinatorial multi-armed bandit (CMAB) setting, in the presence of offline datasets. The authors introduce Off-CMAB, the first offline learning framework for CMAB. The authors propose the combinatorial lower confidence bound (CLCB) algorithm, which combines pessimistic reward estimations with approximation algorithms and propose two data coverage conditions and prove that, under these conditions, CLCB achieves a near-optimal suboptimality gap, matching the theoretical lower bound up to a logarithmic factor. The authors validate Off-CMAB through various applications, including learning to rank, large language model caching, and social influence maximization.

## update after rebuttal

We expressed some concerns clearly in both the review and rebuttal. Unfortunately, the authors appear to have ignored our suggestions and continued to respond with incorrect or misleading assertions. We are not dismissing the contributions of this work. We recognize that semi-bandit feedback and offline learning are both legitimate and practical problem settings, and the paper contributes in these directions. However, our key concern remains: the over-claiming, wrong statements (semi-bandit settings allow for tighter and even exact approximation guarantees), and mischaracterization of related work. Hence, we downgrade the initial score.

**Claims And Evidence:**

The claims made in the submission were supported by proved theoretical results and were validated by experiments.
However, some claims should be relaxed such as: “We validate Off-CMAB through practical applications, … , showing its ability to handle nonlinear reward functions, general feedback models” Should be rather “We validate Off-CMAB through practical applications, … , showing its ability to handle ““some”” nonlinear reward functions”. For example their approach cannot handle submodular reward functions under bandit-feedback (where the reward function is black-box).

**Essential References Not Discussed:**

The paper does not cite recent related works on combinatorial bandits, which similarly rely on offline approximation algorithms (ORACLE) [1-7], dealing with non-linear rewards (for submodular [2, 4, 6] and general rewards [1, 3, 5, 7]), some of which study the same problem of social influence maximization [3, 4].

(minor) Moreover, the authors compare in the experiments to some approaches such as EMP [8], which was cited in the main paper but not discussed or cited within the related works. EMP is directly mentioned in the experiments without an introduction or explanation.

**Reference**

[1] Niazadeh, R., Golrezaei, N., Wang, J. R., Susan, F., & Badanidiyuru, A. (2021, July). Online learning via offline greedy algorithms: Applications in market design and optimization. In Proceedings of the 22nd ACM Conference on Economics and Computation (pp. 737-738).

[2] Fourati, F., Aggarwal, V., Quinn, C., & Alouini, M. S. (2023, April). Randomized greedy learning for non-monotone stochastic submodular maximization under full-bandit feedback. In International Conference on Artificial Intelligence and Statistics (pp. 7455-7471). PMLR.

[3] Nie, G., Nadew, Y. Y., Zhu, Y., Aggarwal, V., & Quinn, C. J. (2023, July). A framework for adapting offline algorithms to solve combinatorial multi-armed bandit problems with bandit feedback. In International Conference on Machine Learning (pp. 26166-26198). PMLR.

[4] Fourati, F., Quinn, C. J., Alouini, M. S., & Aggarwal, V. (2024, March). Combinatorial stochastic-greedy bandit. In Proceedings of the AAAI Conference on Artificial Intelligence (Vol. 38, No. 11, pp. 12052-12060).

[5] Fourati, F., Alouini, M. S., & Aggarwal, V. (2024, July). Federated Combinatorial Multi-Agent Multi-Armed Bandits. In International Conference on Machine Learning (pp. 13760-13782). PMLR.

[6] Sun, X., Guo, T., Han, C., & Zhang, H. (2025). Greedy algorithms for stochastic monotone k-submodular maximization under full-bandit feedback. Journal of Combinatorial Optimization, 49(1), 1-25.

[7] Oki, T., & Sakaue, S. (2025). No-Regret M ${}^{\natural} $-Concave Function Maximization: Stochastic Bandit Algorithms and NP-Hardness of Adversarial Full-Information Setting. Advances in Neural Information Processing Systems, 37, 57418-57438.

[8] Liu, X., Zuo, J., Chen, X., Chen, W., and Lui, J. C. Multi-layered network exploration via random walks: From offline optimization to online learning. In Interna tional Conference on Machine Learning, pp. 7057–7066. PMLR, 2021.

**Experimental Designs Or Analyses:**

The experiments seem fine. However, they lack some details (even in the appendix). For example, while it is mentioned in the Appendix for the LLM experiment that the number of repetitions is 20 times. For the other ones this detail was not mentioned.

**Methods And Evaluation Criteria:**

The considered methods make sense for the problem at hand.

**Other Comments Or Suggestions:**

It should be clear from the beginning (perhaps even from the abstract) whether the considered algorithm is for bandit feedback or semi-bandit feedback settings.

**Other Strengths And Weaknesses:**

**Strengths**

(1)	The authors study an important problem in machine learning which is combinatorial bandits.

(2)	The authors propose CLCB algorithm and derive its theoretical guarantees.

(3)	The authors propose data coverage conditions and prove that, under these conditions, the proposed algorithm is near-optimal matching the lower bound regret guarantees up to a logarithmic term.

(4)	The proposed algorithm is assessed empirically against three different datasets.

**Weaknesses**

(1) *Novelty*: The algorithm is very similar to the online CMAB-T approaches [10, 11] with the key modification to rely on the principle of pessimism, which is borrowed from previous works [12]. Moreover, the idea to use pessimism in the offline bandits can be found in previous bandits works such as [13].

(2) *Limitation in the Algorithm*: While the CLCB algorithm is proposed to consider a given approximation oracle (Algorithm 1, line 7), the oracle is constrained to require as input the estimated arm rewards. However, several optimal approximation oracles do not work this way, as they require the set of arms as input rather than the estimated value of each arm, such as the approximation oracle adapted in the references above [1-7]. For example, the greedy algorithm in [8] or the stochastic-greedy algorithm in [9] cannot be used, unlike the frameworks in [3, 5] or the specialized algorithms in [2, 4, 6], which employ oracles that do not require the estimated reward of arms.

(4) (minor) *Limitation in the Considered Setting*: The paper considers semi-bandit feedback, which is of interest to the community. However, several problems require full-bandit feedback (also called bandit feedback), which limits the applicability of the approach in several settings.

(5) (minor) *Limitation in the Considered Setting*: The paper assumes the presence of an offline data, which can sometimes be available, however, not always, which limits the applicability of the approach in various settings.

**References**

[1-7] *See references above.*

[8] Nemhauser, G. L., Wolsey, L. A., & Fisher, M. L. (1978). An analysis of approximations for maximizing submodular set functions—I. Mathematical programming, 14, 265-294.

[9] Mirzasoleiman, B., Badanidiyuru, A., Karbasi, A., Vondrák, J., & Krause, A. (2015, February). Lazier than lazy greedy. In Proceedings of the AAAI Conference on Artificial Intelligence (Vol. 29, No. 1).

[10] Chen, W., Wang, Y., & Yuan, Y. (2013, February). Combinatorial multi-armed bandit: General framework and applications. In International conference on machine learning (pp. 151-159). PMLR.

[11] Wang, Q., & Chen, W. (2017). Improving regret bounds for combinatorial semi-bandits with probabilistically triggered arms and its applications. Advances in Neural Information Processing Systems, 30.

[12] Jin, C., Yang, Z., Wang, Z., & Jordan, M. I. (2020, July). Provably efficient reinforcement learning with linear function approximation. In Conference on learning theory (pp. 2137-2143). PMLR.

[13] Li, G., Ma, C., & Srebro, N. (2022). Pessimism for Offline Linear Contextual Bandits using $\ell_p $ Confidence Sets. Advances in Neural Information Processing Systems, 35, 20974-20987.

**Questions For Authors:**

In the presence of offline data and the possibility of online learning, we can always use the offline data to warm up the learning agent and then start standard online learning. Given this, how practical is the off-CMAB setting? What types of problems do not allow for any amount of online learning?

How would your approach perform with submodular rewards? Is there any discussion on extensions to bandit feedback settings?

**Relation To Broader Scientific Literature:**

This work studies a problem at the intersection of offline learning and combinatorial bandits literatures. While there are several works on offline learning and several other works on online bandits. A few works studies offline bandits. This work proposes a first framework for offline combinatorial bandits. The work derives theoretical guarantees for the proposed framework and shows, under some condition, near-optimal regret matching the lower bound with logarithmic factor.

**Theoretical Claims:**

Checked the proof ideas and briefly skimmed through the proof.

---

> ### Author Rebuttal · Authors · 2025-03-31
>
> We thank the reviewer for their comments regarding the claims of our paper and the comparison with related work.
>
> **Q1. Clarification on our claim regarding nonlinear reward functions and general feedback models.**
>
> **A1.** We appreciate the reviewer’s suggestion. In the final version, we will clarify in both the abstract and introduction that our framework addresses CMAB problems with semi-bandit feedback under nonlinear reward functions, where the oracles operate on arm-level estimations.
>
> **Q2. Missing references and comparison with combinatorial bandit under full-bandit feedback works.**
>
> **A2.** We appreciate the reviewer pointing out the relevant line of work on online combinatorial bandits with full-bandit feedback [1–7]. We will include these references in the related work section. However, we would like to emphasize two key distinctions:
>
> (1) Online vs. offline setting: The cited works [1–7] focus on online learning, while our work addresses the offline CMAB setting, which poses different challenges and requires different algorithmic strategies.
>
> (2) Semi-bandit vs. full-bandit feedback: Our setting assumes semi-bandit feedback, where the learner observes individual arm-level feedback for selected arms (i.e., components of the super arm). This enables more informative learning and allows us to construct accurate base-arm estimators for use in our oracles, leading to an $O(T^{1/2})$ regret bound. In contrast, full-bandit feedback only provides aggregate rewards for the entire super arm, often resulting in higher regret (e.g., $O(T^{2/3})$) and requiring fundamentally different oracle designs. Moreover, prior full-bandit approaches often rely on additional structural assumptions such as submodularity to achieve these bounds. Similarly, our approach leverages smoothness assumptions on the reward function to ensure statistical efficiency in the offline regime.
>
> We agree that assuming semi-bandit feedback precludes applying our work to some settings, but we point out that such feedback is available in a wide range of applications of interest, and that both full-bandit CMAB [1-7] and semi-bandit CMAB (Gai et al., 2012; Kveton et al., 2015c; Combes et al., 2015; Chen et al., 2016; Wang
> \& Chen, 2017; Merlis \& Mannor, 2019; Saha \& Gopalan, 2019; Agrawal et al., 2019; Liu et al., 2022; 2024, Zimmert et al., 2019; Ito, 2021; Tsuchiya et al., 2023.) have long been studied as distinct lines of work in the literature.
> Our choice of the semi-bandit model is motivated by its natural presence in many real-world domains (beyond the three applications already discussed in this paper), including:
>
> - Online recommendation: Click/no-click feedback is available per item in the recommended list.
>
> - Online routing: The delay or cost of each edge in a chosen path can be observed.
>
> - Crowdsourcing: The quality of individual workers’ contributions is directly measurable or computable.
>
> These applications offer fine-grained, arm-level feedback, making the semi-bandit setting not only realistic but essential. Nevertheless, we agree that exploring offline CMAB under full-bandit feedback is a compelling future direction and will add such a discussion to the paper.
>
> **Q3. On the practicality of the offline CMAB setting.**
>
> **A3.**
> We respectfully refer the reviewer to our response for the Q1 asked by Reviewer mMZh, where we discuss motivating applications and the relevance of the offline CMAB setting in real-world systems.
>
>
>
> **Q4. On combining offline and online CMAB.**
>
> **A4.**
> Please refer to our response for Q3 asked by Reviewer mMZh, where we outline our vision for hybrid offline-online CMAB approaches as a promising future direction.
>
> **Q5. On the novelty and technical contribution of our work.**
>
> **A5.**
> We respectfully refer the reviewer to our response for Q1 asked by Reviewer Feh9, which provides a detailed discussion of the challenges addressed and the key innovations of our framework and theoretical results.
>
>
> **Q6. Clarification on experimental details**
>
> **A6.**
>  As noted in Lines 1962-1963, at the beginning of the experimental setup, each experiment was conducted over 20 independent trials to ensure reliability. This applies to both the cascading bandit scenario and the LLM cache scenario. To avoid ambiguity, we will explicitly reiterate this detail in the final version of Section 5.

---

> > ### Comment · Reviewer_qESa · 2025-04-03
> >
> > We thank the authors for their response.
> >
> > Regarding your answer to A2 (missing references and comparison with combinatorial bandit methods under full-bandit feedback), we would like to emphasize that we are fully aware of the distinction between both settings. The semi-bandit setting is a special case of the full-bandit setting, while the former assumes access to additional feedback, the latter does not. Hence, full-bandit feedback approaches remain applicable to your semi-bandit feedback setting. We acknowledge that some applications, along with the aggregate reward, may provide fine-grained, arm-level feedback, making the semi-bandit setting realistic. **However, this does not imply that it is essential (as claimed by the authors)—full-bandit feedback can still be effectively used in such settings. In fact, for certain problems—particularly those involving non-linear reward structures, as claimed in your paper—the non-linear dependencies between arms in some cases may necessitate reliance on aggregate rewards.** This observation directly relates to the second weakness (W2) we previously raised.
> >
> > Unfortunately, **the authors did not respond to this second weakness.**
> >
> > Moreover, **the authors did not adequately address our Q2.** We reiterate that in the presence of offline data and the possibility of online learning, one can always use the offline data to warm-start the learning agent and then proceed with standard online learning. Given this, we question the practical relevance of the offline-CMAB setting. In other words, when online learning is allowed, the role of offline learning becomes negligible over a long time horizon $T$. What types of problems fundamentally eliminates any form of online learning? While the combination of offline and online appraoches can lead to better outcomes, it is unclear how/when significant this acceleration is in practice. Again, for sufficiently large horizons, the initialization from offline learning becomes negligible.
> >
> > **Conclusion.** The authors did not respond to our W2, did not adequately address our Q2, and should provide a more rigorous treatment of the full-bandit feedback setting, including a precise characterization of the non-linear reward structures their approach can handle (e.g., can it handle submodular rewards?).

---

> > > ### Author Response · Authors · 2025-04-05
> > >
> > > **Response to: “Full-bandit feedback approaches can still be effectively used in semi-bandit settings.”**
> > >
> > > We disagree with the implication that full-bandit approaches are equally effective in semi-bandit settings. While full-bandit algorithms *can* technically be applied by discarding the additional arm-level feedback, doing so results in a loss of valuable information and typically worse performance. For instance, full-bandit feedback often leads to regret bounds of $ O(T^{2/3}) $, while semi-bandit feedback enables tighter bounds like $ O(\sqrt{T}) $. Moreover, full-bandit methods frequently only guarantee *approximate* regret bounds (e.g., $ 1 - 1/e $ approximation for submodular maximization), whereas semi-bandit settings allow for tighter and even *exact* guarantees.
> > >
> > > Hence, full-bandit algorithms are not only suboptimal in semi-bandit settings—they are fundamentally mismatched for the problem structure. These are not interchangeable regimes; they differ in feedback richness, learning potential, and algorithmic design. Our work focuses on the semi-bandit setting because (1) it arises naturally in many real-world applications (as we show in the paper), and (2) it enables significantly stronger theoretical results. We hope the reviewer can acknowledge this distinction and agree that semi-bandit feedback is not merely a special case, but an important setting in its own right.
> > >
> > > ---
> > >
> > > **Response to Weakness 2**
> > >
> > > The reviewer notes our algorithm’s reliance on an oracle that selects actions based on individual base-arm estimates, contrasting it with oracles used in full-bandit feedback settings. This distinction reflects the fundamental difference between the settings. Our oracle is designed specifically for the semi-bandit context, where arm-level feedback enables better decision-making and regret bounds. In contrast, full-bandit oracles operate under more limited feedback and typically allow only approximate solutions.
> > >
> > > This should not be seen as a *limitation* of our approach. Rather, it is a principled design choice aligned with the feedback structure in our setting, enabling stronger theoretical guarantees. Just as full-bandit methods develop oracles to match their own constraints, we design ours to exploit the additional information semi-bandit feedback provides. Thus, we disagree with this being viewed as a weakness—it is an effective and necessary adaptation.
> > >
> > > ---
> > >
> > > **On the importance of offline learning vs. online learning**
> > >
> > > The reviewer claims that “when online learning is allowed, the role of offline learning becomes negligible over a long time horizon $ T $.” While this statement is reasonable in theory, the core issue we address lies in the assumptions: (a) online learning is allowed, and (b) $ T $ is large. As we argue in the paper and in our prior response (see Q1 of Reviewer mMZh), online learning may not be available because the platform does not support tight feedback loops and data may be outsourced to a third party for processing, precluding any real-time feedback.
> > >
> > > Moreover, the long-horizon assumption does not always hold—for instance, in systems with periodic model updates based on fixed-length logs (e.g., using one week's logged data to update the model each week). As such, offline learning is not only useful but necessary. Therefore, we believe our work addresses a practically important and underexplored setting, and it is inappropriate to dismiss it by appealing to results from different regimes.
> > >
> > > ---
> > >
> > > **On the class of reward functions our approach supports**
> > >
> > > In the problem setup (line 132), we clearly define the reward function class we can handle—those satisfying (1) monotonicity and (2) 1-norm TPM smoothness (Conditions 1 and 2). These encompass a broad family of structured, non-linear reward functions, including submodular ones. The examples in our applications (e.g., Learning to Rank in Section 4.1 and Influence Maximization in Section 4.3) are submodular and satisfy both conditions. Thus, our framework is applicable to a rich class of meaningful reward structures.
> > >
> > > ---
> > >
> > > **Summary**
> > >
> > > We believe that comparing our work on *offline learning with semi-bandit feedback* to studies in *online learning with full-bandit feedback* conflates distinct problem settings. These differ in assumptions, information structure, and algorithmic needs. We respectfully suggest that a fair evaluation of our contribution should consider two questions:
> > >
> > > (a) Is the offline semi-bandit setting a reasonable and important setting worth studying, from an application perspective?
> > >
> > > (b) Does our work meaningfully advance the state of the art in this setting?
> > >
> > > To both, the answer is clearly yes. We introduce new algorithms and tighter theoretical bounds specifically tailored to the offline semi-bandit setting—a scenario that arises frequently in modern applications. Dismissing this contribution using results from incompatible setups fails to recognize the core motivation and technical novelty of our work.

---

### Official Review · Reviewer_Feh9 · 2025-03-07

**Overall Recommendation:** 3

**Summary:**

This paper proposes a framework of offline learning for combinatorial multi-armed bandit (Off-CMAB). The authors first provide an algorithm, CLCB, based on constructing the lower confidence bound for each base arms from the offline dataset. In order to theoretically measure the performance of the algorithm in terms of sub-optimality gap with sample complexity (size of the offline dataset), they provides two data coverage conditions to qualify the quality of the dataset: (1) Infinity-norm TPM Data Coverage, and (2) 1-norm TPM Data Coverage. The paper then provides theoretical upper bound on the sub-optimality gap of their CLCB by the two conditions, respectively, and shows that the bound regarding the condition (1) is tight under the special instance, $k-$path problem, up to log factors. Finally, the authors introduces three applications of their framework: Cascading Bandits, LLM cache Bandit, and Influence Maximization, with numerical experiments for the first two applications.

**Claims And Evidence:**

Yes. Very clear.

**Essential References Not Discussed:**

N/A

**Experimental Designs Or Analyses:**

Yes.

**Methods And Evaluation Criteria:**

Yes.

**Other Comments Or Suggestions:**

See Strengths And Weaknesses.

**Other Strengths And Weaknesses:**

Strengths:
1. This paper proposes a clear and general framework for offline combinatorial bandits.
2. This paper provides two novel data coverage conditions to measure the quality of offline datasets.
3. This paper provides insights of their model to LLM related problem.
4. This paper is completed and well-written. The authors clearly conveys and validates their message.

Weaknesses and Questions:
1. There are some typos (for example in remark 4 Line 233).
2. I am quite curious about the technical novelty. For me, the algorithm design and theoretical analysis (theorem 1) is natural and a bit "simple". Specifically, although there is no work studying offline learning in combinatorial bandit previously, it is quite natural to directly leverage the idea of pessimism principle to penalize the arms that have not been explored enough. Based on this principle, the corresponding theoretical analysis is also straightforward. In my opinion, the proof for theorem 1 and 2 consists of proper use of standard techniques and smart extraction of key influencing factors (data coverage condition).

Comments:
1. Given that the paper provides some new insights of data coverage conditions and provides some new connection between traditional combinatorial bandits and LLM, I believe the paper should be accepted, although I am not quite sure whether the technical novelty is enough. I will be happy to further increase my score if I miss something important regarding the technical novelty.

**Questions For Authors:**

See Strengths And Weaknesses.

**Relation To Broader Scientific Literature:**

N/A

**Theoretical Claims:**

Yes.

---

> ### Author Rebuttal · Authors · 2025-03-31
>
> We thank the reviewer for their positive feedback about our work's insights and on building connection between traditional combinatorial bandits and LLM.
>
>
> **Q1. On the technical challenges and novelty.**
>
> **A1.** Our main contribution is a **general and minimalistic framework for offline learning in CMAB**, which we validate through three important and diverse applications. While the idea of pessimism is widely used in offline bandits and RL, its effectiveness and applicability to **combinatorial bandits with large action spaces**—a setting motivated by many real-world applications—remains unclear. Key open questions include whether pessimism works in such settings, under what conditions it succeeds, and how various structural factors influence performance. Our work provides the first steps toward addressing these questions.
>
> From a **framework perspective**, we propose the general **offline CMAB-T framework**, and introduce two novel data coverage conditions (Conditions 3 and 4), i.e., the **1-norm and infinity-norm TPM data coverage conditions**. These conditions are minimal yet insightful, offering a practical way to assess dataset quality when this offline dataset is collected under an aribtrary policy. A critical technical novelty is the use of $p_i^{D\_{\text{arm}}, S^*}$, the arm-wise observation probability under the optimal super arm, as an importance weight. Naively defining coverage without this weighting would incur an additional $1/p^*$ factor in the suboptimality gap. Furthermore, unlike what might be intuitively expected, our framework only requires this importance weight with respect to the **optimal super arm** $S^*$, rather than all super arms, making the condition weaker but still sufficient for achieving near-optimal guarantees.
>
> In **Theorems 1 and 2**, we intentionally present the analysis in a clean and minimalist form to maximize accessibility and generality. Many of the technical challenges are addressed in the **application-specific sections**:
>
> - In the **LLM cache** scenario, it is nontrivial to verify that the problem satisfies the data coverage conditions. Moreover, our analysis must jointly handle the **full-feedback arrival probability** and **semi-bandit feedback cost**, which interact in subtle ways. Our improvements (Theorem 3) over Zhu et al. stem from both satisfying the conditions and carefully handling these feedback structures.
>
> - In the **influence maximization** application, we integrate **variance-adaptive confidence intervals** and consider **node-level feedback**, achieving state-of-the-art results. While these techniques (e.g., constructing confidence intervals for intermediate random variables using Bernstein-type concentration) could be unified under the general framework, we choose to separate them for clarity. This design decision keeps Theorems 1 and 2 clean, but it may mask some of the technical depth—such as advanced estimation strategies and additional uncertainty terms arising from variance-based bounds.
>
> In summary, although our general results are presented in a minimal form, they are backed by nontrivial challenges when instantiated in realistic applications.  We will add a discussion of how these application-motivated extensions could be generalized into the main results to the paper.
>
> **Q2. About the typo in Remark 4.**
>
> **A2.** We thank the reviewer for pointing out the typo in Line 233. We will correct it in the final version of the paper.

---

> > ### Comment · Reviewer_Feh9 · 2025-04-03
> >
> > Thank you for your answer! I believe it is a solid paper and I will keep my score.

---

> > > ### Author Response · Authors · 2025-04-03
> > >
> > > Thank you for your encouraging response and for taking the time to read our rebuttal! We're glad to hear that you find the paper solid and appreciate your continued support.
> > >
> > > If you have any remaining concerns or specific questions regarding our explanation of the technical novelty, we would be happy to clarify further. Otherwise, if you feel that your main concern has been satisfactorily addressed, we kindly ask if you would consider increasing your score to reflect this.
> > >
> > > Thank you again for your thoughtful review and constructive feedback!

---

### Official Review · Reviewer_mMZh · 2025-03-17

**Overall Recommendation:** 3

**Summary:**

This paper studies the offline version of the combinatorial multi-arm bandit (CMAB) problem, which is different from most CMAB papers that consider the online version. The authors provide solid theoretical results on the sample complexity of the offline CMAB problem that is minimax optimal. Moreover, the authors demonstrate the offline CMAB framework through three applications: cascading bandits, LLM caching, and influence maximization. The authors present both theoretical guarantees and numerical results for these applications (except for influence maximization).

**Claims And Evidence:**

Yes, the main contributions are clearly stated and justified.

**Essential References Not Discussed:**

To better motivate the study of offline bandits, a literature review of papers discussing the complexity and value of offline learning in MAB settings is necessary.

**Experimental Designs Or Analyses:**

In Section 5 (Experiments), discussions on the choice of benchmark algorithms are missing.

**Methods And Evaluation Criteria:**

The benchmark datasets and synthetic datasets in Section 5 are reasonable.

**Other Comments Or Suggestions:**

In general, the authors need a stronger justification of why a complete offline learning framework for CMAB is necessary since the problem seems to be just some kind of a best subset selection problem in one shot. For the LLM caching problem, both the offline and the online learning approach would make sense [1].

It will be helpful for the authors to discuss how to leverage the offline data in the online learning setting and how to adapt the offline learning algorithm into an online learning algorithm

[1] Zhu, Banghua, et al. "Towards Optimal Caching and Model Selection for Large Model Inference." Advances in Neural Information Processing Systems 36 (2023): 59062-59094.

**Other Strengths And Weaknesses:**

The paper is very well-written. All the mathematical statements are written in a clear manner, and the paper itself also presents a complete story: a new theory of offline CMAB, extensive applications, and experiments.

One major weakness is the lack of motivation of the offline CMAB. I believe it is more natural to consider the online version of bandits since they are naturally experimentation methods. In the Introduction section, the authors use the healthcare system as an example where experimentation might be infeasible, but they work on applications other than healthcare systems later.

**Questions For Authors:**

As mentioned above, can the authors give a stronger justification of the significance of the offline CMAB problem?

**Relation To Broader Scientific Literature:**

I can see the work relates to many pieces of literature, i.e., offline RL, combinatorial bandits, cascade bandit, LLM caching optimization, and influence maximization. The work offers new insights into efficient RL with combinatorial action spaces (with special structure), and suggests some novel new applications.

**Theoretical Claims:**

I followed the proofs of Theorem 1 & 2, which are the main theoretical results of the paper. They are correct.

---

> ### Author Rebuttal · Authors · 2025-03-31
>
> We thank the reviewer for recognizing our work's connection to a broad literature, which offers new insights for RL with combinatorial action spaces and novel new applications.
>
> **Q1. About the motivation of studying the offline CMAB problem.**
>
> **A1.** While online bandits are a natural choice when online data is readily available and inexpensive, many real-world applications restrict access to only offline data as follows, which motivates the study of offline CMAB.
>
> For instance, consider the cascading bandit model in recommendation systems. Online CMAB learning requires a tight feedback loop where the platform (i.e., the learner) updates its recommendation policy after every user interaction. However, in many practical scenarios, such fine-grained online feedback is unavailable as the platform cannot afford to update at such a high frequency. Instead, data is collected in batches (e.g., over a week), logged, and then used to update the policy in a single offline training phase. This workflow aligns precisely with our offline CMAB setting.
>
> Another motivating scenario involves outsourced system design. For example, if OpenAI or Anthropic outsources the design of an LLM caching system, the consultant (i.e., the learner) typically receives only anonymized user logs. They must learn user behavior and design the system purely based on this private offline dataset and cannot reach out for direct interaction with the users, which fits naturally into the offline CMAB framework.
>
> Moreover, our work on CMAB also mirrors the development trajectory in reinforcement learning (RL). RL began with a focus on online learning [1]; then, around 2020, concerns over the cost and availability of online interactions led to a growing emphasis on offline RL—learning solely from offline data [2]. More recently, hybrid approaches [3] combining offline pretraining with online fine-tuning have emerged. Similarly, after establishing foundational results in online CMAB, we now focus on the offline setting as a crucial step toward enabling future hybrid CMAB approaches.
>
> We will incorporate this discussion and examples into the final version of the paper.
>
> References:
>
> [1] Mnih et al., Playing Atari with Deep Reinforcement Learning, arXiv:1312.5602 (2013).
>
> [2] Levine et al., Offline Reinforcement Learning: Tutorial, Review, and Perspectives, arXiv:2005.01643 (2020).
>
> [3] Lee et al., Offline-to-Online Reinforcement Learning via Balanced Replay and Pessimistic Q-Ensemble, CoRL, 2022.
>
> **Q2.  Justification of baseline algorithms.**
>
> **A2.** In the revised manuscript, we will include a dedicated paragraph in Section 5 to justify our baselines. Specifically, as detailed in Appendix J.1, we evaluate our proposed Algorithm 1 in the cascading bandit scenario by comparing it against the following baselines: (1) CUCB-Offline [1]: An offline variant of the non-parametric Combinatorial Upper Confidence Bound (CUCB) algorithm, adapted to our setting by changing the LCB in line 5 of Algorithm 1 to its UCB counterpart. This baseline is chosen because it represents a well-established approach in combinatorial multi-armed bandit problems.  (2) EMP [2]: A method that selects actions based on the empirical mean of rewards. We include EMP as a simple yet effective baseline that relies on historical data without sophisticated exploration.
>
> In the LLM cache scenario, we compare our Algorithm 2 against:  (1) LFU (Least Frequently Used) [3]: A caching strategy that evicts the least frequently accessed items to optimize cache usage. This is a standard baseline widely used in caching.  (2) LEC (Least Expected Cost) [3]: An advanced caching algorithm that minimizes inference cost by evicting items with the lowest estimated expected cost. We include LEC as it directly aligns with our objective of optimizing inference cost, serving as a strong competitor to our approach.
>
> References:
>
> [1] Chen et al. "Combinatorial multi-armed bandit and its extension to probabilistically triggered arms." JMLR, 2016.
>
> [2] Liu et al. "Multi-layered network exploration via random walks: From offline optimization to online learning." ICML, 2021.
>
> [3] Zhu et al. "On optimal caching and model multiplexing for large model inference." arXiv, 2023.
>
> **Q3. Discussion on how to use offline bandit learning to facilitate online bandit learning.**
>
> **A3.** Our offline CMAB framework provides a solid foundation for developing hybrid offline-online CMAB methods. One promising future direction is to leverage the output of offline CMAB as a baseline policy to ensure stable average performance, while selectively conducting targeted exploration over a small set of promising policies identified during offline training. This strategy can significantly accelerate online learning by reducing the exploration space and focusing only on high-potential actions, effectively combining the strengths of both offline and online approaches. We will add this discussion in the final version.

---

### Decision · Program_Chairs · 2025-05-01

**Decision:**

Accept (poster)

**Comment:**

The paper studies the offline setting of combinatorial bandits problem with semi-bandit feedback. They consider non-linear reward functions and employ an offline optimization oracle that enjoys approximate guarantees. The main contribution of this work are necessary and sufficient conditions for when learning (using pessimism) works. Given the familiar tools used, some reviewers questioned how novel the approach actually was but most were convinced of the final contribution. For a theoretical paper, this paper placed an unusual focus on applications to bolster its claims of its coverage conditions and other assumptions--this is well appreciated.